# ROBUST TIME SERIES FORECASTING VIA BASIS-ALIGNED SAMPLING IN DECYCLED RESIDUAL SPACE

## ABSTRACT

Time series forecasting is crucial in domains such as finance, energy, and traffic, yet real-world data are often contaminated by anomalies and noise. In this work, we first identify a fundamental limitation of existing approaches—their excessive reliance on specific input points, particularly the most recent observation—which makes them highly susceptible to point-wise perturbations and undermines prediction reliability. To further address this challenge, we propose RESAM, a novel approach for robust time series forecasting that effectively mitigates the impact of point-wise perturbations while maintaining high overall forecasting accuracy. RESAM utilizes a basis-aligned randomized sampling strategy to comprehensively exploit the global context and achieve a unified representation for irregularly sampled sequences. Moreover, RESAM employs a learnable periodicity extraction module with a two-stage training protocol to enhance the accuracy and robustness of both periodicity and residual learning. Comprehensive evaluations on eight benchmark datasets show that RESAM achieves competitive forecasting accuracy and significantly surpasses state-of-the-art models in robustness to point-wise perturbations.

## 1 INTRODUCTION

Time series data are widespread across various areas such as economics, energy, environment, traffic, and AIOps (Faloutsos et al., 2018; Wen et al., 2022; Zhao et al., 2023; Qiu et al., 2025). In these areas, time series forecasting (TSF)—which leverages historical values to predict future values—plays a crucial role (Qiu et al., 2024). Its accuracy is critical as it enhances advancements and efficiencies in numerous applications within these fields.

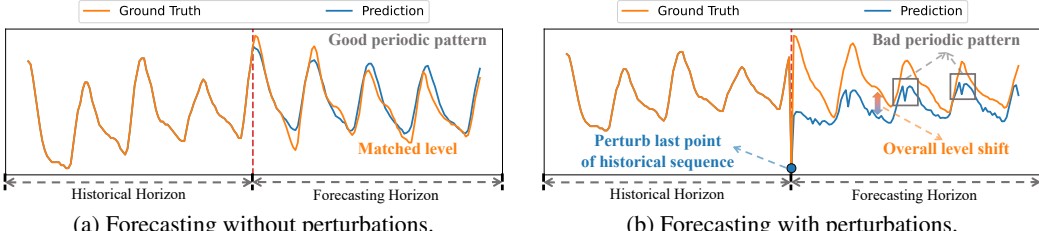

(a) Forecasting without perturbations.          (b) Forecasting with perturbations.

Figure 1: An example of perturbation and prediction on ETTh2 by a single-layer linear model. The blue point in Fig. 1b represents an out-of-distribution perturbation, resulting in poor forecasting.

In real-world scenarios, time series data are often contaminated with anomalies and noise. It remains unclear how these methods perform under such conditions. Fig. 1 illustrates a forecasting example on a segment of the ETTh2 (Zhou et al., 2021) dataset using a *single-layer linear model*. Under the original condition, without any perturbation (Fig. 1a), the model generates accurate forecasts. However, when an out-of-distribution perturbation is applied to the last point in the input window (the blue point in Fig. 1b), the predicted series exhibits significant errors in both its overall level and periodic pattern, leading to a significant degradation in forecasting performance.

Motivated by this observation, we investigate the robustness of time series forecasting models to **point-wise perturbations** in the historical sequence, where such perturbations may manifest as

either significant out-of-distribution outliers or minor in-distribution noise. To this end, we first conduct an empirical study to examine the robustness of existing approaches against perturbations, with a particular focus on popular deep learning-based methods (see Sec. 2.3). To systematically assess this issue, we employ a gradient-based input importance scoring method to quantify the influence of each input point. Our findings reveal that these methods tend to rely excessively on specific input points, especially the most recent observation, rendering them highly susceptible to point-wise perturbations and compromising prediction reliability. This insight motivates our work on developing more robust time series forecasting models.

Existing robust TSF approaches commonly utilize seasonal-trend decomposition (Wen et al., 2019; Oreshkin et al., 2020; Zhou et al., 2022b; Zeng et al., 2023; He et al., 2023). However, these methods do not fully address point-wise perturbations, as such perturbations can still persist in the seasonal component, the trend component, or both. Furthermore, some studies (Liu et al., 2022; 2023b) focus on enhancing robustness in non-stationary time series, which is distinct from our focus on point-wise perturbations. Approaches (Connor et al., 1994; Bohlke-Schneider et al., 2020; Cheng et al., 2024) centered on time series forecasting with anomalies (TSFA) are relevant to robustness against such perturbations. Classic TSFA strategies generally follow a detection-imputation-retraining pipeline (Connor et al., 1994). Nevertheless, the effectiveness of this paradigm is constrained by the cumulative inaccuracies introduced at each step (detection, imputation, and retraining), ultimately limiting both robustness and forecasting accuracy. Overall, existing approaches for time series forecasting commonly either lack robustness against point-wise perturbations or compromise prediction accuracy. Achieving enhanced robustness to such perturbations while maintaining forecasting accuracy remains a significant challenge.

In this paper, we aim to develop a robust TSF method that is resistant to point-wise perturbations while preserving overall forecasting accuracy. The key idea is to *comprehensively exploit the global context* rather than relying heavily on few specific time points. Specifically, we propose to randomly sample time points from the historical input sequence during model training. This strategy compels the model to learn predictive patterns from arbitrary, potentially non-contiguous combinations of sampled time points, thereby enhancing the utilization of global context. However, developing such a sampling-based approach for robust and accurate forecasting faces two challenges:

**Challenge 1: Irregular representation.** Random sampling makes each element in time series correspond to irregular time intervals, so treating the sampled series as a vector misrepresents temporal structure. Imputation or padding introduces artificial values, distorting the original data distribution.

**Challenge 2: Information loss.** Random sampling can inadvertently remove key temporal information that is crucial for accurate forecasting. Important patterns or dependencies may be omitted if the sampled points fail to capture critical segments of the sequence, leading to incomplete context for the model and ultimately degrading forecasting accuracy.

To address Challenge 1, we propose **B**asis-**A**ligned **R**andomized **S**ampling (**BARS**), which represents the randomly sampled time series in a predetermined basis function domain. Unlike direct vectorization or imputation, BARS projects any subset of sampled time points onto a fixed set of trigonometric basis functions spanning various frequencies. Regardless of which time points are sampled, the resulting representation always lies in the same basis domain with a fixed dimensionality. As a result, BARS provides a unified and consistent representation for irregularly sampled sequences, eliminating the need for error-prone imputation of unsampled points in the time domain.

To address Challenge 2, recognizing that periodicity is crucial for accurate time series forecasting (Wang et al., 2024; Wu et al., 2023; Lin et al., 2024), we design a **L**earnable **P**eriodicity **E**xtraction (**LPE**) module. This module extracts the periodic component from historical data before sampling, ensuring that the underlying periodic structure is preserved. Unlike classic trend-periodicity decompostion methods like moving average (Li et al., 2023; Lin et al., 2024) and seaonal-trend decomposition (Cleveland et al., 1990; He et al., 2023), the LPE module learns periodic term from long-term, multi-cycle data, which is inherently robust to point-wise perturbations and does not require sampling. Therefore, sampling is necessary only in the residual space, which has been decycled. The ultimate forecast combines periodicity forecasting through LPE and decycled residual forecasting using BARS. To improve the accuracy and stability of both periodicity and residual learning, we future propose a two-stage training protocol.

Ultimately, we propose **RESAM**, a robust time series forecasting approach via **SAM**pling in de-cycled **RES**idual space, which integrates basis-aligned randomized sampling, learnable periodicity extraction, and two-stage training. The contributions of this paper are summarized as follows:

- To the best of our knowledge, we are the first to identify and systematically validate the vulnerability of mainstream time series forecasting models to point-wise perturbations, particularly when applied to the last point within historical input sequence.
- We introduce RESAM, a time series forecasting method which is robust to point-wise perturbations. RESAM integrates BARS to achieve robust and consistent representation of sampled time series, as well as learnable periodicity extraction to preserve essential periodic patterns. Through two-stage training, RESAM attains accuracy and stability in both periodic and residual learning.
- Extensive experiments on eight datasets spanning five domains demonstrate that the proposed RESAM can effectively capture global historical context, thereby enhancing robustness to point-wise perturbations while maintaining forecasting accuracy.

## 2 PRELIMINARY

### 2.1 MULTIVARIATE TIME SERIES FORECASTING

Given a historical time series of $H$ time points $\mathbf{X} = \{\mathbf{x}_{t-H+1}, \mathbf{x}_{t-H+2}, \ldots, \mathbf{x}_t\} \in \mathbb{R}^{H \times C}$, where $\mathbf{x}_t = [x_t^{(1)}, x_t^{(2)}, \ldots, x_t^{(C)}]^\top$ is the observation vector at time $t$, and $C$ is the number of variables (or channels), the forecasting task aims to predict the future $F$-step time series $\hat{\mathbf{Y}} = \{\hat{\mathbf{x}}_{t+1}, \hat{\mathbf{x}}_{t+2}, \ldots, \hat{\mathbf{x}}_{t+F}\} \in \mathbb{R}^{F \times C}$.

### 2.2 MOTIVATION

For the time series forecasting task, a simple deep learning forecaster is a *single-layer linear model* with input dimension matching the historical sequence length ($H$) and output dimension corresponding to the prediction length ($F$). In this architecture, the linear model's weight matrix possesses dimensions of $H \times F$.

We train this model on the ETTh2 (Zhou et al., 2021) dataset and analyze a case of poor prediction performance. As shown in Fig. 2a, when anomalies occur in the final few points of the input sequence, the model's predictions deviate significantly from the ground truth. Furthermore, we visualize the normalized weight matrix of the trained model in Fig. 2b, where the absolute values indicate the relative contributions to the forecast. The heatmap reveals two main patterns. First, there are **diagonal banding patterns** that correspond to the intrinsic periodicity of the dataset (24 steps for ETTh2). Second, **vertical high-weight bands** are con-

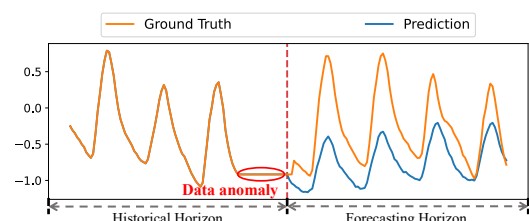

(a) A bad prediction case by a single-layer linear model.

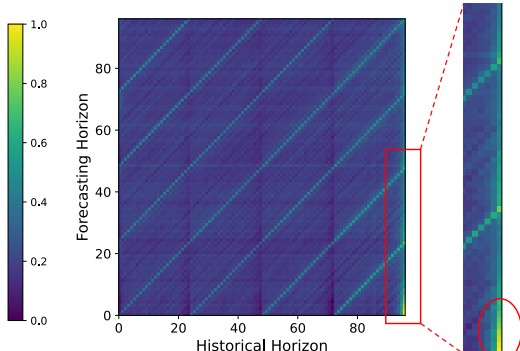

(b) Weight matrix of the trained single-layer linear model.

Figure 2: A single-layer linear model trained on ETTh2 shows bad predictions when anomalies occur in the last few historical points (Fig. 2a). The normalized weight matrix in Fig. 2b shows high weights in the right columns, indicating these points strongly affect forecasting.

centrated in the rightmost columns, indicating that the final few input points—especially the last one—exert a greater influence on future predictions than other points. Consequently, when anomalies occur in the last few points of the context window, the model is prone to make bad predictions.

### 2.3 EMPIRICAL STUDY

For the single-linear model, the last few input points heavily impact the prediction. Driven by this observation, we explore whether more complex forecasting models exhibit similar dependencies. Unlike the single-linear model which can offer directly an interpretable weight matrix, inspired by Grad-CAM (Selvaraju et al., 2017) and Assaf & Schumann (2019), we propose to utilize a gradient-

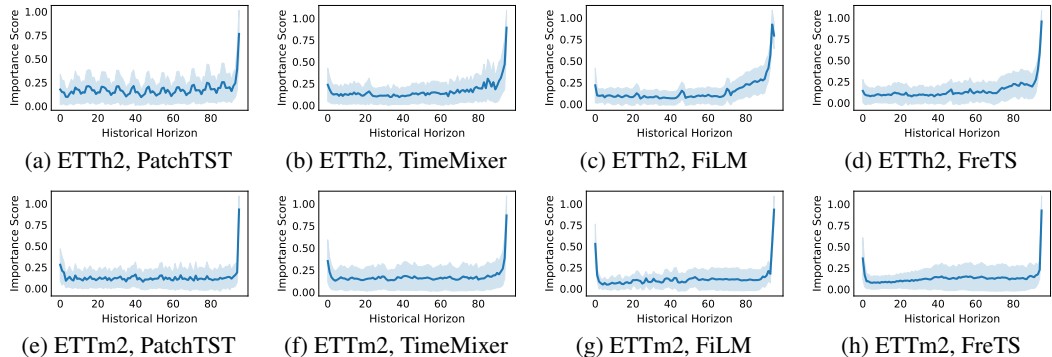

Figure 3: Importance score distribution of historical time points on ETTh2 and ETTm2 across different model architectures. The central line represents mean of importance scores ($\mu(t)$), and the shaded confidence band refers to the 95% confidence interval ($\text{CI}_{95\%}(t)$).

based point importance scoring method for complex forecasting models. More details of the method are provided in Appendix C.

We evaluate the importance score distribution of four representative model architectures (see Appendix A) on ETTh2 (Zhou et al., 2021) and ETTm2 (Zhou et al., 2021), containing PatchTST (Nie et al., 2023) (attention-based, time domain), TimeMixer (Wang et al., 2024) (linear-based, time domain), FiLM (Zhou et al., 2022a) (attention-based, Legendre-Fourier domain), and FreTS (Yi et al., 2023) (linear-based, frequency domain). Utilizing these models, we aim to encompass the range of prevalent design paradigms in contemporary forecasting architectures, incorporating both linear and attention-based models that function across time, frequency, and mathematical transform domains.

Fig. 3 presents the resulting importance score distribution. Across all models and datasets, we consistently observe that importance scores progressively rise for the final few historical sequence points, with the highest scores typically assigned to the last point. This pattern suggests that current architectures are not effectively leveraging the global context, instead showing a significant dependence on local, recent time points, regardless of any specific input time series. Although recent time points are theoretically beneficial for maintaining continuity and accuracy in forecasting, assigning excessive importance to these points may reduce the effectiveness of utilizing the broader context. This limitation reduces model robustness against point-wise perturbations, especially when anomalies impact critical endpoints. To enhance robustness against such perturbations, it is crucial to prioritize the global context over specific time points, thereby ensuring a relatively balanced importance score distribution across various historical time points.

## 3 METHODOLOGY

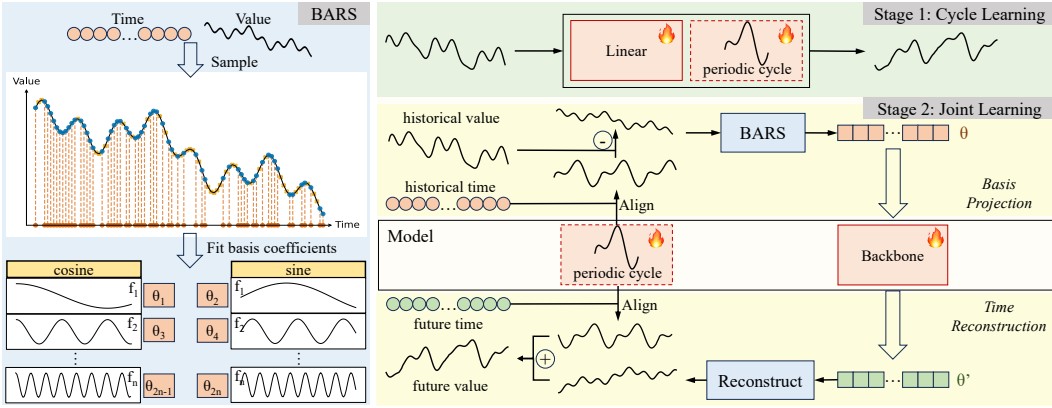

Figure 4: The overall RESAM forecasting approach (Sec. 3.3). The left part is the BARS strategy, which performs random time point sampling and transforms sampled time series into basis domain. The right part is the training and forecasting workflow incorporating BARS (Sec. 3.1), Learnable Periodicity Extraction (Sec. 3.2) and 2-Stage Training (Sec. 3.4).

Based on the idea of "comprehensively exploiting the global context" and "being robust against point-wise perturbations", we propose RESAM (Fig. 4), a robust time series forecasting approach via sampling in decycled residual space. Firstly, we introduce BARS to dynamically utilize the entire context window. Next, we present LPE to facilitate the robust decomposition of periodicity and residuals. On this basis, we design a two-stage training technique to enhance the accuracy and stability of dual-space learning (periodic and residual space).

## 3.1 BARS

To dynamically leverage the entire context window, we propose Basis-Aligned Randomized Sampling (**BARS**). The core insight is to stochastically sample time points within the historical sequence during training, compelling the model to learn predictions based on randomly selected points. Consequently, during inference, the trained model can effectively utilize all available time points without relying heavily on specific points. The complete process of BARS is detailed in Appendix D.1.

As illustrated in the left panel of Fig. 4, BARS operates through random times point sampling governed by a hyperparameter sampling rate $\rho$. Given a historical time series $\mathbf{X} \in \mathbb{R}^{H \times C}$ spanning $H$ time points with $C$ channels, the sampled subsequence contains $\lfloor \rho H \rfloor$ points while preserving the channel dimension. During each training iteration, the random omission of temporal points prevents the model from over-relying on any particular points.

Traditional techniques (Connor et al., 1994) for processing unsampled time points typically involve imputation in the time domain, such as zero-padding, forward/backward filling, or linear interpolation. However, these imputation techniques can introduce artificial biases, thereby distorting predictions. A well-established mathematical principle (Itō, 1993) demonstrates that any continuous function within a function space can be represented as a linear combination of basis functions, similar to vector decomposition in linear algebra. DAM (Darlow et al., 2024) leverages this principle by transforming time series into the basis function domain to support any forecasting horizon. Inspired by this principle and DAM, we treat time series as a continuous function which maps temporal indices to observed values. Through function fitting, we project irregularly sampled time series onto a meticulously constructed basis function domain to avoid the need for imputation in the time domain.

Specifically, considering the inherent characteristics of time series, we select Fourier trigonometric base functions for their ability to capture temporal patterns. we define a set of 179 frequencies $\{f_i\}_{i=1}^{179}$ corresponding to minutely, hourly, daily, and weekly periodicity. Further information regarding the chosen frequencies is available in Appendix D.1. Each frequency contributes two orthogonal basis functions (sine and cosine), yielding 364 basis functions that capture both short-term and long-term temporal dynamics. For a given set of sampled indices and their corresponding observed values, we solve for the basis coefficients $\theta$ via regularized least-squares:

$$\min_{\theta} \|\Phi\theta - \mathbf{X}_{\text{sampled}}\|_2^2 + \lambda\|\theta\|_2^2$$

where $\Phi$ denotes the basis matrix evaluated at the sampled time indices $\tau$, and $\lambda$ is a regularization parameter that ensures numerical stability by mitigating ill-conditioned systems. This basis transformation offers three advantages: (1) avoids problematic time-domain imputation artifacts, (2) ensures a unified functional representation regardless of sampling positions, and (3) projects sampled sequences with varying lengths into a fixed-dimensional coefficient space.

## 3.2 LEARNABLE PERIODICITY EXTRACTION

Random sampling can compromise key temporal information, leading to reduced forecasting accuracy. Recent studies (Wang et al., 2024; Wu et al., 2023; Lin et al., 2024) have thoroughly investigated periodicity modeling in time series forecasting, underscoring its crucial importance in enhancing forecasting accuracy. In response, we propose to extract periodicity from raw time series data before sampling, thus avoiding information loss in periodicity modeling.

Classic moving average (MOV) uses sliding windows for trend-periodicity decomposition, which suffer from two issues (Li et al., 2023; Lin et al., 2024): (1) Padding causes edge distortion, leading to forecasting artifacts. (2) It struggles to separate periodic patterns from perturbations, compromising robustness.

To ensure the robustness of periodicity extraction against perturbations, we propose a learnable periodicity extraction (LPE) module inspired by CycleNet (Lin et al., 2024). Specifically, for each

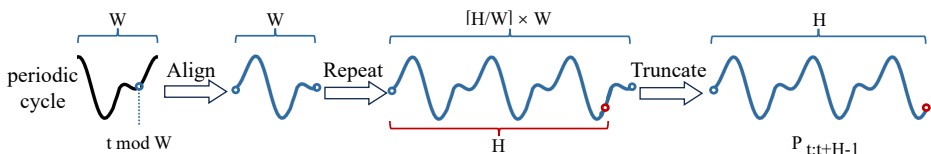

Figure 5: Learnable Periodicity Extraction. It extracts the periodic component with timestamp $t$ and length $H$ based on the periodic cycle.

channel in a dataset, we initialize a learnable periodic cycle whose length $W$ corresponds to the dataset's inherent periodicity (*e.g.*, 24 hours for daily cycles). Fig. 5 illustrates the workflow of the LPE module, which consists of three distinct processes. Firstly, the alignment process adjusts the cycle's starting point to correspond with the input timestamp. Secondly, the repetition process replicates the cycle to ensure it spans the entire historical and forecasting horizon. Lastly, the truncation process removes any cycle portions that exceed the necessary length.

### 3.3 OVERALL FORECASTING APPROACH

By combining the LPE and the BARS module, We present a unified forecasting approach called RESAM, which is illustrated in Fig. 4. The LPE module generates the future periodic component, while BARS operates in the residual space. Consequently, RESAM executes time series forecasting through a dual-space learning mechanism.

Given an historical input sequence $\mathbf{X} \in \mathbb{R}^{H \times C}$, historical timestamps $\mathbf{T}_{\text{hist}}$ and future timestamps $\mathbf{T}_{\text{future}}$, the LPE module first generates corresponding periodic components:

$$\mathbf{P}_{\text{hist}} = \text{LPE}(\mathbf{X}, \mathbf{T}_{\text{hist}}), \ \mathbf{P}_{\text{future}} = \text{LPE}(\mathbf{X}, \mathbf{T}_{\text{future}})$$

The historical residual component is then computed as $\mathbf{R}_{\text{hist}} = \mathbf{X} - \mathbf{P}_{\text{hist}}$. This residual sequence serves as input to the BARS module, and is transformed to a basis coefficient representation, *i.e.*, $\theta_{\text{hist}} = \text{BARS}(\mathbf{R}_{\text{hist}}, \rho, \lambda)$, where $\theta_{\text{hist}} \in \mathbb{R}^{2K \times C}$ represents the basis coefficients capturing the non-periodic residual patterns. Here, random sampling is utilized during the training phase, whereas time points are not sampled during the inference phase for a global temporal information.

A deep learning backbone model $\mathcal{M}$ (*e.g.*, Transformer, CNN, or MLP) then processes these coefficients to predict future residual coefficients, *i.e.*, $\hat{\theta}_{\text{future}} = \mathcal{M}(\theta_{\text{hist}})$. Here, each channel utilizes a shared backbone for modeling through parameter sharing, also known as the Channel Independent strategy (Han et al., 2024). The future residual sequence is reconstructed using basis function, *i.e.*, $\hat{\mathbf{R}}_{\text{future}} = \Phi_{\text{future}} \hat{\theta}_{\text{future}}$, where $\Phi_{\text{future}}$ is the basis matrix evaluated at future timestamps. The final forecast combines the predicted residuals with the periodic component ($\hat{\mathbf{Y}} = \hat{\mathbf{R}}_{\text{future}} + \mathbf{P}_{\text{future}}$).

Note that we deliberately avoid directly reconstructing future residuals from $\theta_{\text{hist}}$ for two primary reasons. Firstly, the basis coefficients $\theta_{\text{hist}}$ are tailored to fit the historical residuals uniquely, thereby leading to overfitting to past data patterns. They lack the capability to generalize to future series. Secondly, real-world time series exhibit fluctuations that do not adhere to strict mathematical function relationships. The intrinsic non-functional characteristics of time series data hinder the straightforward extrapolation of coefficients into future values. Instead, the backbone model $\mathcal{M}$ is designed to learn the transition dynamics between historical and future coefficient spaces, effectively capturing the evolving residual patterns.

### 3.4 2-STAGE TRAINING

RESAM utilizes a dual-space approach for forecasting, wherein the LPE captures periodicity within the time domain and BARS addresses residuals in the basis function domain. The incorporation of learnable periodic cycle, randomized sampling and basis function transformation introduces additional complexity to the model learning process. Conducting direct end-to-end training on the entire architecture may result in suboptimal periodic and residual patterns that do not accurately reflect the data's intrinsic characteristics. To ensure the accuracy and stability of dual-space learning, we develop a meticulously crafted two-stage training protocol. The entire two-stage training process is illustrated in Appendix D.2.

**Stage 1: Periodic Cycle Learning.** As illustrated in the top-right panel of Fig. 4, the initial stage is designed to learn accurate periodic cycles using a simple architecture. We replace the BARS module in RESAM with a single-layer linear model that maps historical residuals directly to future residuals, bypassing complex basis transformations or sampling operations. The future periodic component keeps to be predicted using the LPE module. The straightforward architecture of this stage, operating directly in the time domain without intricate transformations, forces the LPE module to accurately learn periodic cycles.

**Stage 2: Joint space Learning.** As depicted in the lower-right panel of Fig. 4, this stage incorporates the periodic cycles learned in Stage 1 into RESAM by initializing the LPE module using the learned cycle parameters. Then, we train RESAM which contains the LPE module for periodicity space learning and the BARS module for residual space learning. All parameters, including the periodic cycle, are jointly optimized during this stage, enabling fine-tuning of the periodic cycle to complement the basis domain.

**Loss Function.** Drawing inspiration from RobustTSF (Cheng et al., 2024), which demonstrated that Mean Absolute Error (MAE) offers improved robustness against anomalies over Mean Squared Error (MSE), we employ MAE loss across both stages.

## 4 EVALUATION

We comprehensively evaluate the performance of RESAM in terms of overall accuracy and robustness, using 8 widely-used dataset across 5 key domains. We compare RESAM to baselines across various model architectures and representation domains. We use a *dual-layer MLP* with ReLU activation as the backbone, $\mathcal{M}$. Comprehensive datasets, baselines, and implementation details are provided in Appendix B. Due to space limitations, the ablation studies, the analysis of various parameters and the efficiency analysis are presented in Appendix E, F, and H respectively.

**Evaluation Metrics.** We evaluate forecasting accuracy using two well-established metrics: Mean Squared Error (MSE) and Mean Absolute Error (MAE). Both metrics are calculated on normalized datasets, with lower values indicating better performance. For perturbation robustness evaluation, we implement two point-wise perturbation types on historical sequences: last-point (modifying the final time point $x_T^{\text{pert}} = x_T + \delta$) and random-point (modifying a randomly selected time point $x_t^{\text{pert}} = x_t + \delta$). The perturbation magnitude $\delta$ is sampled from $\mathcal{N}(0, (\alpha \cdot \sigma_{\text{hist}})^2)$, where $\sigma_{\text{hist}}$ is the standard deviation of input historical time series. $\alpha$ quantifies the extent of perturbations, with $\alpha = 3$ for out-of-distribution perturbation. We evaluate the impact of perturbations by reporting post-perturbation MSE and MAE.

### 4.1 OVERALL FORECASTING PERFORMANCE

Table 1: Results of multivariate time series forecasting task. The results are averaged from all prediction lengths of $F \in \{96, 192, 336, 720\}$. **1ˢᵗ Count** is the total number of times a model achieved the best performance of two metrics across all datasets and prediction lengths. **Avg Rank** is a model's average rank of two metrics across all datasets and prediction lengths. **Bold** indicates best result, underlined second best per dataset and metric.

| Models | ETTh1 | | ETTh2 | | ETTm1 | | ETTm2 | | Weather | | Exchange | | Traffic | | Solar | | 1ˢᵗ Count | Avg Rank |
|---|---|---|---|---|---|---|---|---|---|---|---|---|---|---|---|---|---|---|
| | MSE | MAE | MSE | MAE | MSE | MAE | MSE | MAE | MSE | MAE | MSE | MAE | MSE | MAE | MSE | MAE | | |
| DLinear | 0.443 | 0.433 | 0.452 | 0.446 | 0.396 | 0.387 | 0.286 | 0.330 | 0.271 | 0.290 | 0.318 | 0.387 | 0.665 | 0.352 | 0.328 | 0.308 | 0 | 4.62 |
| TimeMixer | 0.461 | 0.439 | 0.374 | 0.395 | 0.398 | 0.388 | 0.278 | 0.319 | 0.254 | 0.270 | 0.381 | 0.414 | 0.495 | 0.304 | 0.334 | 0.321 | 0 | 3.59 |
| CycleNet | **0.429** | **0.418** | 0.375 | 0.394 | 0.386 | 0.383 | 0.273 | 0.312 | 0.258 | 0.276 | 0.390 | 0.419 | 0.489 | 0.292 | 0.292 | 0.266 | 6 | 2.52 |
| Crossformer | 0.518 | 0.498 | 0.976 | 0.688 | 0.484 | 0.445 | 0.554 | 0.486 | 0.247 | 0.275 | 0.606 | 0.585 | 0.593 | 0.300 | 0.226 | **0.218** | 8 | 5.54 |
| PatchTST | 0.431 | 0.424 | **0.367** | **0.389** | 0.383 | 0.383 | 0.278 | 0.318 | 0.256 | 0.272 | 0.373 | 0.409 | 0.495 | 0.292 | 0.272 | 0.305 | 5 | 2.40 |
| MICN | 0.435 | 0.441 | 0.463 | 0.450 | 0.381 | 0.388 | 0.285 | 0.337 | 0.248 | 0.274 | **0.309** | **0.379** | 0.619 | 0.311 | 0.283 | 0.279 | 10 | 3.76 |
| TimesNet | 0.490 | 0.464 | 0.417 | 0.419 | 0.406 | 0.403 | 0.289 | 0.325 | 0.256 | 0.278 | 0.430 | 0.444 | 0.679 | 0.348 | 0.364 | 0.336 | 0 | 5.80 |
| FEDformer | 0.439 | 0.446 | 0.417 | 0.431 | 0.569 | 0.520 | 0.341 | 0.383 | 0.306 | 0.339 | 0.529 | 0.497 | 0.649 | 0.376 | 0.548 | 0.549 | 0 | 6.65 |
| FreTS | 0.511 | 0.462 | 0.541 | 0.469 | 0.415 | 0.404 | 0.300 | 0.340 | **0.240** | **0.267** | 0.394 | 0.425 | 0.519 | 0.304 | 0.263 | 0.261 | 2 | 4.59 |
| FiLM | 0.444 | 0.427 | 0.382 | 0.400 | 0.401 | 0.387 | 0.286 | 0.322 | 0.282 | 0.287 | 0.376 | 0.417 | 1.233 | 0.687 | 0.377 | 0.362 | 0 | 5.11 |
| iTransformer | 0.448 | 0.439 | 0.381 | 0.401 | 0.394 | 0.389 | 0.281 | 0.321 | 0.254 | 0.271 | 0.365 | 0.406 | **0.450** | 0.282 | 0.249 | 0.245 | 6 | 3.08 |
| TimeKAN | 0.445 | 0.427 | 0.378 | 0.397 | 0.388 | 0.384 | 0.281 | 0.321 | 0.249 | 0.269 | 0.394 | 0.422 | 0.633 | 0.349 | 0.321 | 0.291 | 1 | 3.66 |
| Leddam | 0.437 | 0.425 | 0.371 | 0.391 | 0.390 | 0.383 | 0.277 | 0.317 | 0.243 | 0.272 | 0.362 | 0.403 | 0.491 | **0.278** | **0.220** | 0.256 | 7 | 2.07 |
| RobustTSF | 0.446 | 0.438 | 0.435 | 0.438 | 0.395 | 0.390 | 0.284 | 0.332 | 0.270 | 0.293 | 0.374 | 0.396 | 0.639 | 0.359 | 0.342 | 0.343 | 3 | 4.80 |
| RESAM | 0.432 | 0.423 | 0.369 | 0.392 | **0.379** | **0.379** | **0.267** | **0.309** | 0.248 | 0.269 | 0.358 | 0.403 | 0.537 | 0.308 | 0.243 | 0.244 | **16** | **1.82** |

Tbl. 1 illustrates the multivariate forecasting performance of RESAM, revealing its competitive accuracy, comparable to state-of-the-art models. RESAM achieves the highest average rank and the most 1st Count across all datasets and prediction lengths. Due to its channel-independent mechanism, RESAM performs slightly worse on datasets with greater correlation, such as Solar and Traffic, compared to lower-dimensional datasets like ETTh1, ETTh2, ETTm1, ETTm2. The forecasting accuracy of RESAM on Exchange is worse than MICN and DLinear. As illustrated by Tbl. 4 in Appendix B.1, Exchange exhibits a pronounced shifting characteristic. Since RESAM relies partially on reconstructing basis functions for predictions, its effectiveness decreases when the time series demonstrates significant shifts. Notably, RESAM, utilizing only a dual-layer MLP backbone, consistently outperforms more structurally complex models, including TimeNet and TimeMixer, across most datasets. This demonstrates its effectiveness in capturing temporal patterns for forecasting.

## 4.2 ROBUSTNESS

Table 2: Robustness evaluation under last-point perturbation ratio $\alpha = 3.0$. Results are averaged from prediction lengths $F \in \{96, 192, 336, 720\}$. Post-perturbation MSE and MAE are reported.

| Models | ETTh1 | | ETTh2 | | ETTm1 | | ETTm2 | | Weather | | Exchange | | Traffic | | Solar | | 1st Count | Avg Rank |
|---|---|---|---|---|---|---|---|---|---|---|---|---|---|---|---|---|---|---|
| | MSE | MAE | MSE | MAE | MSE | MAE | MSE | MAE | MSE | MAE | MSE | MAE | MSE | MAE | MSE | MAE | | |
| DLinear | 0.566 | 0.493 | 0.448 | 0.451 | 0.623 | 0.477 | 0.331 | 0.366 | 0.371 | 0.345 | 0.373 | 0.434 | 0.902 | 0.422 | 3.175 | 0.851 | 5 | 5.38 |
| TimeMixer | 0.552 | 0.486 | 0.407 | 0.419 | 0.534 | 0.452 | 0.318 | 0.352 | 0.298 | 0.311 | 0.439 | 0.465 | 0.568 | 0.346 | 0.569 | 0.449 | 0 | 3.46 |
| CycleNet | 0.565 | 0.483 | 0.434 | 0.436 | 0.619 | 0.479 | 0.320 | 0.353 | 0.349 | 0.331 | 0.429 | 0.452 | 0.588 | 0.340 | 0.664 | 0.405 | 0 | 4.25 |
| Crossformer | 0.524 | 0.503 | 0.980 | 0.690 | 0.514 | 0.467 | 0.571 | 0.496 | 0.274 | 0.303 | 0.612 | 0.591 | 0.593 | 0.289 | 0.327 | 0.288 | 8 | 4.13 |
| PatchTST | 0.554 | 0.486 | 0.400 | 0.415 | 0.645 | 0.491 | 0.346 | 0.369 | 0.306 | 0.316 | 0.468 | 0.485 | 0.601 | 0.353 | 0.725 | 0.507 | 0 | 4.39 |
| MICN | 0.474 | 0.467 | 0.474 | 0.459 | 0.484 | 0.441 | 0.309 | 0.360 | 0.295 | 0.313 | 0.325 | 0.398 | 0.623 | 0.318 | 0.359 | 0.345 | 6 | 2.51 |
| TimesNet | 0.472 | 0.453 | 0.396 | 0.409 | 0.418 | 0.411 | 0.296 | 0.331 | 0.262 | 0.284 | 0.435 | 0.448 | 0.700 | 0.360 | 0.382 | 0.354 | 9 | 2.20 |
| FEDformer | 0.467 | 0.459 | 0.408 | 0.426 | 0.789 | 0.591 | 0.348 | 0.388 | 0.314 | 0.346 | 0.530 | 0.497 | 0.666 | 0.389 | 0.657 | 0.607 | 3 | 4.97 |
| FreTS | 0.677 | 0.532 | 0.576 | 0.492 | 0.745 | 0.519 | 0.404 | 0.404 | 0.310 | 0.336 | 0.423 | 0.456 | 0.792 | 0.405 | 1.369 | 0.586 | 0 | 5.93 |
| FiLM | 0.492 | 0.453 | 0.389 | 0.406 | 0.511 | 0.438 | 0.315 | 0.348 | 0.303 | 0.305 | 0.380 | 0.421 | 1.281 | 0.708 | 1.173 | 0.629 | 0 | 3.24 |
| iTransformer | 0.500 | 0.470 | 0.404 | 0.418 | 0.575 | 0.475 | 0.327 | 0.359 | 0.317 | 0.322 | 0.410 | 0.445 | 0.583 | 0.365 | 0.754 | 0.470 | 1 | 3.68 |
| TimeKAN | 0.501 | 0.461 | 0.404 | 0.418 | 0.521 | 0.450 | 0.323 | 0.357 | 0.286 | 0.302 | 0.451 | 0.467 | 0.678 | 0.379 | 0.803 | 0.482 | 1 | 3.48 |
| Leddam | 0.505 | 0.467 | 0.405 | 0.418 | 0.585 | 0.471 | 0.350 | 0.372 | 0.340 | 0.347 | 0.441 | 0.461 | 0.625 | 0.351 | 2.340 | 0.761 | 0 | 4.70 |
| RobustTSF | 0.608 | 0.510 | 0.503 | 0.483 | 0.739 | 0.509 | 0.362 | 0.389 | 0.471 | 0.382 | 0.411 | 0.435 | 0.926 | 0.450 | 2.071 | 0.761 | 0 | 6.36 |
| RESAM | 0.461 | 0.442 | 0.379 | 0.401 | 0.416 | 0.404 | 0.278 | 0.319 | 0.263 | 0.286 | 0.421 | 0.460 | 0.550 | 0.319 | 0.340 | 0.308 | 31 | 1.33 |

Tbl. 2 presents experimental results of perturbations applied to the last point of the historical input window, with a perturbation ratio of $\alpha = 3.0$. RESAM demonstrates a substantial performance advantage, achieving an average rank of 1.33 and a 1st Count of 32 across all datasets and prediction lengths, in comparison to scenarios without perturbations shown in Tbl. 1. RESAM consistently achieves either the best or second-best robustness performance across all datasets, except for the Exchange dataset, highlighting its resilience against point-wise perturbations. The Exchange dataset, characterized by higher distribution shifts and significant non-stationarity, poses a challenge for RESAM, which partially relies on basis function reconstruction, to maintain robustness.

To quantify the proportional increase in MSE after perturbations, we introduce $MSE \Uparrow$ which is calculated by $\frac{MSE_{pert} - MSE_{clean}}{MSE_{clean}} \times 100\%$, where $MSE_{clean}$ and $MSE_{pert}$ represent MSE calculated on clean and perturbed data respectively. A higher $MSE \Uparrow$ indicates a greater decline in forecasting accuracy. Tbl. 3 shows forecasting $MSE \Uparrow$ under last-point perturbations with ratio $\alpha$ as 3.0. Our findings highlight unique robustness patterns associated with various *model architectures* and *representation domains*. **Findings:**

1. *Linear-based* models such as DLinear, TimeMixer, and CycleNet exhibit heightened vulnerability to point-wise perturbations, primarily due to their strong dependency on recent observations.
2. *Attention-based* models, such as PatchTST, iTransformer and Leddam, remain susceptible to point-wise perturbations despite their intricate structures with numerous parameters. Crossformer exhibits a slight increase in MSE after perturbations. However, as demonstrated in Tbl. 2, its absolute forecasting MSE after such perturbations exceeds that of RESAM.
3. *Convolution-based* models exhibit superior robustness, with TimesNet experiencing notably minimal performance degradation under perturbed scenarios. Yet, TimesNet's absolute forecasting accuracy after perturbations is inferior to RESAM across most datasets (see Tbl. 2).
4. *Frequency-domain* models such as FreTS and *mathematical-transform-domain* models like FiLM experience significant performance deterioration under perturbations. Although FED-

Table 3: Robustness evaluation under last-point perturbation with ratio $\alpha = 3.0$. $MSE \Uparrow$ (proportional rise in MSE) is reported.

| Architecture | Models | ETTh1 | ETTh2 | ETTm1 | ETTm2 | Weather | Exchange | Traffic | Solar |
|---|---|---|---|---|---|---|---|---|---|
| Linear | DLinear | 29.70 | 2.69 | 61.74 | 17.61 | 41.75 | 55.38 | 35.58 | 897.13 |
| | TimeMixer | 21.63 | 10.22 | 36.30 | 16.46 | 22.36 | 40.88 | 14.94 | 72.92 |
| | CycleNet | 33.18 | 17.19 | 65.02 | 21.50 | 42.07 | 17.26 | 20.26 | 134.26 |
| Attention | Crossformer | 1.11 | 0.51 | 8.02 | 5.45 | 15.89 | 2.90 | **0.34** | 44.94 |
| | PatchTST | 29.82 | 9.88 | 75.53 | 28.51 | 21.98 | 56.85 | 21.79 | 181.09 |
| | iTransformer | 12.19 | 6.70 | 49.88 | 19.96 | 29.45 | 25.55 | 30.23 | 217.50 |
| | Leddam | 16.45 | 10.13 | 53.08 | 30.61 | 47.12 | 38.25 | 28.05 | 973.75 |
| Convolution | MICN | 9.46 | 3.14 | 30.04 | 11.13 | 21.89 | 10.36 | 0.67 | 30.63 |
| | TimesNet | **-3.11** | **-4.74** | **3.22** | **2.85** | **2.67** | 2.14 | 3.12 | **4.93** |
| Frequency | FEDformer | 5.97 | -1.81 | 39.51 | 2.94 | 3.44 | **0.21** | 2.47 | 27.72 |
| | FreTS | 37.17 | 10.85 | 84.59 | 45.68 | 34.17 | 22.38 | 53.59 | 434.65 |
| | TimeKAN | 13.22 | 7.44 | 37.60 | 17.71 | 16.65 | 26.49 | 7.03 | 162.94 |
| Mathematics | FiLM | 11.48 | 1.74 | 30.75 | 12.88 | 8.87 | 0.54 | 4.71 | 245.92 |
| TSFA | RobustTSF | 38.26 | 19.08 | 93.11 | 33.76 | 82.72 | 26.92 | 44.92 | 521.72 |
| - | RESAM | 7.07 | 3.01 | 10.33 | 4.94 | 7.16 | 44.50 | 2.37 | 43.45 |

former shows minor performance degradation, its overall forecasting accuracy is limited due to insufficient attention to recent temporal patterns, thus restricting its practical applicability.

5. RobustTSF, designed for time series forecasting with anomalies, exhibits significant performance degradation when faced with last-point perturbations. This is primarily because RobustTSF is not specifically optimized to address anomalies that occur in the most recent data points within historical windows, and the anomaly detection technique employed may introduce errors during model training.

**Comparison of different perturbation types.** We further compare $MSE \Uparrow$ after last-point perturbations to random-point perturbations. Experimental results on ETTm1 are presented in Fig. 6. All models exhibit a slight performance degradation under random-point perturbations, with $MSE \Uparrow$ significantly lower than those observed under last-point perturbations. This phenomenon indicates that most current forecast-

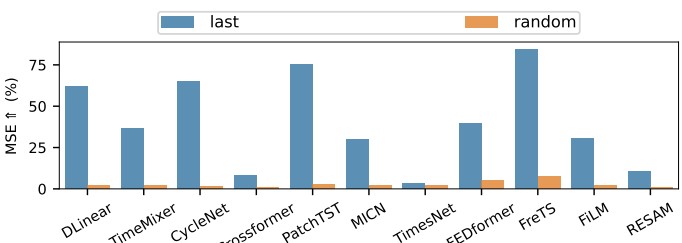

Figure 6: Robustness evaluation ($MSE \Uparrow$) under different perturbation types (last-point & random-point) on ETTm1.

ing models predominantly rely on specific crucial points, particularly the most recent observation in the historical window, rather than uniformly utilizing information from the entire sequence.

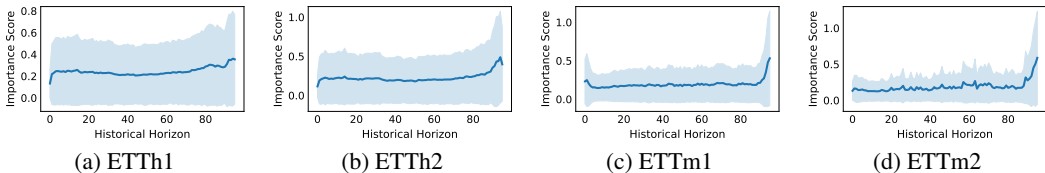

| (a) ETTh1 | (b) ETTh2 | (c) ETTm1 | (d) ETTm2 |
|---|---|---|---|

Figure 7: Importance score distribution of historical time points on four ETT datasets for RESAM.

**Temporal importance of RESAM.** To further analyze the temporal importance (see Sec. 2.3) of RESAM, we present the impact of different time points on forecasting performance, using RESAM with a historical sequence length $H = 96$ and prediction length $F = 96$ across four ETT datasets, as depicted in Fig. 7. Compared to other models (Fig. 3), RESAM exhibits a relatively uniform distribution of importance across all historical points, with marginally higher weights assigned to the

most recent points. This balanced temporal utilization allows for robust predictions that withstand local perturbations while effectively capturing recent trends. Overall, RESAM shows enhanced robustness against point-wise perturbations in time series forecasting.

## 5 RELATED WORK

**Robust time series forecasting** is focused on developing models that can maintain accurate predictions despite the presence of data outliers or inherent variability. Seasonal-trend decomposition techniques are well-established to enhance model architectures against natural fluctuations in time series patterns. Deep learning models (Oreshkin et al., 2020; Zhou et al., 2022b; Zeng et al., 2023) have successfully integrated these decomposition techiniques, whereas some methods (Wen et al., 2019) specifically target robust seasonal-trend decomposition. Additionally, some methods aim to robustly capture correlations among multiple variables for time series forecasting. Graph neural networks (Shang et al., 2021; Yu et al., 2022) are commonly employed to model both explicit and implicit spatial dependencies among variables. DARF (Cheng et al., 2023) employs adversarial domain adaptation to enhance the modeling of cross-variable relationships. Furthermore, a substantial amount of studies, such as Non-stationary Transformer (Liu et al., 2022) and Koopa (Liu et al., 2023b), are dedicated to processing non-stationarity for robust forecasting.

Time series forecasting with anomalies (TSFA) closely resembles the point-wise perturbations we investigate. Classic TSFA methods employs a detection-imputation-retraining pipeline (Connor et al., 1994; Bohlke-Schneider et al., 2020), while RobustTSF (Cheng et al., 2024) provides a comprehensive theoretical analysis of how anomalies affect time series forecasting. Similarly to our work, an existing study (Yoon et al., 2022) examines the robustness to input perturbations by introducing a randomized smoothing technique, primarily from an adversarial perspective over *entire* time series. In contrast, our work focuses on enhancing robustness against point-wise perturbations, which are prevalent in real-world time series due to factors like measurement noise, missing values, or anomalies.

## 6 CONCLUSION

In this paper, we identify and systematically validate the vulnerability of mainstream time series forecasting models to point-wise perturbations, particularly when perturbations occur at the last point within the historical input sequence. To address this issue, we introduce RESAM, a time series forecasting method which is robust against point-wise perturbations. RESAM integrates a basis-aligned randomized sampling (BARS) module, which randomly samples time points during training, thereby ensuring the utilization of global context during inference. Through a learnable periodicity extraction (LPE) module, we prevent periodic components from perturbations and sampling. Overall, RESAM forecasts in a dual space, *i.e.*, periodicity forecasting via the LPE and residual forecasting via BARS. Additionally, it employs a two-stage learning protocol to ensure the accuracy and stability of dual-space learning. Experiments across 8 datasets from 5 key domains validate RESAM's high forecasting accuracy and robustness to point-wise perturbations. Limitations and directions for future research are discussed in Appendix I.

### ETHICS STATEMENT

This study exclusively addresses the scientific problem, thereby eliminating any potential ethical concerns.

### REPRODUCIBILITY STATEMENT

The implementation details, including dataset descriptions, baseline selections, and experimental configuration, are provided in Appendix B. A comprehensive overview of the evaluation metrics can be found in Sec. 4. The methodology for point importance scoring is thoroughly discussed in Appendix C. The source code is included in the supplementary materials and will be made open-source upon acceptance of this paper. Detailed reproduction steps are available in the README.md file within the supplemental materials.

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

## A    TIME SERIES FORECASTING APPROACHES

Traditional statistical methods, such as ARIMA (Box & Pierce, 1970) and ETS (Hyndman et al., 2008), are highly effective in forecasting linear patterns in time series data but struggle with nonlinear dynamics. Machine learning models like XGBoost (Zhang et al., 2021) and Random Forests (Mei et al., 2014) have been extensively used to address nonlinear relationships and complex patterns, yet they require manual feature engineering and model design. Recently, deep learning techniques have emerged as powerful alternatives, employing deep neural networks (DNNs) to automatically extract complex temporal patterns through representation learning. These modern deep learning methods are systematically organized based on two primary aspects: *model architecture* and *representation domain*.

### MODEL ARCHITECTURES

**Attention-based models** utilize self-attention and cross-attention mechanisms to capture long-range dependencies across time points, such as Informer (Zhou et al., 2021) (sparse attention for efficiency), Autoformer (Wu et al., 2021) (decomposition-enhanced attention), Crossformer (Zhang & Yan, 2022) (cross-dimension dependency modeling), iTransformer (Liu et al., 2023a) (inverted architecture), PatchTST (Nie et al., 2023) (patch-based processing), and Leddam (Yu et al., 2024) (learnable decomposition). **Linear-based models** employ efficient architectures with minimal parameters, achieving competitive performance through carefully designed linear operations. Representative models include N-BEATS (Oreshkin et al., 2020) (interpretable basis expansions), DLinear (Zeng et al., 2023) (decomposition + linear projection), TSMixer (Chen et al., 2023) (all-MLP architecture with cross-variate mixing), and TimeMixer (Wang et al., 2024) (Multi-scale decomposition). **Convolution-based models** capture local patterns through specialized convolution operations. Representative models include TCN (Bai et al., 2018) (temporal dilated convolution), MICN (Wang et al., 2023) (multi-scale isometric convolution) and TimesNet (Wu et al., 2023) (2D convolution).

### REPRESENTATION DOMAINS

**Time domain models** process raw time series directly, preserving the original temporal structure. Most deep learning methods (Zhang & Yan, 2022; Nie et al., 2023; Zeng et al., 2023; Wang et al., 2024; Wu et al., 2023), operate primarily in time domain. **Frequency domain models** employ frequency analysis to extract periodic patterns through Fourier or wavelet analysis, such as FEDformer (Zhou et al., 2022b) (frequency attention), FreTS (Yi et al., 2023) (frequency-domain MLPs) and TimeKAN (Huang et al., 2025) (KAN-based frequency decomposition). **Mathematical transform domains** utilize functional projections to represent time series in alternative mathematical spaces, like FiLM (Zhou et al., 2022a) (Legendre polynomial representation) and DAM (Darlow et al., 2024) (trigonometric basis domain).

Despite architectures and representation domains, most models exhibit imbalanced temporal dependency, which rely heavily on recent input points. Therefore, they exhibit limited robustness to point-wise perturbations. RESAM utilizes a random sampling technique in basis domain, enabling balanced context utilization and perturbation robustness.

## B    EXPERIMENTAL DETAILS

### B.1    DATASETS

We conduct comprehensive evaluations using multiple datasets following a popular benchmark, TFB (Qiu et al., 2024), adopting its standardized data partitioning scheme and rolling forecasting protocol. Our selected datasets encompass five key domains to ensure broad coverage of real-world forecasting scenarios: **Electricity** (ETTh1 (Zhou et al., 2021), ETTh2 (Zhou et al., 2021), ETTm1 (Zhou et al., 2021), ETTm2 (Zhou et al., 2021)), **Environment** (Weather (Wu et al., 2021)), **Energy** (Solar (Lai et al., 2018)), **Traffic** (Traffic (Wu et al., 2021)), and **Economic** (Exchange (Lai et al., 2018)). Tbl. 4 provides detailed information of the selected datasets. These datasets exhibit significant diversity across sampling frequencies, channel dimensions, and whole sequence lengths.

Table 4: Statistics of datasets

| Dataset | Domain | Frequency | Lengths | Dim | Split | Shifting | Correlation | Cycle $W$ |
|---|---|---|---|---|---|---|---|---|
| ETTh1 (Zhou et al., 2021) | Electricity | 1 hour | 14,400 | 7 | 6:2:2 | -0.06 | 0.63 | 24 |
| ETTh2 (Zhou et al., 2021) | Electricity | 1 hour | 14,400 | 7 | 6:2:2 | -0.40 | 0.51 | 24 |
| ETTm1 (Zhou et al., 2021) | Electricity | 15 mins | 57,600 | 7 | 6:2:2 | -0.06 | 0.61 | 96 |
| ETTm2 (Zhou et al., 2021) | Electricity | 15 mins | 57,600 | 7 | 6:2:2 | -0.41 | 0.50 | 96 |
| Weather (Wu et al., 2021) | Environment | 10 mins | 52,696 | 21 | 7:1:2 | 0.21 | 0.69 | 144 |
| Exchange (Lai et al., 2018) | Economic | 1 day | 7,588 | 8 | 7:1:2 | 0.33 | 0.57 | 512 |
| Solar (Lai et al., 2018) | Energy | 10 mins | 52,560 | 137 | 6:2:2 | 0.20 | 0.79 | 144 |
| Traffic (Wu et al., 2021) | Traffic | 1 hour | 17,544 | 862 | 7:1:2 | 0.07 | 0.81 | 168 |

In Tbl. 4, we also present the cycle length $W$ configuration for different datasets. For domains with natural periodicity, like electricity or weather, the cycle length $W$ is set to the natural period (*e.g.*, 24 hours for daily cycles). In particular, for domains such as economics or finance that lack clear periodicity, we configure $W \geq H$ to capture patterns across broader contexts.

TFB also defines datasets with distinct attributes: *Shifting* indicates a change in the probability distribution of a time series over time, with higher severity as the shifting value approaches 1. *Correlation* denotes the dependency among different variables in a multivariate time series. We employ TFB's unified evaluation protocol with four prediction lengths $F$ (96, 192, 336, 720). For a fair comparison, the historical sequence length $H$ is uniformly set to 96 for both RESAM and all baseline models.

## B.2  BASELINES

We select representative deep learning models for time series forecasting spanning diverse architectures and operational domains for comprehensive comparison. Approaches operating in the time domain include: linear-based models (DLinear (Zeng et al., 2023), TimeMixer (Wang et al., 2024), CycleNet (Lin et al., 2024)), attention-based models (Crossformer (Zhang & Yan, 2022), PatchTST (Nie et al., 2023), iTransformer (Liu et al., 2023a) and Leddam (Yu et al., 2024)), and convolution-based models (MICN (Wang et al., 2023), TimesNet (Wu et al., 2023)). Complementary approaches operating in different domains include frequency-domain models (FEDformer (Zhou et al., 2022b), FreTS (Yi et al., 2023) and TimeKAN (Huang et al., 2025)), and mathematical-transform-domain models (FiLM (Zhou et al., 2022a)). We also include RobustTSF (Cheng et al., 2024) which aims for time series forecasting with anomalies. Detailed information of these models can be found in Appendix A.

## B.3  IMPLEMENTATION DETAILS

We implement RESAM within both Time Series Library (Wu et al., 2023) and TFB (Qiu et al., 2024), with model and experimental code available in the supplementary materials. All experiments were conducted using PyTorch (Paszke et al., 2019) with the Adam optimizer (Kingma & Ba, 2014), executed on one NVIDIA Tesla V100 GPU with 32GB memory. For RESAM, we employ RevIN (Kim et al., 2021) normalization, a sampling rate $\rho = 0.75$ and a regularization $\lambda = 0.1$ in BARS. For different datasets, we assign distinct cycle lengths $W$ based on their inherent periodicity (see Tbl. 4). Notably, in the case of the Exchange dataset, which lacks an intrinsic period, we set $W = 512$ to represent a prolonged cycle length. We select a *dual-layer MLP* with ReLU activation as the backbone architecture. Hyperparameter searching is performed for hidden size of MLP from 64 to 512, learning rate from 0.0001 to 0.01, batch size from 8 to 64. Baseline models utilize hyperparameters derived from TFB, and all models are trained using MAE loss to ensure fair robustness comparisons. The initial batch size is set according to the default TFB settings. In the event of an Out-Of-Memory (OOM) error, the batch size can be reduced by half, with a minimum size of 4.

## C    POINT IMPORTANCE SCORING

Techniques such as Grad-CAM (Selvaraju et al., 2017) from the computer vision area and methods (Assaf & Schumann, 2019) in the time series analysis area employ gradient-based interpretability strategies. Grad-CAM generates gradient-weighted class activation maps to highlight influential regions in input images, whereas Assaf & Schumann (2019) quantifies feature importance through input gradient magnitudes within multivariate time series. Inspired by these methods, we propose a gradient-based point importance scoring method for complex time series forecasting models.

Specifically, given a historical time series $\mathbf{X} = \{\mathbf{x}_{t-H+1}, \mathbf{x}_{t-H+2}, \ldots, \mathbf{x}_t\} \in \mathbb{R}^{H \times C}$ with historical length $H$ and $C$ channels, we first compute the raw temporal importance for each channel as:

$$w_t^{(c)} = \left\| \frac{\partial \mathcal{L}_{\text{MSE}}}{\partial \mathbf{x}_t^{(c)}} \right\|_1 \tag{1}$$

where $\mathcal{L}_{\text{MSE}}$ refers to Mean Squared Error which is well established to assess the prediction accuracy. To enable temporal comparisons across all channels, we apply channel-wise normalization:

$$\tilde{w}_t^{(c)} = \frac{w_t^{(c)} - \min_{k \in [1,T]} w_k^{(c)}}{\max_{k \in [1,T]} w_k^{(c)} - \min_{k \in [1,T]} w_k^{(c)}} \tag{2}$$

yielding normalized importance scores $\tilde{w}_t^{(c)} \in [0, 1]$ along the temporal dimension per channel. This represents each time point's relative contribution to the prediction within its channel context.

To generate a global temporal importance profile, we apply statistical interval estimation (Neyman, 1937) by aggregating $\tilde{w}_t^{(c)}$ from the test dataset $\mathcal{D}_{\text{test}}$, which comprises $N$ sequence samples used for forecasting. Because we only focus on the importance of different time points in each sequence, we treat the sequence from each channel as an individual sample. Consequently, the number of all samples for interval estimation achieving $N \times C$. For each time point $t$, we compute the mean of $\tilde{w}_t^{(c)}$, $i.e.$, $\mu(t)$, as follows:

$$\mu(t) = \frac{1}{N \times C} \sum_{n=1}^{N} \sum_{c=1}^{C} \tilde{w}_{t,n}^{(c)} \tag{3}$$

where $\tilde{w}_{t,n}^{(c)}$ denotes the normalized importance at time $t$ for channel $c$ in sequence $n$. The standard error of the mean is given by:

$$\text{SE}(t) = \frac{s(t)}{\sqrt{N \times C}} \text{ with } s(t) = \sqrt{\frac{1}{N \times C - 1} \sum_{n=1}^{N} \sum_{c=1}^{C} \left( \tilde{w}_{t,n}^{(c)} - \mu(t) \right)^2} \tag{4}$$

According to the central limit theorem (Kwak & Kim, 2017), the sampling distribution of $\mu(t)$ converges to a normal distribution as the number of samples increases. We therefore construct the 95% confidence interval:

$$\text{CI}_{95\%}(t) = [\mu(t) - z_{0.975} \cdot \text{SE}(t), \ \mu(t) + z_{0.975} \cdot \text{SE}(t)] \tag{5}$$

where $z_{0.975} \approx 1.96$ is the $97.5th$ percentile of the standard normal distribution. Based on the line plot of $\mu(t)$ and $\text{CI}_{95\%}(t)$, we can analyze the importance score distribution of historical time points for complex models.

## D    DETAILS OF RESAM

### D.1    PSEUDOCODE OF BARS

The pseudocode of the BARS strategy is shown in Alg. 1. The selected frequencies for the basis functions are categorized into four tiers based on temporal scales, encompassing a total of 179 distinct frequencies. The specifics are as follows (the right endpoint for each level is excluded.):

- **Minute-level** (5-minute interval): Frequencies range from 1 minute to 60 minutes, with a 5-minute step size. This tier comprises 12 frequencies.
- **Hour-level** (15-minute interval): Frequencies range from 60 minutes to 1440 minutes (24 hours), with a 15-minute step size. This tier comprises 92 frequencies.
- **Day-level** (6-hour interval): Frequencies range from 24 hours to 168 hours (7 days), with a 6-hour step size. This tier comprises 24 frequencies.
- **Week-level** (1-week interval): Frequencies range from 1 week to 52 weeks, with a 1-week step size. This tier comprises 51 frequencies.

---

**Algorithm 1** Basis-Aligned Randomized Sampling (BARS)

---

**Input:** Historical time series $\mathbf{X} \in \mathbb{R}^{H \times C}$, sampling rate $\rho$, regularization $\lambda$
**Output:** Basis coefficients $\theta \in \mathbb{R}^{2K \times C}$
1: $\mathcal{I} \leftarrow \{1, 2, \ldots, H\}$     *# Full time indices*
2: $\tau \leftarrow \text{RandomSample}(\mathcal{I}, \lfloor \rho H \rfloor)$     *# Random time point sampling*
3: $\mathbf{X}_{\text{sampled}} \leftarrow \mathbf{X}[\tau, :]$     *# Time series values*

   **Basis Transformation**
4: $K \leftarrow 179$     *# Frequencies: min/hr/day/week*
5: Initialize $\Phi \leftarrow \mathbf{0}_{\lfloor \rho H \rfloor \times 2K}$     *# Basis matrix*
6: **for** $k \in \{1, 2, \ldots, K\}$ **do**
7:     $\Phi[:, 2k-1] \leftarrow \sin(2\pi f_k \tau)$     *# Sine basis*
8:     $\Phi[:, 2k] \leftarrow \cos(2\pi f_k \tau)$     *# Cosine basis*
9: **end for**

   **Per-Channel Basis Coefficient Solution**
10: Initialize $\theta \leftarrow \mathbf{0}_{2K \times C}$     *# Coefficients matrix*
11: $\mathbf{A} \leftarrow \Phi^\top \Phi + \lambda I$
12: **for** $c = 1$ **to** $C$ **do**
13:     $\mathbf{b}_c \leftarrow \Phi^\top \mathbf{X}_{\text{sampled}}[:, c]$
14:     $\theta[:, c] \leftarrow \mathbf{A}^{-1} \mathbf{b}_c$
15: **end for**
16: **return** $\theta$

---

### D.2 PSEUDOCODE OF 2-STAGE TRAINING

The pseudocode for the BARS strategy is presented in Alg. 2. During stage 1, only the parameters of the LPE module are updated. In stage 2, both the backbone model and the LPE module are optimized simultaneously.

## E ABLATION STUDY

### E.1 CONTRIBUTIONS OF DIFFERENT COMPONENTS

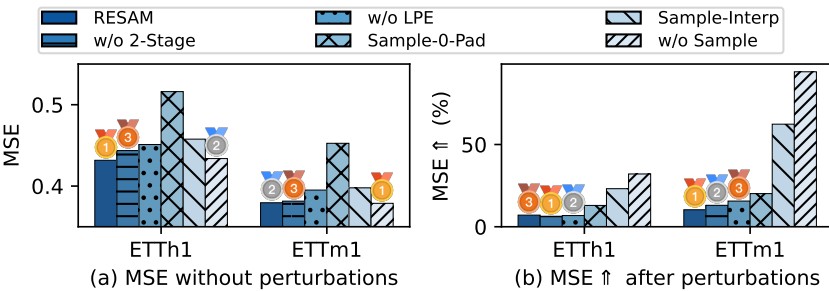

Figure 8: Ablation study on ETTh1 and ETTm1. $MSE \Uparrow$ is the proportional rise in MSE after perturbations. The results are averaged from all prediction lengths of $F \in \{96, 192, 336, 720\}$.

---

**Algorithm 2** 2-Stage Training

---

**Input:** Stage-1 single-layer linear model Linear, Stage-2 backbone model $\mathcal{M}$, Dataset $\mathcal{D}$, historical length $H$, forecast length $F$, sampling rate $\rho$, regularization $\lambda$
**Output:** Trained model parameters $\Theta_{\text{LPE}}, \Theta_{\mathcal{M}}$

---

    **Stage 1: Periodic Cycle Learning**
1: **for** each batch $(\mathbf{X}, \mathbf{Y}) \in \mathcal{D}$ **do**
2:     $\mathbf{P}_{\text{hist}}, \mathbf{P}_{\text{future}} \leftarrow \text{LPE}(\mathbf{X})$
3:     $\mathbf{R}_{\text{hist}} \leftarrow \mathbf{X} - \mathbf{P}_{\text{hist}}$
4:     $\hat{\mathbf{R}}_{\text{future}} \leftarrow \text{Linear}(\mathbf{R}_{\text{hist}})$
5:     $\hat{\mathbf{Y}} \leftarrow \hat{\mathbf{R}}_{\text{future}} + \mathbf{P}_{\text{future}}$
6:     $\mathcal{L} \leftarrow \text{MAE}(\mathbf{Y}, \hat{\mathbf{Y}})$
7:     Update $\Theta_{\text{LPE}}$ and linear layer parameters
8: **end for**

    **Stage 2: Joint Space Learning**
9: $\text{LPE}_{\text{init}} \leftarrow \Theta_{\text{LPE}}$                            *# Initialize with Stage 1 cycles*
10: **for** each batch $(\mathbf{X}, \mathbf{Y}) \in \mathcal{D}$ **do**
11:     $\mathbf{P}_{\text{hist}}, \mathbf{P}_{\text{future}} \leftarrow \text{LPE}(\mathbf{X})$
12:     $\mathbf{R}_{\text{hist}} \leftarrow \mathbf{X} - \mathbf{P}_{\text{hist}}$
13:     $\theta_{\text{hist}} \leftarrow \text{BARS}(\mathbf{R}_{\text{hist}}, \rho, \lambda)$
14:     $\hat{\theta}_{\text{future}} \leftarrow \mathcal{M}(\theta_{\text{hist}})$
15:     $\hat{\mathbf{R}}_{\text{future}} \leftarrow \Phi_{\text{future}}\hat{\theta}_{\text{future}}$
16:     $\hat{\mathbf{Y}} \leftarrow \hat{\mathbf{R}}_{\text{future}} + \mathbf{P}_{\text{future}}$
17:     $\mathcal{L} \leftarrow \text{MAE}(\mathbf{Y}, \hat{\mathbf{Y}})$
18:     Update $\Theta_{\text{LPE}}, \Theta_{\mathcal{M}}$
19: **end for**

---

To evaluate the impact of each component in RESAM, we conduct comprehensive ablation studies using five architectural variants: (1) **RESAM w/o 2-Stage** utilizes only joint space learning, removing the periodic cycle learning stage; (2) **RESAM w/o LPE** removes the LPE module; (3) **RESAM Sample-0-Pad** substitutes basis transformation in BARS with zero-padding in the time domain; (4) **RESAM Sample-Interp** replaces basis transformation in BARS with linear interpolation in the time domain; and (5) **RESAM w/o Sample** eliminates time point sampling during training, thus utilizing all available time points. Experiments are conducted on the ETTh1 and ETTm1 datasets, analyzing both forecasting accuracy (MSE) without perturbations and robustness ($MSE \Uparrow$) under last-point perturbations, as presented in Fig. 8.

Subfigure (a) of Fig. 8 illustrates the forecasting accuracy of different variants. All variants, except RESAM w/o Sample, exhibit an increase in MSE. This underscores the critical importance of the LPE module and two-stage training techniques for achieving optimal forecasting accuracy. The comparable or superior performance of RESAM w/o Sample relative to the full RESAM suggests that the approach of sampling and transforming into the basis domain successfully preserves predictive capability while enhancing robustness. It is noteworthy that time-domain sampling variants, particularly RESAM Sample-Interp and RESAM Sample-0-Pad, suffer significant accuracy degradation. In particular, the 0-padding method performs worse due to its introduction of artificial values that disrupt temporal patterns.

In subfigure (b) of Fig. 8, we evaluate the robustness of different variants against last-point perturbations. We find that different variants exhibit unique patterns of increase in MSE. Although RESAM w/o Sample demonstrates strong performance in unperturbed time series forecasting, its MSE significantly increases after perturbations. This highlights time point sampling as the critical role of perturbation robustness. RESAM Sample-Interp also exhibits a high $MSE \Uparrow$, showing poor robustness due to preserved temporal patterns. By randomly filtering some time points, RESAM Sample-0-Pad achieves a lower $MSE \Uparrow$ compared to the aforementioned linear interpolation variant. Notably, RESAM, even without utilizing the LPE module or undergoing 2-stage training, maintains relatively low $MSE \Uparrow$, indicating that the BARS strategy significantly contributes to robustness.

Overall, the integration of random sampling, basis transformation, learnable periodicity extraction, and the two-stage training process allows RESAM to effectively balance forecasting accuracy with robustness against perturbations.

### E.2 Effectiveness of 2-Stage Training

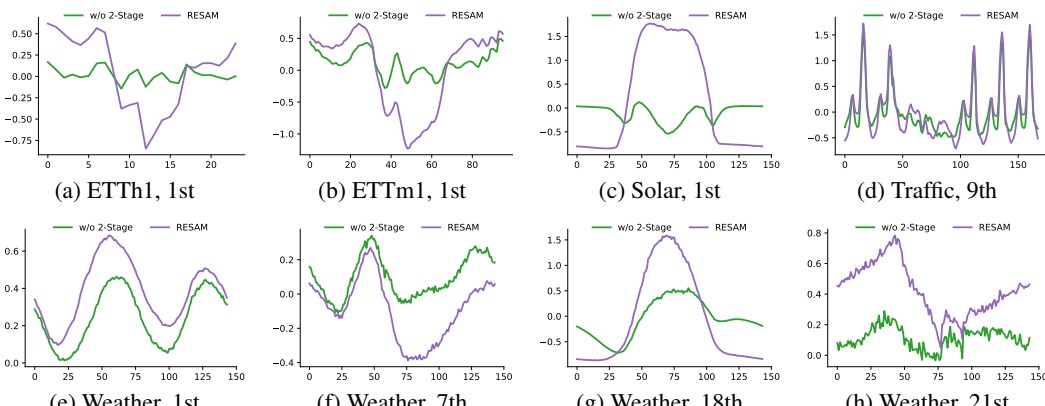

(a) ETTh1, 1st     (b) ETTm1, 1st     (c) Solar, 1st     (d) Traffic, 9th

(e) Weather, 1st     (f) Weather, 7th     (g) Weather, 18th     (h) Weather, 21st

Figure 9: Visualization of the periodic cycles learned by RESAM with and without 2-stage training. Panels (a-d) illustrate various periodic cycles extracted from different datasets. Conversely, panels (e-h) showcase diverse periodic cycles identified from different channels within the same dataset. The $i$ th denotes the index of the channel within the dataset.

To further investigate the impact of 2-Stage-Training, we visualize the learned periodic cycles with and without this training approach. As illustrated in Fig. 9, without 2-Stage-Training, the learned periodic cycles tend to cluster around zero with minor fluctuations, which impairs the accurate representation of the data's inherent periodicity. Conversely, employing 2-Stage-Training allows RESAM to derive more precise periodic cycles. In Fig. 9a and 9b, the periodic cycles extracted from ETTh1 and ETTm1 by RESAM show a high degree of consistency, aligning with the expectation given that the primary difference between ETTh1 and ETTm1 lies in the sampling frequency. Fig. 9d highlights the distinct traffic data patterns observed on weekdays compared to weekends. Furthermore, as depicted in Fig. 9e, 9f, 9g, and 9h, different channels exhibit varying periodic patterns, warranting the use of an independent cycle for each channel when designing the cycle.

## F Parameter sensitivity

This section investigates how different parameters—including the sampling rate $\rho$, perturbation ratio $\alpha$, historical sequence length $H$, and prediction length $F$—affect the accuracy and robustness of RESAM in forecasting tasks. We perform a series of experiments using the ETTm1 and ETTm2 datasets, and the results are depicted in Fig. 10. In each subfigure, MSE denotes the forecasting accuracy on clean data, whereas $\hat{\text{MSE}}$ and $MSE \Uparrow$ indicate robustness against point-wise perturbations.

**Sampling rate** $\rho$. RESAM enhances the use of global context information by employing randomized sampling during training. Fig. 10a reveals key insights on the effects of sampling rate $\rho$. Firstly, forecasting accuracy slightly increases with a higher $\rho$, although the difference between $\rho \in \{0.60, 0.75, 0.90\}$ is minimal. Secondly, perturbation robustness decreases as $\rho$ increases; specifically, $\rho = 0.60$ exhibits the lowest degradation under last-point perturbations, while $\rho = 0.90$ shows the highest degradation. This is bacause fewer sampled time points during training encourage RESAM to learn predictions based on complete contextual information rather than isolated points. Among the three sampling rates, the $\rho = 0.75$ configuration effectively balances forecasting accuracy and robustness.

**Perturbation ratio** $\alpha$. Fig. 10b demonstrates the effect of varying perturbation ratios on the robustness of RESAM. Both the prediction $\hat{\text{MSE}}$ and $MSE \Uparrow$ after pertuabations exhibit an approximately linear relationship with the perturbation ratio $\alpha$, increasing as $\alpha$ grows. Notably, under in-

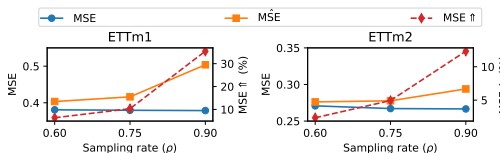
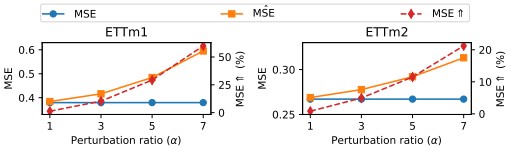

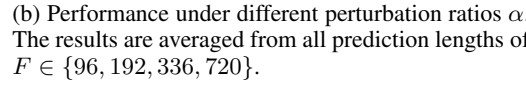

(a) Performance under different sampling rates $\rho$. The results are averaged from all prediction lengths of $F \in \{96, 192, 336, 720\}$.

(b) Performance under different perturbation ratios $\alpha$. The results are averaged from all prediction lengths of $F \in \{96, 192, 336, 720\}$.

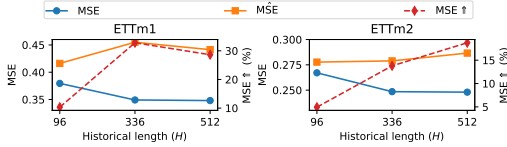
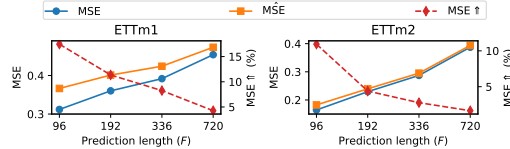

(c) Performance under different historical sequence lengths $H$. The results are averaged from all prediction lengths of $F \in \{96, 192, 336, 720\}$.

(d) Performance under different prediction lengths $F$.

Figure 10: Model analysis for RESAM in cases of different sampling rates $\rho$, perturbation ratios $\alpha$, historical sequence lengths $H$ and prediction lengths $F$. Experiments are conducted on ETTm1 and ETTm2 with historical sequence length $H = 96$ as default. All perturbations here are applied to the last point of historical sequence with perturbation ratio $\alpha = 3$ as default. $\hat{MSE}$ refers to MSE after perturbations, and $MSE \Uparrow$ is the proportional rise in MSE after perturbations.

distribution perturbations ($\alpha = 1.0$), which simulate typical measurement noise, RESAM maintains nearly identical forecasting accuracy compared to scenarios without perturbations. This stability demonstrates RESAM's inherent resilience to common data collection variability, highlighting its robustness for practical applications.

**Historical sequence length** $H$. As demonstrated in Fig. 10c, prediction accuracy peaks at $H = 336$ while diminishing at $H = 96$, indicating longer historical context windows are beneficial for latent temporal pattern learning. However, further extension to $H = 512$ does not result in improved accuracy, suggesting there are diminishing returns beyond optimal context lengths. Under point-wise perturbations, models with $H = 96$ exhibit the least performance degradation, whereas longer sequences show increased sensitivity to such perturbations. This may be attributed to the fact that shorter sequences maintain tighter coupling between time points, thereby enhancing their resilience to perturbations when a single point is distorted.

**Prediction length** $F$. As illustrated in Fig. 10d, increasing the prediction length leads to a higher MSE due to error accumulation over longer horizons. However, when considering point-wise perturbations, longer prediction horizons exhibit less relative performance degradation. This is because a perturbation at the last point mainly impacts predictions that are temporally closer. The farther the predicted point is from the perturbated point, the less affected it will be. As the prediction length increases, the perturbation of a single point has a smaller average effect on the overall prediction, resulting in a smaller decrease in prediction accuracy after perturbations.

## G  ANALYSIS OF DIFFERENT PERTURBATION TYPES

This is a newly added section during rebuttal period.

To thoroughly evaluate the robustness of RESAM against diverse data corruption scenarios, we extend our experiments to include seven different perturbation types across two temporal positions. This comprehensive analysis provides a more complete assessment of model performance under various realistic data quality issues.

### G.1  PERTURBATION TAXONOMY AND EXPERIMENTAL SETUP

We categorize perturbations based on two dimensions: *temporal position* and *corruption type*. The temporal position distinguishes between:

- **Last perturbations**: Affecting the most recent points in the historical window
- **Random perturbations**: Affecting randomly selected positions throughout the sequence

The corruption types include:

- **Single anomalous point**: Single point corruption with perturbation ratio $\alpha = 3.0$. This type is consistent with experiment in 4.2.
- **Block**: Continuous segment corruption with random length 2-5
- **Missing point**: Complete absence of value (set to zero)
- **Multi anomalous points**: Multiple isolated point corruptions (2-5 points)

This taxonomy yields seven distinct perturbation scenarios that cover common real-world data quality issues. All experiments are conducted with perturbation ratio $\alpha = 3.0$, and results are averaged across prediction lengths $F \in \{96, 192, 336, 720\}$ to ensure comprehensive evaluation.

## G.2 EXPERIMENTAL RESULTS AND ANALYSIS

Table 5: Robustness evaluation under last-block perturbation ratio $\alpha = 3.0$. Results are averaged from prediction lengths $F \in \{96, 192, 336, 720\}$. Post-perturbation MSE and MAE are reported.

| Models | ETTh1 | | ETTh2 | | ETTm1 | | ETTm2 | | Weather | | Exchange | | Traffic | | Solar | | 1st Count | Avg Rank |
|---|---|---|---|---|---|---|---|---|---|---|---|---|---|---|---|---|---|---|
| | MSE | MAE | MSE | MAE | MSE | MAE | MSE | MAE | MSE | MAE | MSE | MAE | MSE | MAE | MSE | MAE | | |
| DLinear | 0.594 | 0.507 | 0.456 | 0.457 | 0.654 | 0.488 | 0.337 | 0.371 | 0.388 | 0.351 | 0.373 | 0.435 | 1.113 | 0.481 | 3.403 | 0.877 | 5 | 5.01 |
| TimeMixer | 0.617 | 0.514 | 0.434 | 0.436 | 0.617 | 0.480 | 0.344 | 0.370 | 0.324 | 0.326 | 0.484 | 0.490 | 0.610 | 0.378 | 1.073 | 0.593 | 0 | 4.05 |
| CycleNet | 0.589 | 0.495 | 0.450 | 0.446 | 0.895 | 0.596 | 0.436 | 0.441 | 0.423 | 0.380 | 0.459 | 0.472 | 0.686 | 0.386 | 1.156 | 0.576 | 0 | 5.34 |
| Crossformer | 0.536 | 0.512 | 0.895 | 0.658 | 0.522 | 0.472 | 0.583 | 0.503 | 0.282 | 0.309 | 0.619 | 0.596 | 0.681 | 0.352 | 0.342 | 0.295 | 10 | 3.99 |
| PatchTST | 0.583 | 0.500 | 0.405 | 0.419 | 0.681 | 0.501 | 0.350 | 0.371 | 0.311 | 0.317 | 0.483 | 0.493 | 0.631 | 0.365 | 0.850 | 0.546 | 0 | 3.62 |
| MICN | 0.503 | 0.475 | 0.475 | 0.460 | 0.495 | 0.446 | 0.312 | 0.363 | 0.315 | 0.326 | 0.329 | 0.401 | 0.625 | 0.320 | 0.370 | 0.354 | 7 | 2.01 |
| TimesNet | 0.501 | 0.468 | 0.412 | 0.419 | 0.445 | 0.428 | 0.309 | 0.342 | 0.292 | 0.308 | 0.443 | 0.454 | 0.766 | 0.397 | 0.439 | 0.399 | 3 | 2.32 |
| FEDformer | 0.487 | 0.472 | 0.410 | 0.427 | 0.959 | 0.642 | 0.353 | 0.393 | 0.357 | 0.373 | 0.530 | 0.497 | 0.691 | 0.408 | 0.823 | 0.680 | 4 | 4.61 |
| FreTS | 0.734 | 0.553 | 0.509 | 0.473 | 0.894 | 0.564 | 0.422 | 0.413 | 0.307 | 0.334 | 0.445 | 0.472 | 0.837 | 0.436 | 1.541 | 0.627 | 0 | 5.66 |
| FiLM | 0.530 | 0.474 | 0.400 | 0.414 | 0.584 | 0.465 | 0.330 | 0.359 | 0.342 | 0.326 | 0.400 | 0.435 | 1.321 | 0.722 | 1.673 | 0.730 | 0 | 3.63 |
| iTransformer | 0.542 | 0.491 | 0.424 | 0.432 | 0.655 | 0.505 | 0.349 | 0.374 | 0.327 | 0.328 | 0.439 | 0.465 | 0.633 | 0.398 | 0.968 | 0.543 | 1 | 3.86 |
| TimeKAN | 0.535 | 0.478 | 0.414 | 0.426 | 0.551 | 0.462 | 0.337 | 0.367 | 0.304 | 0.315 | 0.465 | 0.477 | 0.751 | 0.412 | 1.037 | 0.573 | 2 | 3.37 |
| Leddam | 0.535 | 0.482 | 0.421 | 0.428 | 0.647 | 0.493 | 0.372 | 0.383 | 0.397 | 0.367 | 0.461 | 0.473 | 0.674 | 0.383 | 3.153 | 0.909 | 0 | 4.66 |
| RobustTSF | 0.639 | 0.525 | 0.512 | 0.488 | 0.749 | 0.514 | 0.364 | 0.391 | 0.479 | 0.383 | 0.417 | 0.440 | 1.034 | 0.491 | 2.073 | 0.762 | 0 | 6.02 |
| RESAM | 0.492 | 0.459 | 0.391 | 0.410 | 0.436 | 0.434 | 0.290 | 0.329 | 0.276 | 0.296 | 0.458 | 0.483 | 0.596 | 0.340 | 1.618 | 0.609 | 32 | 1.85 |

Table 6: Robustness evaluation under last-missing-point perturbation. Results are averaged from prediction lengths $F \in \{96, 192, 336, 720\}$. Post-perturbation MSE and MAE are reported.

| Models | ETTh1 | | ETTh2 | | ETTm1 | | ETTm2 | | Weather | | Exchange | | Traffic | | Solar | | 1st Count | Avg Rank |
|---|---|---|---|---|---|---|---|---|---|---|---|---|---|---|---|---|---|---|
| | MSE | MAE | MSE | MAE | MSE | MAE | MSE | MAE | MSE | MAE | MSE | MAE | MSE | MAE | MSE | MAE | | |
| DLinear | 0.520 | 0.486 | 0.451 | 0.478 | 0.532 | 0.475 | 0.409 | 0.433 | 77.463 | 2.232 | 19.005 | 2.652 | 0.729 | 0.383 | 0.771 | 0.440 | 2 | 5.28 |
| TimeMixer | 0.534 | 0.490 | 0.416 | 0.443 | 0.474 | 0.446 | 0.391 | 0.422 | 340.153 | 4.787 | 29.535 | 4.570 | 0.516 | 0.317 | 0.343 | 0.328 | 3 | 4.41 |
| CycleNet | 0.514 | 0.473 | 0.451 | 0.470 | 0.747 | 0.587 | 0.494 | 0.483 | 68.055 | 2.266 | 10.530 | 2.924 | 0.517 | 0.310 | 0.442 | 0.341 | 1 | 5.22 |
| Crossformer | 0.548 | 0.521 | 0.887 | 0.655 | 0.548 | 0.489 | 0.554 | 0.494 | 0.488 | 0.438 | 1.851 | 1.109 | 0.664 | 0.338 | 0.310 | 0.265 | 6 | 4.02 |
| PatchTST | 0.543 | 0.498 | 0.399 | 0.427 | 0.539 | 0.492 | 0.425 | 0.454 | 2.503 | 0.561 | 10.197 | 3.026 | 0.588 | 0.333 | 0.309 | 0.324 | 1 | 3.91 |
| MICN | 0.481 | 0.473 | 0.467 | 0.473 | 0.471 | 0.448 | 0.331 | 0.396 | 27.866 | 1.569 | 5.791 | 2.253 | 0.622 | 0.314 | 0.357 | 0.315 | 1 | 3.38 |
| TimesNet | 0.471 | 0.454 | 0.393 | 0.412 | 0.414 | 0.410 | 0.303 | 0.344 | 28.109 | 1.622 | 2.353 | 1.188 | 0.682 | 0.350 | 0.350 | 0.328 | 0 | 2.84 |
| FEDformer | 0.464 | 0.457 | 0.416 | 0.432 | 0.649 | 0.562 | 0.365 | 0.404 | 26.379 | 2.979 | 0.523 | 0.507 | 0.696 | 0.406 | 1.043 | 0.742 | 5 | 4.45 |
| FreTS | 0.613 | 0.532 | 0.495 | 0.490 | 0.587 | 0.512 | 0.537 | 0.513 | 4.172 | 0.631 | 8.267 | 2.574 | 0.619 | 0.360 | 0.327 | 0.290 | 0 | 4.91 |
| FiLM | 0.474 | 0.448 | 0.385 | 0.408 | 0.467 | 0.436 | 0.362 | 0.392 | 15.618 | 1.114 | 1.260 | 0.791 | 1.251 | 0.693 | 0.503 | 0.420 | 3 | 3.29 |
| iTransformer | 0.504 | 0.480 | 0.406 | 0.439 | 0.536 | 0.496 | 0.415 | 0.448 | 31.469 | 2.012 | 7.377 | 2.582 | 0.534 | 0.332 | 0.401 | 0.316 | 1 | 4.25 |
| TimeKAN | 0.495 | 0.462 | 0.396 | 0.427 | 0.468 | 0.442 | 0.366 | 0.404 | 156.218 | 2.945 | 13.858 | 3.206 | 0.663 | 0.366 | 0.349 | 0.305 | 1 | 3.72 |
| Leddam | 0.494 | 0.468 | 0.408 | 0.441 | 0.538 | 0.488 | 0.423 | 0.448 | 69.210 | 2.582 | 10.800 | 3.123 | 0.595 | 0.341 | 0.321 | 0.280 | 0 | 4.23 |
| RobustTSF | 0.474 | 0.463 | 0.623 | 0.559 | 0.454 | 0.437 | 0.523 | 0.497 | 0.387 | 0.410 | 1.131 | 0.816 | 0.663 | 0.375 | 0.556 | 0.489 | 7 | 4.22 |
| RESAM | 0.455 | 0.440 | 0.373 | 0.407 | 0.394 | 0.394 | 0.277 | 0.330 | 3.029 | 0.732 | 69.465 | 7.477 | 0.555 | 0.308 | 0.306 | 0.276 | 33 | 1.89 |

Results of last perturbations are presented in Tbl. 2, Tbl. 5 and Tbl. 6. Results of random perturbations are presented in Tbl. 7, Tbl. 8, Tbl. 9 and Tbl. 10. The comprehensive evaluation demonstrates RESAM's consistent superiority across all perturbation types. Several key observations emerge from the results: First, RESAM exhibits remarkable robustness against **last-position perturbations**, outperforming all baseline methods. Traditional models show severe performance degradation due to their over-reliance on recent temporal points. Second, for **random-position perturbations**, while

Table 7: Robustness evaluation under random-anomalous-point perturbation ratio $\alpha = 3.0$. Results are averaged from prediction lengths $F \in \{96, 192, 336, 720\}$. Post-perturbation MSE and MAE are reported.

| Models | ETTh1 | | ETTh2 | | ETTm1 | | ETTm2 | | Weather | | Exchange | | Traffic | | Solar | | 1st Count | Avg Rank |
|---|---|---|---|---|---|---|---|---|---|---|---|---|---|---|---|---|---|---|
| | MSE | MAE | MSE | MAE | MSE | MAE | MSE | MAE | MSE | MAE | MSE | MAE | MSE | MAE | MSE | MAE | | |
| DLinear | 0.454 | 0.442 | 0.418 | 0.430 | 0.403 | 0.392 | 0.287 | 0.331 | 0.272 | 0.291 | 0.318 | 0.387 | 0.723 | 0.372 | 0.427 | 0.336 | 0 | 3.52 |
| TimeMixer | 0.469 | 0.444 | 0.376 | 0.396 | 0.405 | 0.392 | 0.280 | 0.320 | 0.246 | 0.266 | 0.383 | 0.416 | 0.509 | 0.317 | 0.358 | 0.338 | 4 | 2.41 |
| CycleNet | 0.435 | 0.422 | 0.377 | 0.396 | 0.392 | 0.387 | 0.275 | 0.314 | 0.259 | 0.277 | 0.394 | 0.421 | 0.520 | 0.312 | 0.294 | 0.271 | 6 | 1.87 |
| Crossformer | 0.509 | 0.492 | 0.977 | 0.688 | 0.487 | 0.448 | 0.555 | 0.487 | 0.242 | 0.273 | 0.606 | 0.585 | 0.578 | 0.279 | 0.223 | 0.220 | 15 | 3.80 |
| PatchTST | 0.447 | 0.432 | 0.368 | 0.391 | 0.393 | 0.389 | 0.280 | 0.319 | 0.258 | 0.274 | 0.376 | 0.410 | 0.514 | 0.304 | 0.295 | 0.313 | 9 | 1.84 |
| MICN | 0.438 | 0.444 | 0.463 | 0.451 | 0.389 | 0.392 | 0.285 | 0.338 | 0.247 | 0.276 | 0.309 | 0.379 | 0.613 | 0.310 | 0.313 | 0.309 | 11 | 2.63 |
| TimesNet | 0.470 | 0.452 | 0.394 | 0.407 | 0.414 | 0.408 | 0.292 | 0.327 | 0.259 | 0.281 | 0.432 | 0.446 | 0.701 | 0.360 | 0.379 | 0.352 | 0 | 3.84 |
| FEDformer | 0.454 | 0.451 | 0.407 | 0.425 | 0.599 | 0.530 | 0.342 | 0.383 | 0.305 | 0.337 | 0.530 | 0.497 | 0.654 | 0.380 | 0.548 | 0.551 | 1 | 4.80 |
| FreTS | 0.510 | 0.470 | 0.517 | 0.456 | 0.448 | 0.418 | 0.303 | 0.342 | 0.242 | 0.272 | 0.394 | 0.425 | 0.550 | 0.325 | 0.288 | 0.281 | 1 | 3.29 |
| FiLM | 0.452 | 0.433 | 0.386 | 0.403 | 0.408 | 0.391 | 0.288 | 0.324 | 0.283 | 0.288 | 0.376 | 0.418 | 1.241 | 0.690 | 0.423 | 0.382 | 0 | 3.59 |
| RESAM | 0.436 | 0.426 | 0.371 | 0.394 | 0.383 | 0.382 | 0.269 | 0.310 | 0.250 | 0.272 | 0.358 | 0.404 | 0.546 | 0.316 | 0.290 | 0.265 | 17 | 1.41 |

Table 8: Robustness evaluation under random-block perturbation ratio $\alpha = 3.0$. Results are averaged from prediction lengths $F \in \{96, 192, 336, 720\}$. Post-perturbation MSE and MAE are reported.

| Models | ETTh1 | | ETTh2 | | ETTm1 | | ETTm2 | | Weather | | Exchange | | Traffic | | Solar | | 1st Count | Avg Rank |
|---|---|---|---|---|---|---|---|---|---|---|---|---|---|---|---|---|---|---|
| | MSE | MAE | MSE | MAE | MSE | MAE | MSE | MAE | MSE | MAE | MSE | MAE | MSE | MAE | MSE | MAE | | |
| DLinear | 0.469 | 0.452 | 0.420 | 0.431 | 0.417 | 0.400 | 0.289 | 0.333 | 0.276 | 0.292 | 0.319 | 0.387 | 0.856 | 0.411 | 0.439 | 0.344 | 0 | 4.54 |
| TimeMixer | 0.485 | 0.453 | 0.383 | 0.401 | 0.422 | 0.402 | 0.282 | 0.322 | 0.248 | 0.268 | 0.391 | 0.421 | 0.538 | 0.343 | 0.483 | 0.399 | 0 | 3.53 |
| CycleNet | 0.452 | 0.432 | 0.381 | 0.398 | 0.651 | 0.520 | 0.383 | 0.409 | 0.323 | 0.333 | 0.403 | 0.426 | 0.573 | 0.337 | 0.365 | 0.316 | 3 | 4.27 |
| Crossformer | 0.515 | 0.497 | 0.892 | 0.656 | 0.493 | 0.451 | 0.557 | 0.488 | 0.243 | 0.274 | 0.606 | 0.585 | 0.670 | 0.344 | 0.237 | 0.230 | 13 | 5.26 |
| PatchTST | 0.466 | 0.445 | 0.369 | 0.392 | 0.406 | 0.397 | 0.282 | 0.321 | 0.262 | 0.277 | 0.384 | 0.415 | 0.586 | 0.339 | 0.363 | 0.338 | 7 | 2.63 |
| MICN | 0.439 | 0.445 | 0.463 | 0.451 | 0.398 | 0.398 | 0.286 | 0.338 | 0.251 | 0.279 | 0.310 | 0.380 | 0.616 | 0.313 | 0.335 | 0.327 | 12 | 2.88 |
| TimesNet | 0.492 | 0.462 | 0.404 | 0.413 | 0.433 | 0.420 | 0.300 | 0.332 | 0.266 | 0.287 | 0.438 | 0.449 | 0.767 | 0.397 | 0.425 | 0.394 | 0 | 5.60 |
| FEDformer | 0.462 | 0.457 | 0.408 | 0.426 | 0.659 | 0.550 | 0.343 | 0.384 | 0.309 | 0.340 | 0.530 | 0.497 | 0.669 | 0.392 | 0.583 | 0.570 | 1 | 6.34 |
| FreTS | 0.525 | 0.478 | 0.432 | 0.429 | 0.562 | 0.464 | 0.305 | 0.346 | 0.245 | 0.276 | 0.394 | 0.426 | 0.603 | 0.358 | 0.348 | 0.315 | 0 | 4.45 |
| FiLM | 0.466 | 0.442 | 0.390 | 0.406 | 0.424 | 0.399 | 0.290 | 0.325 | 0.284 | 0.289 | 0.382 | 0.420 | 1.257 | 0.696 | 0.437 | 0.390 | 0 | 4.75 |
| iTransformer | 0.483 | 0.458 | 0.391 | 0.408 | 0.431 | 0.411 | 0.288 | 0.326 | 0.258 | 0.275 | 0.373 | 0.411 | 0.508 | 0.332 | 0.327 | 0.314 | 4 | 3.52 |
| TimeKAN | 0.467 | 0.440 | 0.381 | 0.400 | 0.410 | 0.398 | 0.285 | 0.324 | 0.253 | 0.273 | 0.404 | 0.428 | 0.690 | 0.381 | 0.384 | 0.339 | 0 | 3.38 |
| Leddam | 0.458 | 0.438 | 0.375 | 0.395 | 0.419 | 0.400 | 0.282 | 0.321 | 0.242 | 0.264 | 0.370 | 0.408 | 0.543 | 0.323 | 0.900 | 0.454 | 3 | 2.55 |
| RobustTSF | 0.471 | 0.454 | 0.440 | 0.442 | 0.412 | 0.400 | 0.286 | 0.334 | 0.271 | 0.294 | 0.375 | 0.396 | 0.743 | 0.404 | 0.356 | 0.357 | 3 | 4.27 |
| RESAM | 0.446 | 0.431 | 0.375 | 0.397 | 0.392 | 0.390 | 0.272 | 0.313 | 0.253 | 0.275 | 0.367 | 0.411 | 0.587 | 0.332 | 0.378 | 0.326 | 18 | 2.02 |

Table 9: Robustness evaluation under random-missing-point perturbation. Results are averaged from prediction lengths $F \in \{96, 192, 336, 720\}$. Post-perturbation MSE and MAE are reported.

| Models | ETTh1 | | ETTh2 | | ETTm1 | | ETTm2 | | Weather | | Exchange | | Traffic | | Solar | | 1st Count | Avg Rank |
|---|---|---|---|---|---|---|---|---|---|---|---|---|---|---|---|---|---|---|
| | MSE | MAE | MSE | MAE | MSE | MAE | MSE | MAE | MSE | MAE | MSE | MAE | MSE | MAE | MSE | MAE | | |
| DLinear | 0.451 | 0.442 | 0.417 | 0.431 | 0.400 | 0.392 | 0.287 | 0.333 | 1.691 | 0.372 | 0.571 | 0.457 | 0.678 | 0.360 | 0.340 | 0.314 | 2 | 4.25 |
| TimeMixer | 0.470 | 0.446 | 0.375 | 0.398 | 0.401 | 0.391 | 0.280 | 0.323 | 13.074 | 0.768 | 2.027 | 0.950 | 0.498 | 0.306 | 0.338 | 0.325 | 0 | 4.16 |
| CycleNet | 0.434 | 0.421 | 0.376 | 0.396 | 0.616 | 0.509 | 0.387 | 0.415 | 2.971 | 0.619 | 1.775 | 0.918 | 0.494 | 0.296 | 0.320 | 0.280 | 5 | 4.23 |
| Crossformer | 0.511 | 0.493 | 0.888 | 0.655 | 0.486 | 0.446 | 0.554 | 0.487 | 0.257 | 0.284 | 0.696 | 0.627 | 0.654 | 0.333 | 0.220 | 0.219 | 13 | 4.75 |
| PatchTST | 0.443 | 0.432 | 0.367 | 0.392 | 0.389 | 0.387 | 0.278 | 0.320 | 0.316 | 0.296 | 1.199 | 0.745 | 0.551 | 0.315 | 0.274 | 0.304 | 5 | 2.42 |
| MICN | 0.439 | 0.445 | 0.461 | 0.451 | 0.389 | 0.392 | 0.283 | 0.337 | 0.825 | 0.377 | 0.509 | 0.460 | 0.612 | 0.309 | 0.284 | 0.282 | 2 | 3.51 |
| TimesNet | 0.465 | 0.450 | 0.399 | 0.413 | 0.404 | 0.403 | 0.297 | 0.334 | 13.479 | 1.168 | 1.454 | 0.897 | 0.681 | 0.349 | 0.365 | 0.338 | 0 | 5.81 |
| FEDformer | 0.453 | 0.450 | 0.406 | 0.426 | 0.577 | 0.524 | 0.340 | 0.383 | 2.670 | 0.910 | 0.470 | 0.469 | 0.656 | 0.380 | 0.670 | 0.597 | 1 | 5.72 |
| FreTS | 0.503 | 0.467 | 0.427 | 0.428 | 0.420 | 0.407 | 0.303 | 0.347 | 0.373 | 0.327 | 0.645 | 0.499 | 0.533 | 0.313 | 0.264 | 0.262 | 0 | 4.23 |
| FiLM | 0.449 | 0.432 | 0.386 | 0.405 | 0.405 | 0.391 | 0.288 | 0.325 | 1.116 | 0.389 | 1.219 | 0.684 | 1.236 | 0.687 | 0.382 | 0.365 | 0 | 4.45 |
| iTransformer | 0.455 | 0.444 | 0.385 | 0.406 | 0.398 | 0.394 | 0.285 | 0.327 | 1.281 | 0.463 | 1.448 | 0.901 | 0.460 | 0.290 | 0.255 | 0.250 | 5 | 3.66 |
| TimeKAN | 0.452 | 0.432 | 0.377 | 0.398 | 0.391 | 0.388 | 0.281 | 0.323 | 8.772 | 0.611 | 1.854 | 0.896 | 0.642 | 0.355 | 0.323 | 0.293 | 0 | 3.88 |
| Leddam | 0.444 | 0.431 | 0.373 | 0.395 | 0.395 | 0.388 | 0.279 | 0.322 | 8.217 | 0.838 | 1.300 | 0.766 | 0.504 | 0.287 | 0.232 | 0.232 | 6 | 2.64 |
| RobustTSF | 0.447 | 0.440 | 0.443 | 0.445 | 0.396 | 0.392 | 0.289 | 0.338 | 0.271 | 0.294 | 0.401 | 0.411 | 0.641 | 0.362 | 0.343 | 0.344 | 7 | 3.88 |
| RESAM | 0.434 | 0.425 | 0.369 | 0.394 | 0.379 | 0.380 | 0.267 | 0.311 | 1.301 | 0.531 | 4.390 | 1.151 | 0.552 | 0.305 | 0.246 | 0.245 | 18 | 2.40 |

Table 10: Robustness evaluation under random-multi-anomalous-point perturbation ratio $\alpha = 3.0$. Results are averaged from prediction lengths $F \in \{96, 192, 336, 720\}$. Post-perturbation MSE and MAE are reported.

| Models | ETTh1 | | ETTh2 | | ETTm1 | | ETTm2 | | Weather | | Exchange | | Traffic | | Solar | | 1st Count | Avg Rank |
|---|---|---|---|---|---|---|---|---|---|---|---|---|---|---|---|---|---|---|
| | MSE | MAE | MSE | MAE | MSE | MAE | MSE | MAE | MSE | MAE | MSE | MAE | MSE | MAE | MSE | MAE | | |
| DLinear | 0.500 | 0.470 | 0.427 | 0.436 | 0.447 | 0.419 | 0.295 | 0.338 | 0.282 | 0.297 | 0.324 | 0.392 | 1.078 | 0.487 | 0.749 | 0.442 | 0 | 4.49 |
| TimeMixer | 0.520 | 0.472 | 0.391 | 0.407 | 0.465 | 0.427 | 0.292 | 0.330 | 0.259 | 0.279 | 0.400 | 0.428 | 0.595 | 0.389 | 0.580 | 0.458 | 3 | 3.43 |
| CycleNet | 0.478 | 0.448 | 0.393 | 0.406 | 0.716 | 0.547 | 0.406 | 0.425 | 0.349 | 0.347 | 0.416 | 0.434 | 0.670 | 0.398 | 0.479 | 0.387 | 0 | 4.38 |
| Crossformer | 0.531 | 0.509 | 0.896 | 0.658 | 0.506 | 0.461 | 0.562 | 0.491 | 0.247 | 0.278 | 0.608 | 0.586 | 0.703 | 0.369 | 0.263 | 0.247 | 13 | 4.66 |
| PatchTST | 0.505 | 0.468 | 0.376 | 0.397 | 0.441 | 0.417 | 0.290 | 0.327 | 0.266 | 0.283 | 0.401 | 0.425 | 0.692 | 0.401 | 0.519 | 0.422 | 7 | 2.90 |
| MICN | 0.448 | 0.451 | 0.466 | 0.453 | 0.422 | 0.415 | 0.289 | 0.341 | 0.262 | 0.291 | 0.310 | 0.381 | 0.630 | 0.322 | 0.631 | 0.536 | 15 | 2.69 |
| TimesNet | 0.539 | 0.483 | 0.428 | 0.427 | 0.477 | 0.445 | 0.314 | 0.344 | 0.282 | 0.300 | 0.448 | 0.457 | 0.900 | 0.465 | 0.519 | 0.460 | 0 | 5.67 |
| FEDformer | 0.481 | 0.468 | 0.410 | 0.427 | 0.752 | 0.585 | 0.345 | 0.386 | 0.316 | 0.345 | 0.530 | 0.497 | 0.693 | 0.409 | 0.643 | 0.604 | 1 | 5.78 |
| FreTS | 0.559 | 0.496 | 0.443 | 0.435 | 0.751 | 0.533 | 0.315 | 0.354 | 0.251 | 0.283 | 0.399 | 0.430 | 0.701 | 0.418 | 0.464 | 0.390 | 0 | 4.56 |
| FiLM | 0.493 | 0.459 | 0.400 | 0.413 | 0.452 | 0.419 | 0.296 | 0.331 | 0.291 | 0.295 | 0.389 | 0.425 | 1.288 | 0.709 | 0.587 | 0.457 | 0 | 4.34 |
| iTransformer | 0.534 | 0.483 | 0.409 | 0.419 | 0.497 | 0.446 | 0.303 | 0.337 | 0.264 | 0.282 | 0.386 | 0.421 | 0.588 | 0.395 | 0.531 | 0.418 | 1 | 4.12 |
| TimeKAN | 0.503 | 0.460 | 0.391 | 0.406 | 0.448 | 0.422 | 0.292 | 0.331 | 0.261 | 0.282 | 0.432 | 0.445 | 0.745 | 0.426 | 0.527 | 0.430 | 1 | 3.53 |
| Leddam | 0.487 | 0.456 | 0.381 | 0.400 | 0.460 | 0.425 | 0.289 | 0.328 | 0.257 | 0.276 | 0.372 | 0.400 | 0.633 | 0.387 | 2.407 | 0.784 | 1 | 2.78 |
| RobustTSF | 0.508 | 0.475 | 0.450 | 0.448 | 0.456 | 0.421 | 0.294 | 0.340 | 0.287 | 0.302 | 0.380 | 0.401 | 0.884 | 0.474 | 0.574 | 0.430 | 2 | 4.59 |
| BARS-MLP | 0.465 | 0.442 | 0.383 | 0.402 | 0.409 | 0.402 | 0.281 | 0.320 | 0.260 | 0.282 | 0.368 | 0.415 | 0.654 | 0.380 | 0.554 | 0.427 | 20 | 2.05 |

all methods generally show better performance compared to last-position scenarios, RESAM maintains its competitive advantage.

This phenomenon provides strong empirical validation of our core hypothesis: existing time series forecasting models exhibit excessive dependence on recent observations, making them particularly vulnerable to perturbations affecting the most recent temporal points. The superior performance of RESAM across both last and random perturbation scenarios demonstrates its ability to maintain robust forecasting capabilities by effectively utilizing global temporal context rather than overemphasizing specific temporal regions.

# H  EFFICICENY

As demonstrated in Tbl. 11, the RESAM model achieves minimal forecasting error with low computational overhead. Compared to mainstream time series forecasting approaches, RESAM operates with significantly fewer parameters—a key advantage stemming from its parameter count being solely determined by the number of selected frequencies rather than scaling with input or prediction lengths. This architectural efficiency translates to substantially reduced GPU memory consumption during both training and inference phases. While the bisis coefficient solution in BARS introduces moderate computational overhead relative to pure time-domain linear models,

Table 11: Model efficiency comparison and their MSEs in forecasting on the ETTm1 dataset, with historical sequence length $H = 96$ and prediction length $F = 720$. Training Time denotes the average time required per epoch.

| Model | Parameters | GPU Memory (MB) | Training Time (s) | MSE |
|---|---|---|---|---|
| DLinear | 139.7K | 417 | 12.7 | 0.4688 |
| TimeMixer | 177.6K | 1779 | 64.5 | 0.4812 |
| CycleNet | 70.5K | 495 | 10.7 | 0.4603 |
| Crossformer | 171.8K | 2143 | 68.2 | 0.8870 |
| PatchTST | 1.37M | 553 | 20.7 | 0.4588 |
| MICN | 263.0K | 591 | 21.8 | 0.4578 |
| TimesNet | 666.0K | 3045 | 192.6 | 0.5450 |
| FEDformer | 16.8M | 3573 | 451.6 | 0.6584 |
| FreTS | 3.40M | 671 | 16.1 | 0.4893 |
| FiLM | 12.6M | 1301 | 114.5 | 0.4768 |
| RESAM | 129.2K | 509 | 38.8 | 0.4540 |

its overall training time still remains competitive. On ETTm1 dataset with $H = 96$ and $F = 720$, RESAM completes training 1.7-11.6× faster than structurally complex models like TimesMixer, TimesNet, and FEDformer, while maintaining slight time increase compared to simpler linear baselines. This efficiency-performance balance positions RESAM as a practical solution for resource-constrained forecasting scenarios requiring both accuracy and computational economy.

## I LIMITATION AND FUTURE WORK

There are some limitations and future works for RESAM. RESAM employs a channel-independent technique, which exhibits limited efficacy on time series with a high number of variates. Future research could explore the potential of channel clustering within the BARS module. Additionally, RESAM uses a fixed sampling rate for all time series inputs during training. We plan to explore dynamic sampling rates to enhance both forecasting accuracy and robustness in future research. Lastly, we only investigate perturbations applied to the single last time point. A more comprehensive analysis could be conducted by introducing perturbations to more recent time points within the historical sequence.

## J FULL RESULTS

Tbl. 12 presents the multivariate forecasting performance of all methods for various prediction lengths, using a consistent historical length of $H = 96$ to ensure a fair comparison.

For robustness evaluation, Tbl. 13 presents the forecasting performance when perturbations are applied to the last point of the historical input window, with a perturbation ratio of $\alpha = 3.0$. In this scenario, RESAM demonstrates superior performance in the majority of settings.

## K SHOWCASES

To assess the performance of different models, we perform a qualitative comparison by plotting a random dimension of the forecasting results from the test set of each dataset (Fig. 11, 12, 13, 14, 15, 16, 17). Among the various models, RESAM demonstrates superior performance, particularly when anomalies or large noise occur in the last few points of the historical window.

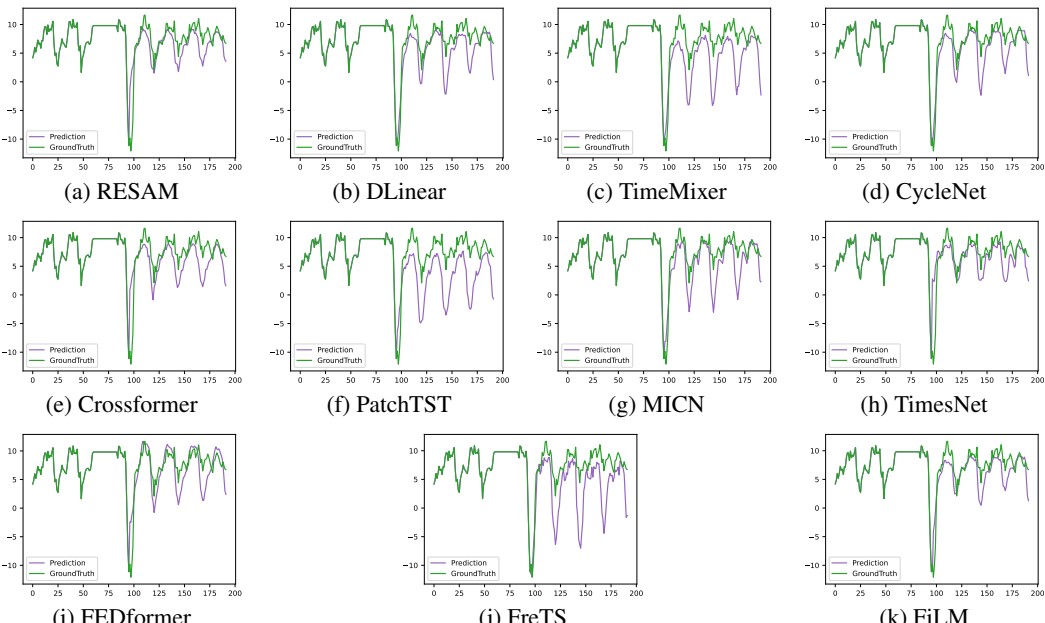

Figure 11: Prediction cases from ETTh1 by different models with historical length $H = 96$ and prediction length $F = 96$. Green lines are the ground truths and purple lines are the model predictions.

## L USAGE OF LLMS

We employed the GPT-4o (OpenAI, 2024) exclusively for the purpose of polishing and refining the writing. Specifically, the model was used to rephrase selected passages to enhance their academic

Table 12: Detailed multivariate time series forecasting results. Results for each prediction length $F \in \{96, 192, 336, 720\}$ are presented per row, with the average result on the bottom row per model.

| Models | F | ETTh1 | | ETTh2 | | ETTm1 | | ETTm2 | | Weather | | Exchange | | Traffic | | Solar | |
|---|---|---|---|---|---|---|---|---|---|---|---|---|---|---|---|---|---|
| | | MSE | MAE | MSE | MAE | MSE | MAE | MSE | MAE | MSE | MAE | MSE | MAE | MSE | MAE | MSE | MAE |
| DLinear | 96 | 0.379 | 0.386 | 0.293 | 0.343 | 0.331 | 0.350 | 0.183 | 0.258 | 0.208 | 0.233 | 0.098 | 0.221 | 0.688 | 0.364 | 0.280 | 0.289 |
| | 192 | 0.429 | 0.416 | 0.384 | 0.404 | 0.376 | 0.372 | 0.245 | 0.302 | 0.244 | 0.268 | 0.158 | 0.288 | 0.646 | 0.340 | 0.312 | 0.306 |
| | 336 | 0.472 | 0.441 | 0.460 | 0.458 | 0.407 | 0.394 | 0.307 | 0.348 | 0.287 | 0.306 | 0.263 | 0.382 | 0.649 | 0.343 | 0.356 | 0.322 |
| | 720 | 0.490 | 0.490 | 0.673 | 0.577 | 0.469 | 0.432 | 0.409 | 0.413 | 0.346 | 0.355 | 0.755 | 0.656 | 0.678 | 0.362 | 0.366 | 0.316 |
| | Avg | 0.443 | 0.433 | 0.452 | 0.446 | 0.396 | 0.387 | 0.286 | 0.330 | 0.271 | 0.290 | 0.318 | 0.387 | 0.665 | 0.352 | 0.328 | 0.308 |
| TimeMixer | 96 | 0.382 | 0.389 | 0.287 | 0.331 | 0.326 | 0.352 | 0.175 | 0.253 | 0.167 | 0.201 | 0.095 | 0.214 | 0.469 | 0.289 | 0.274 | 0.290 |
| | 192 | 0.441 | 0.422 | 0.374 | 0.390 | 0.374 | 0.372 | 0.239 | 0.296 | 0.215 | 0.246 | 0.186 | 0.307 | 0.485 | 0.299 | 0.378 | 0.343 |
| | 336 | 0.496 | 0.460 | 0.413 | 0.421 | 0.411 | 0.393 | 0.300 | 0.334 | 0.276 | 0.290 | 0.329 | 0.414 | 0.496 | 0.304 | 0.356 | 0.335 |
| | 720 | 0.523 | 0.484 | 0.420 | 0.437 | 0.481 | 0.435 | 0.399 | 0.392 | 0.358 | 0.344 | 0.913 | 0.723 | 0.532 | 0.325 | 0.327 | 0.317 |
| | Avg | 0.461 | 0.439 | 0.374 | 0.395 | 0.398 | 0.388 | 0.278 | 0.319 | 0.254 | 0.270 | 0.381 | 0.414 | 0.495 | 0.304 | 0.334 | 0.321 |
| CycleNet | 96 | 0.374 | 0.382 | 0.287 | 0.332 | 0.319 | 0.346 | 0.167 | 0.245 | 0.176 | 0.212 | 0.084 | 0.202 | 0.471 | 0.283 | 0.258 | 0.253 |
| | 192 | 0.422 | 0.410 | 0.373 | 0.385 | 0.367 | 0.369 | 0.233 | 0.288 | 0.225 | 0.255 | 0.190 | 0.308 | 0.480 | 0.286 | 0.282 | 0.262 |
| | 336 | 0.461 | 0.430 | 0.416 | 0.422 | 0.397 | 0.391 | 0.294 | 0.328 | 0.279 | 0.294 | 0.372 | 0.441 | 0.491 | 0.292 | 0.309 | 0.273 |
| | 720 | 0.461 | 0.449 | 0.424 | 0.438 | 0.460 | 0.425 | 0.396 | 0.387 | 0.352 | 0.344 | 0.915 | 0.722 | 0.516 | 0.308 | 0.319 | 0.277 |
| | Avg | 0.429 | 0.418 | 0.375 | 0.394 | 0.386 | 0.383 | 0.273 | 0.312 | 0.258 | 0.276 | 0.390 | 0.419 | 0.489 | 0.292 | 0.292 | 0.266 |
| Crossformer | 96 | 0.444 | 0.443 | 0.644 | 0.547 | 0.367 | 0.389 | 0.228 | 0.315 | 0.150 | 0.189 | 0.172 | 0.304 | 0.519 | 0.261 | 0.196 | 0.185 |
| | 192 | 0.455 | 0.450 | 0.758 | 0.600 | 0.391 | 0.399 | 0.301 | 0.376 | 0.201 | 0.246 | 0.373 | 0.463 | 0.611 | 0.325 | 0.222 | 0.207 |
| | 336 | 0.522 | 0.489 | 1.039 | 0.713 | 0.428 | 0.420 | 0.631 | 0.545 | 0.274 | 0.301 | 0.699 | 0.665 | 0.581 | 0.270 | 0.258 | 0.251 |
| | 720 | 0.651 | 0.610 | 1.464 | 0.892 | 0.748 | 0.571 | 1.057 | 0.710 | 0.365 | 0.365 | 1.182 | 0.907 | 0.661 | 0.345 | 0.229 | 0.227 |
| | Avg | 0.518 | 0.498 | 0.976 | 0.688 | 0.484 | 0.445 | 0.554 | 0.486 | 0.247 | 0.275 | 0.606 | 0.585 | 0.593 | 0.300 | 0.226 | 0.218 |
| PatchTST | 96 | 0.375 | 0.387 | 0.284 | 0.329 | 0.310 | 0.340 | 0.174 | 0.251 | 0.174 | 0.210 | 0.084 | 0.200 | 0.473 | 0.283 | 0.211 | 0.243 |
| | 192 | 0.423 | 0.416 | 0.362 | 0.380 | 0.363 | 0.372 | 0.239 | 0.294 | 0.221 | 0.249 | 0.184 | 0.304 | 0.476 | 0.284 | 0.269 | 0.316 |
| | 336 | 0.467 | 0.436 | 0.408 | 0.416 | 0.400 | 0.393 | 0.299 | 0.333 | 0.276 | 0.289 | 0.324 | 0.412 | 0.497 | 0.292 | 0.324 | 0.337 |
| | 720 | 0.461 | 0.455 | 0.414 | 0.432 | 0.459 | 0.429 | 0.400 | 0.391 | 0.352 | 0.339 | 0.901 | 0.719 | 0.533 | 0.309 | 0.283 | 0.322 |
| | Avg | 0.431 | 0.424 | 0.367 | 0.389 | 0.383 | 0.383 | 0.278 | 0.318 | 0.256 | 0.272 | 0.373 | 0.409 | 0.495 | 0.292 | 0.272 | 0.305 |
| MICN | 96 | 0.388 | 0.402 | 0.298 | 0.351 | 0.307 | 0.341 | 0.175 | 0.263 | 0.187 | 0.225 | 0.089 | 0.209 | 0.575 | 0.296 | 0.196 | 0.229 |
| | 192 | 0.413 | 0.423 | 0.379 | 0.402 | 0.362 | 0.373 | 0.242 | 0.310 | 0.230 | 0.262 | 0.151 | 0.280 | 0.595 | 0.300 | 0.244 | 0.249 |
| | 336 | 0.448 | 0.444 | 0.464 | 0.460 | 0.396 | 0.399 | 0.302 | 0.349 | 0.262 | 0.289 | 0.246 | 0.374 | 0.622 | 0.311 | 0.321 | 0.293 |
| | 720 | 0.493 | 0.494 | 0.710 | 0.589 | 0.458 | 0.439 | 0.419 | 0.426 | 0.312 | 0.322 | 0.751 | 0.654 | 0.684 | 0.336 | 0.372 | 0.347 |
| | Avg | 0.435 | 0.441 | 0.463 | 0.450 | 0.381 | 0.388 | 0.285 | 0.337 | 0.248 | 0.274 | 0.309 | 0.379 | 0.619 | 0.311 | 0.283 | 0.279 |
| TimesNet | 96 | 0.419 | 0.420 | 0.319 | 0.356 | 0.336 | 0.364 | 0.180 | 0.256 | 0.162 | 0.204 | 0.108 | 0.234 | 0.662 | 0.338 | 0.292 | 0.295 |
| | 192 | 0.451 | 0.437 | 0.412 | 0.411 | 0.376 | 0.387 | 0.244 | 0.300 | 0.223 | 0.256 | 0.195 | 0.320 | 0.669 | 0.347 | 0.385 | 0.357 |
| | 336 | 0.518 | 0.481 | 0.456 | 0.441 | 0.404 | 0.405 | 0.313 | 0.341 | 0.283 | 0.301 | 0.379 | 0.442 | 0.674 | 0.348 | 0.359 | 0.322 |
| | 720 | 0.572 | 0.520 | 0.480 | 0.466 | 0.506 | 0.457 | 0.419 | 0.401 | 0.357 | 0.351 | 1.037 | 0.779 | 0.711 | 0.360 | 0.421 | 0.371 |
| | Avg | 0.490 | 0.464 | 0.417 | 0.419 | 0.406 | 0.403 | 0.289 | 0.325 | 0.256 | 0.278 | 0.430 | 0.444 | 0.679 | 0.348 | 0.364 | 0.336 |
| FEDformer | 96 | 0.377 | 0.404 | 0.328 | 0.370 | 0.450 | 0.462 | 0.227 | 0.317 | 0.210 | 0.272 | 0.124 | 0.255 | 0.637 | 0.376 | 0.331 | 0.409 |
| | 192 | 0.427 | 0.434 | 0.407 | 0.419 | 0.562 | 0.518 | 0.295 | 0.355 | 0.258 | 0.307 | 0.255 | 0.361 | 0.637 | 0.369 | 0.442 | 0.502 |
| | 336 | 0.463 | 0.458 | 0.482 | 0.476 | 0.607 | 0.536 | 0.375 | 0.406 | 0.347 | 0.375 | 0.477 | 0.505 | 0.648 | 0.373 | 0.995 | 0.826 |
| | 720 | 0.492 | 0.486 | 0.452 | 0.460 | 0.658 | 0.566 | 0.468 | 0.453 | 0.410 | 0.401 | 1.261 | 0.865 | 0.675 | 0.388 | 0.424 | 0.460 |
| | Avg | 0.439 | 0.446 | 0.417 | 0.431 | 0.569 | 0.520 | 0.341 | 0.383 | 0.306 | 0.339 | 0.529 | 0.497 | 0.649 | 0.376 | 0.548 | 0.549 |
| FreTS | 96 | 0.408 | 0.401 | 0.308 | 0.347 | 0.330 | 0.353 | 0.183 | 0.261 | 0.167 | 0.204 | 0.086 | 0.207 | 0.500 | 0.297 | 0.230 | 0.247 |
| | 192 | 0.460 | 0.431 | 0.388 | 0.399 | 0.407 | 0.394 | 0.258 | 0.312 | 0.207 | 0.243 | 0.164 | 0.295 | 0.498 | 0.296 | 0.260 | 0.263 |
| | 336 | 0.534 | 0.467 | 0.508 | 0.480 | 0.429 | 0.415 | 0.329 | 0.364 | 0.259 | 0.283 | 0.427 | 0.489 | 0.515 | 0.302 | 0.279 | 0.270 |
| | 720 | 0.643 | 0.551 | 0.961 | 0.651 | 0.493 | 0.453 | 0.429 | 0.423 | 0.329 | 0.336 | 0.899 | 0.710 | 0.562 | 0.323 | 0.282 | 0.265 |
| | Avg | 0.511 | 0.462 | 0.541 | 0.469 | 0.415 | 0.404 | 0.300 | 0.340 | 0.240 | 0.267 | 0.394 | 0.425 | 0.519 | 0.304 | 0.263 | 0.261 |
| FiLM | 96 | 0.382 | 0.386 | 0.296 | 0.340 | 0.334 | 0.350 | 0.183 | 0.258 | 0.208 | 0.231 | 0.115 | 0.248 | 0.811 | 0.477 | 0.298 | 0.291 |
| | 192 | 0.435 | 0.417 | 0.385 | 0.395 | 0.380 | 0.373 | 0.247 | 0.299 | 0.250 | 0.266 | 0.180 | 0.302 | 1.281 | 0.717 | 0.337 | 0.324 |
| | 336 | 0.478 | 0.439 | 0.421 | 0.426 | 0.412 | 0.394 | 0.308 | 0.338 | 0.299 | 0.302 | 0.321 | 0.409 | 1.386 | 0.765 | 0.383 | 0.368 |
| | 720 | 0.481 | 0.468 | 0.427 | 0.440 | 0.477 | 0.430 | 0.408 | 0.394 | 0.370 | 0.348 | 0.886 | 0.711 | 1.455 | 0.789 | 0.491 | 0.466 |
| | Avg | 0.444 | 0.427 | 0.382 | 0.400 | 0.401 | 0.387 | 0.286 | 0.322 | 0.282 | 0.287 | 0.376 | 0.417 | 1.233 | 0.687 | 0.377 | 0.362 |
| RESAM | 96 | 0.377 | 0.389 | 0.286 | 0.332 | 0.312 | 0.339 | 0.164 | 0.242 | 0.163 | 0.202 | 0.083 | 0.200 | 0.488 | 0.290 | 0.196 | 0.220 |
| | 192 | 0.425 | 0.417 | 0.365 | 0.382 | 0.360 | 0.366 | 0.229 | 0.286 | 0.212 | 0.247 | 0.192 | 0.313 | 0.501 | 0.298 | 0.232 | 0.240 |
| | 336 | 0.462 | 0.433 | 0.406 | 0.418 | 0.392 | 0.388 | 0.287 | 0.323 | 0.268 | 0.286 | 0.343 | 0.422 | 0.514 | 0.308 | 0.273 | 0.263 |
| | 720 | 0.462 | 0.453 | 0.419 | 0.435 | 0.454 | 0.422 | 0.387 | 0.384 | 0.348 | 0.340 | 0.813 | 0.679 | 0.647 | 0.334 | 0.283 | 0.265 |
| | Avg | 0.432 | 0.423 | 0.369 | 0.392 | 0.379 | 0.379 | 0.267 | 0.309 | 0.248 | 0.269 | 0.358 | 0.403 | 0.537 | 0.308 | 0.246 | 0.247 |

Table 13: Detailed robustness evaluation under last-point perturbation ratio $\alpha = 3.0$. Results for each prediction length $F \in \{96, 192, 336, 720\}$ are presented per row, with the average result on the bottom row per model.

| Models | F | ETTh1 | | ETTh2 | | ETTm1 | | ETTm2 | | Weather | | Exchange | | Traffic | | Solar | |
|---|---|---|---|---|---|---|---|---|---|---|---|---|---|---|---|---|---|
| | | MSE | MAE | MSE | MAE | MSE | MAE | MSE | MAE | MSE | MAE | MSE | MAE | MSE | MAE | MSE | MAE |
| DLinear | 96 | 0.606 | 0.483 | 0.345 | 0.384 | 0.717 | 0.492 | 0.222 | 0.296 | 0.384 | 0.326 | 0.312 | 0.405 | 0.981 | 0.450 | 3.883 | 0.945 |
| | 192 | 0.567 | 0.484 | 0.415 | 0.427 | 0.618 | 0.466 | 0.311 | 0.353 | 0.347 | 0.327 | **0.160** | **0.291** | 0.914 | 0.416 | 3.535 | 0.923 |
| | 336 | 0.569 | 0.494 | 0.461 | 0.460 | 0.601 | 0.474 | 0.353 | 0.383 | 0.375 | 0.352 | 0.265 | **0.385** | 0.846 | 0.400 | 2.964 | 0.830 |
| | 720 | 0.521 | 0.510 | 0.570 | 0.535 | 0.556 | 0.475 | 0.436 | 0.433 | 0.379 | 0.375 | **0.755** | **0.656** | 0.865 | 0.420 | 2.318 | 0.704 |
| | Avg | 0.566 | 0.493 | 0.448 | 0.451 | 0.623 | 0.477 | 0.331 | 0.366 | 0.371 | 0.345 | 0.373 | 0.434 | 0.902 | 0.422 | 3.175 | 0.851 |
| TimeMixer | 96 | 0.574 | 0.480 | 0.362 | 0.386 | 0.528 | 0.445 | 0.219 | 0.294 | 0.261 | 0.288 | 0.217 | 0.332 | 0.569 | 0.342 | 0.544 | 0.434 |
| | 192 | 0.527 | 0.470 | 0.394 | 0.406 | 0.503 | 0.434 | 0.288 | 0.338 | 0.256 | 0.286 | 0.218 | 0.340 | 0.571 | 0.348 | 0.510 | 0.437 |
| | 336 | 0.532 | 0.482 | 0.440 | 0.440 | 0.548 | 0.456 | 0.348 | 0.371 | 0.308 | 0.317 | 0.376 | 0.449 | 0.558 | 0.340 | 0.649 | 0.481 |
| | 720 | 0.574 | 0.511 | 0.433 | 0.446 | 0.555 | 0.472 | 0.416 | 0.405 | 0.365 | 0.353 | 0.944 | 0.737 | **0.575** | 0.356 | 0.575 | 0.446 |
| | Avg | 0.552 | 0.486 | 0.407 | 0.419 | 0.534 | 0.452 | 0.318 | 0.352 | 0.298 | 0.311 | 0.439 | 0.465 | 0.568 | 0.346 | 0.569 | 0.449 |
| CycleNet | 96 | 0.607 | 0.484 | 0.381 | 0.401 | 0.719 | 0.500 | 0.241 | 0.311 | 0.330 | 0.302 | 0.104 | 0.232 | 0.608 | 0.344 | 0.968 | 0.493 |
| | 192 | 0.566 | 0.479 | 0.435 | 0.430 | 0.602 | 0.466 | 0.284 | 0.334 | 0.328 | 0.316 | 0.241 | 0.357 | 0.592 | 0.337 | 0.689 | 0.418 |
| | 336 | 0.559 | 0.481 | 0.461 | 0.453 | 0.575 | 0.469 | 0.332 | 0.361 | 0.347 | 0.336 | 0.422 | 0.477 | 0.570 | 0.337 | 0.553 | 0.372 |
| | 720 | 0.530 | 0.486 | 0.461 | 0.461 | 0.580 | 0.481 | 0.421 | 0.408 | 0.391 | 0.368 | 0.950 | 0.740 | 0.580 | 0.344 | 0.445 | 0.335 |
| | Avg | 0.565 | 0.483 | 0.434 | 0.436 | 0.619 | 0.479 | 0.320 | 0.353 | 0.349 | 0.331 | 0.429 | 0.452 | 0.588 | 0.340 | 0.664 | 0.405 |
| Crossformer | 96 | 0.443 | 0.445 | 0.654 | 0.553 | 0.442 | 0.440 | 0.237 | 0.321 | 0.218 | 0.264 | 0.188 | 0.325 | 0.561 | **0.292** | **0.297** | **0.271** |
| | 192 | 0.470 | 0.462 | 0.761 | 0.601 | 0.421 | 0.422 | 0.353 | 0.405 | 0.230 | 0.270 | 0.380 | 0.469 | 0.593 | **0.282** | **0.271** | **0.265** |
| | 336 | 0.518 | 0.488 | 1.041 | 0.714 | 0.445 | 0.433 | 0.634 | 0.546 | 0.284 | 0.314 | 0.700 | 0.664 | 0.582 | **0.279** | 0.362 | 0.313 |
| | 720 | 0.665 | 0.619 | 1.465 | 0.892 | 0.748 | 0.572 | 1.060 | 0.711 | 0.362 | 0.364 | 1.181 | 0.906 | 0.635 | **0.304** | 0.379 | 0.302 |
| | Avg | 0.524 | 0.503 | 0.980 | 0.690 | 0.514 | 0.467 | 0.571 | 0.496 | 0.274 | 0.303 | 0.612 | 0.591 | 0.593 | **0.289** | 0.327 | 0.288 |
| PatchTST | 96 | 0.596 | 0.491 | 0.343 | 0.377 | 0.839 | 0.536 | 0.259 | 0.322 | 0.234 | 0.266 | 0.209 | 0.333 | 0.637 | 0.368 | 0.956 | 0.555 |
| | 192 | 0.531 | 0.473 | 0.398 | 0.406 | 0.624 | 0.484 | 0.306 | 0.349 | 0.278 | 0.299 | 0.267 | 0.379 | 0.594 | 0.350 | 0.649 | 0.504 |
| | 336 | 0.542 | 0.478 | 0.429 | 0.432 | 0.508 | 0.450 | 0.389 | 0.393 | 0.323 | 0.329 | 0.402 | 0.473 | 0.566 | 0.339 | 0.590 | 0.465 |
| | 720 | 0.547 | 0.500 | 0.430 | 0.444 | 0.608 | 0.492 | 0.428 | 0.412 | 0.390 | 0.370 | 0.993 | 0.756 | 0.607 | 0.354 | 0.704 | 0.503 |
| | Avg | 0.554 | 0.486 | 0.400 | 0.415 | 0.645 | 0.491 | 0.346 | 0.369 | 0.306 | 0.316 | 0.468 | 0.485 | 0.601 | 0.353 | 0.725 | 0.507 |
| MICN | 96 | 0.470 | 0.454 | 0.318 | 0.368 | 0.512 | 0.442 | 0.220 | 0.306 | 0.269 | 0.291 | **0.104** | **0.231** | 0.595 | 0.314 | 0.300 | 0.314 |
| | 192 | 0.451 | 0.448 | 0.394 | 0.411 | 0.461 | 0.426 | 0.271 | 0.336 | 0.294 | 0.311 | 0.178 | 0.312 | 0.599 | 0.307 | 0.339 | 0.323 |
| | 336 | **0.474** | 0.463 | 0.471 | 0.465 | 0.454 | 0.431 | 0.315 | 0.361 | 0.301 | 0.319 | 0.259 | 0.387 | 0.616 | 0.313 | 0.388 | 0.351 |
| | 720 | 0.501 | 0.503 | 0.712 | 0.592 | 0.510 | 0.466 | 0.429 | 0.435 | 0.316 | 0.331 | 0.759 | 0.661 | 0.680 | 0.338 | 0.409 | 0.393 |
| | Avg | 0.474 | 0.467 | 0.474 | 0.459 | 0.484 | 0.441 | 0.309 | 0.360 | 0.295 | 0.313 | **0.325** | **0.398** | 0.623 | 0.318 | 0.359 | 0.345 |
| TimesNet | 96 | 0.419 | 0.421 | 0.313 | 0.353 | **0.355** | 0.378 | 0.191 | 0.268 | **0.171** | **0.213** | 0.112 | 0.239 | 0.684 | 0.351 | 0.310 | 0.312 |
| | 192 | 0.465 | 0.445 | 0.391 | 0.398 | **0.385** | **0.393** | 0.248 | 0.303 | **0.229** | **0.262** | 0.201 | 0.325 | 0.689 | 0.359 | 0.408 | 0.378 |
| | 336 | 0.503 | 0.468 | 0.444 | 0.438 | **0.412** | **0.410** | 0.322 | 0.348 | 0.288 | 0.306 | 0.383 | 0.445 | 0.697 | 0.361 | 0.375 | 0.339 |
| | 720 | 0.500 | 0.480 | 0.436 | 0.445 | 0.521 | 0.464 | 0.422 | 0.403 | 0.361 | 0.354 | 1.045 | 0.783 | 0.732 | 0.370 | 0.434 | 0.385 |
| | Avg | 0.472 | 0.453 | 0.396 | 0.409 | 0.418 | 0.411 | 0.296 | 0.331 | **0.262** | **0.284** | 0.435 | 0.448 | 0.700 | 0.360 | 0.382 | 0.354 |
| FEDformer | 96 | **0.392** | **0.415** | 0.329 | 0.371 | 0.646 | 0.536 | 0.251 | 0.336 | 0.226 | 0.287 | 0.125 | 0.256 | 0.653 | 0.388 | 0.432 | 0.473 |
| | 192 | **0.449** | 0.448 | 0.408 | 0.420 | 0.894 | 0.613 | 0.299 | 0.358 | 0.272 | 0.321 | 0.255 | 0.361 | 0.659 | 0.387 | 0.649 | 0.602 |
| | 336 | 0.486 | 0.470 | 0.444 | 0.451 | 0.764 | 0.593 | 0.375 | 0.407 | 0.345 | 0.372 | 0.477 | 0.506 | 0.655 | 0.380 | 0.969 | 0.803 |
| | 720 | 0.540 | 0.503 | 0.451 | 0.461 | 0.853 | 0.622 | 0.468 | 0.453 | 0.414 | 0.405 | 1.261 | 0.865 | 0.695 | 0.402 | 0.577 | 0.549 |
| | Avg | 0.467 | 0.459 | 0.408 | 0.426 | 0.789 | 0.591 | 0.348 | 0.388 | 0.314 | 0.346 | 0.530 | 0.497 | 0.666 | 0.389 | 0.657 | 0.607 |
| FreTS | 96 | 0.733 | 0.513 | 0.420 | 0.416 | 0.800 | 0.522 | 0.389 | 0.384 | 0.280 | 0.318 | 0.131 | 0.264 | 0.883 | 0.428 | 2.009 | 0.650 |
| | 192 | 0.667 | 0.517 | 0.440 | 0.430 | 0.732 | 0.506 | 0.377 | 0.388 | 0.284 | 0.317 | 0.218 | 0.347 | 0.868 | 0.421 | 1.025 | 0.557 |
| | 336 | 0.637 | 0.520 | 0.467 | 0.459 | 0.764 | 0.528 | 0.382 | 0.399 | 0.317 | 0.339 | 0.436 | 0.496 | 0.692 | 0.377 | 1.298 | 0.589 |
| | 720 | 0.672 | 0.577 | 0.977 | 0.664 | 0.683 | 0.522 | 0.466 | 0.446 | 0.359 | 0.368 | 0.908 | 0.715 | 0.725 | 0.393 | 1.146 | 0.548 |
| | Avg | 0.677 | 0.532 | 0.576 | 0.492 | 0.745 | 0.519 | 0.404 | 0.404 | 0.310 | 0.336 | 0.423 | 0.456 | 0.792 | 0.405 | 1.369 | 0.586 |
| FiLM | 96 | 0.472 | 0.431 | 0.307 | 0.350 | 0.609 | 0.463 | 0.238 | 0.306 | 0.247 | 0.262 | 0.115 | 0.248 | 0.905 | 0.521 | 1.932 | 0.747 |
| | 192 | 0.486 | 0.444 | 0.387 | 0.397 | 0.460 | 0.415 | 0.281 | 0.329 | 0.274 | 0.286 | 0.181 | 0.303 | 1.335 | 0.741 | 1.347 | 0.673 |
| | 336 | 0.510 | 0.458 | 0.425 | 0.430 | 0.467 | 0.425 | 0.320 | 0.349 | 0.314 | 0.315 | 0.322 | 0.409 | 1.413 | 0.775 | 0.820 | 0.569 |
| | 720 | 0.500 | 0.478 | 0.435 | 0.446 | 0.509 | 0.450 | 0.421 | 0.406 | 0.378 | 0.356 | 0.901 | 0.722 | 1.470 | 0.794 | 0.595 | 0.526 |
| | Avg | 0.492 | 0.453 | 0.389 | 0.406 | 0.511 | 0.438 | 0.315 | 0.348 | 0.303 | 0.305 | 0.380 | 0.421 | 1.281 | 0.708 | 1.173 | 0.629 |
| RESAM | 96 | 0.421 | 0.416 | **0.303** | **0.348** | 0.366 | **0.377** | **0.182** | **0.260** | 0.187 | 0.231 | 0.183 | 0.308 | **0.503** | 0.306 | 0.356 | 0.328 |
| | 192 | 0.453 | **0.436** | **0.374** | **0.390** | 0.401 | 0.393 | **0.240** | **0.296** | 0.229 | 0.265 | 0.238 | 0.350 | **0.510** | 0.309 | 0.315 | 0.299 |
| | 336 | 0.488 | **0.450** | **0.413** | **0.424** | 0.424 | 0.410 | **0.295** | **0.330** | 0.280 | 0.300 | 0.461 | 0.505 | **0.523** | 0.319 | 0.338 | 0.307 |
| | 720 | **0.485** | 0.467 | **0.425** | **0.441** | 0.473 | 0.435 | **0.394** | **0.389** | 0.355 | 0.347 | 0.803 | 0.677 | 0.664 | 0.344 | 0.349 | 0.297 |
| | Avg | **0.461** | **0.442** | 0.379 | 0.401 | 0.416 | 0.404 | **0.278** | **0.319** | 0.263 | 0.286 | 0.421 | 0.460 | **0.550** | 0.319 | 0.340 | 0.308 |

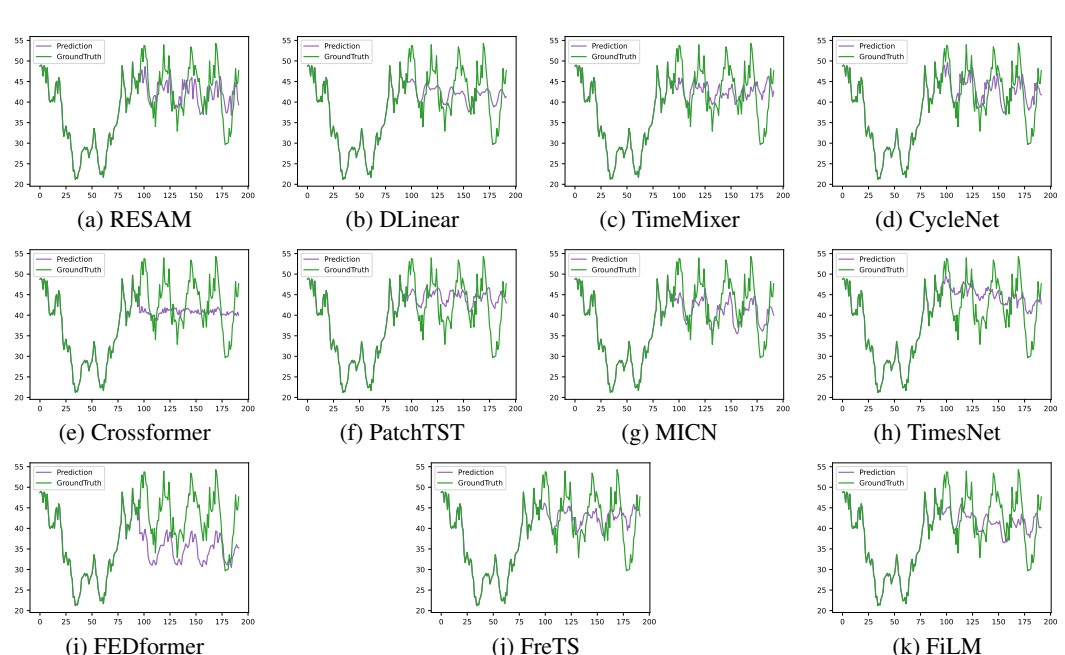

Figure 12: Prediction cases from ETTh2 by different models with historical length $H = 96$ and prediction length $F = 96$.

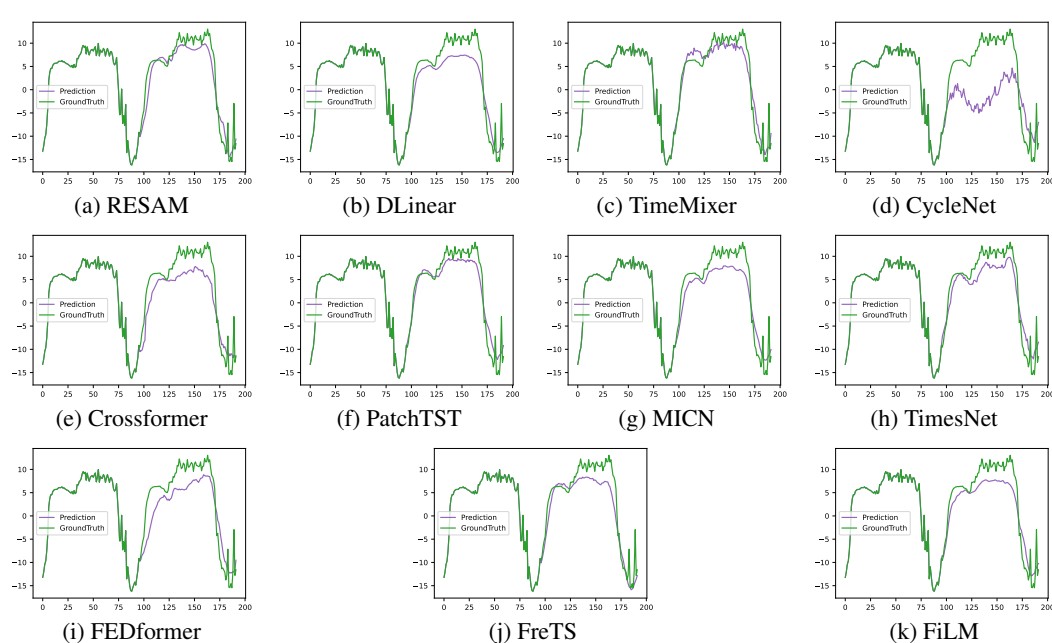

Figure 13: Prediction cases from ETTm1 by different models with historical length $H = 96$ and prediction length $F = 96$.

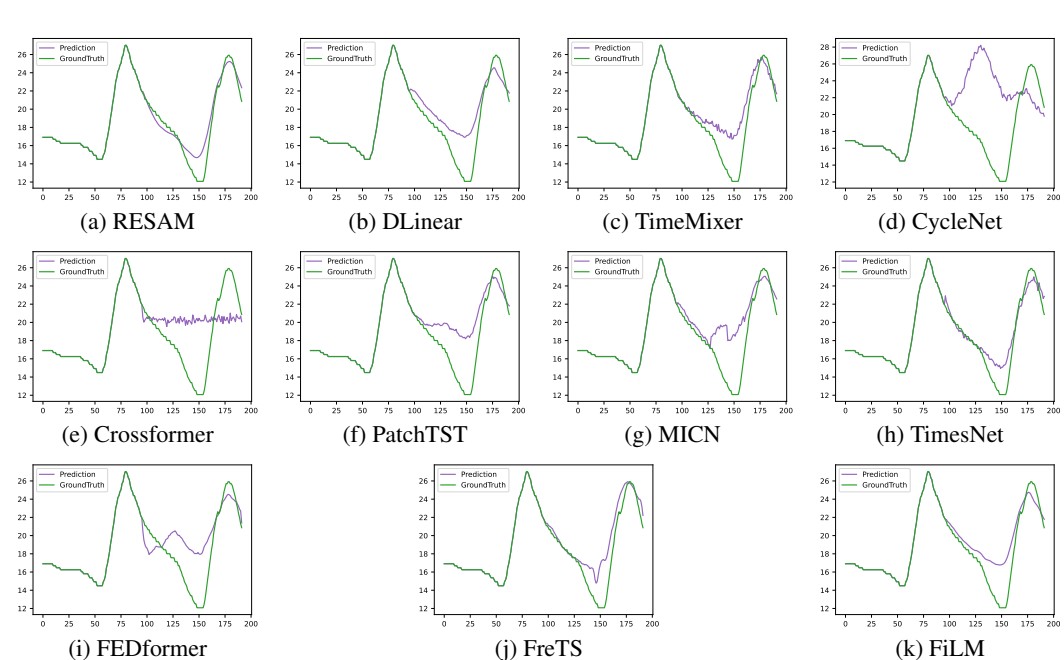

Figure 14: Prediction cases from ETTm2 by different models with historical length $H = 96$ and prediction length $F = 96$.

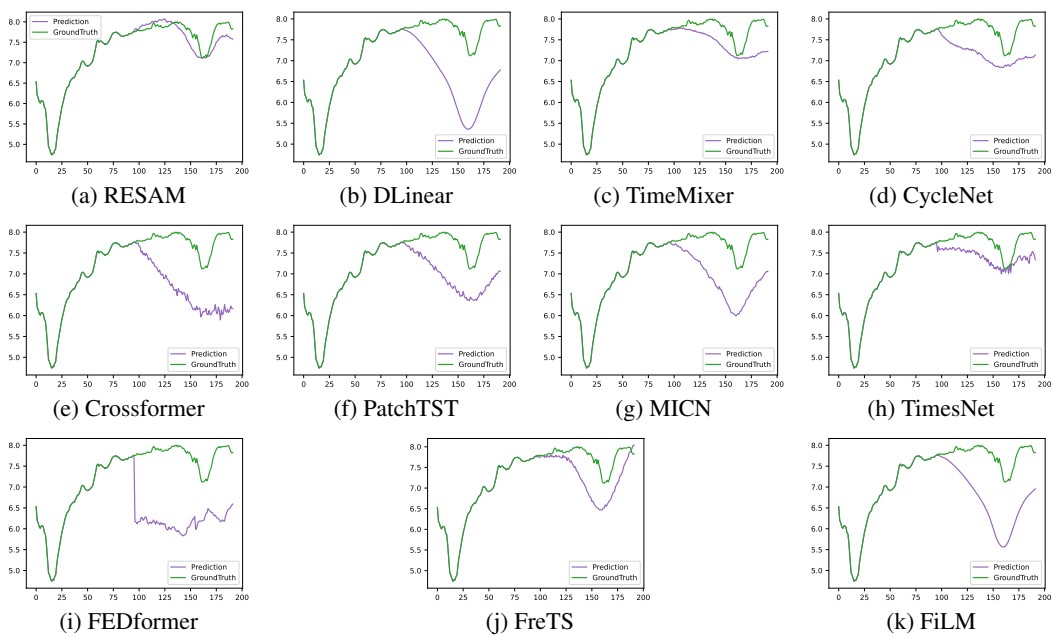

Figure 15: Prediction cases from Weather by different models with historical length $H = 96$ and prediction length $F = 96$.

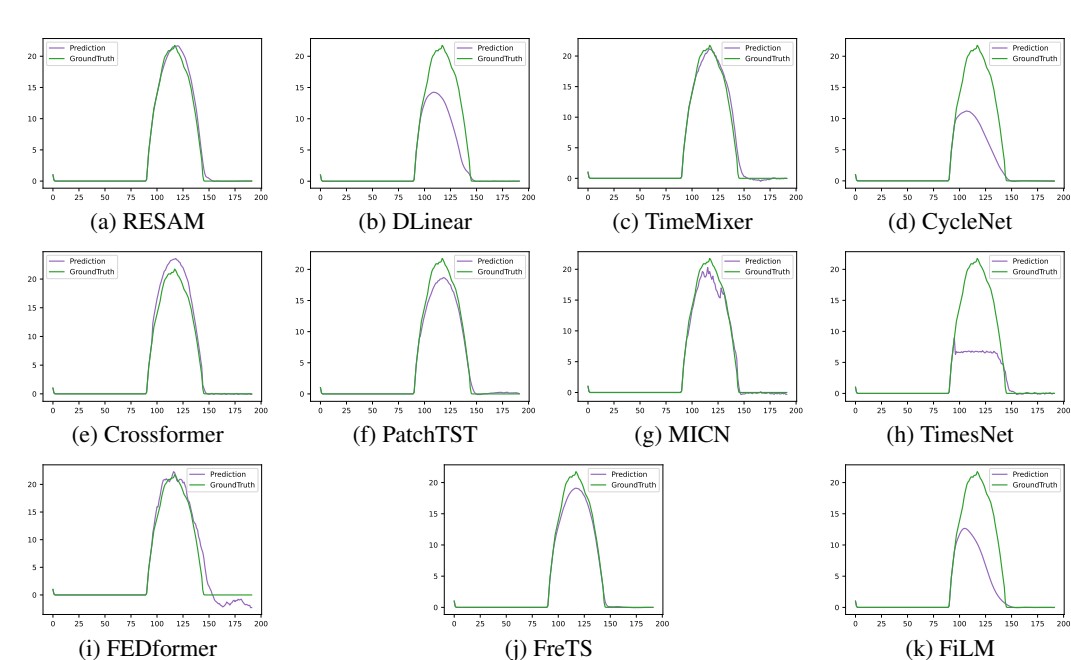

Figure 16: Prediction cases from Solar by different models with historical length $H = 96$ and prediction length $F = 96$.

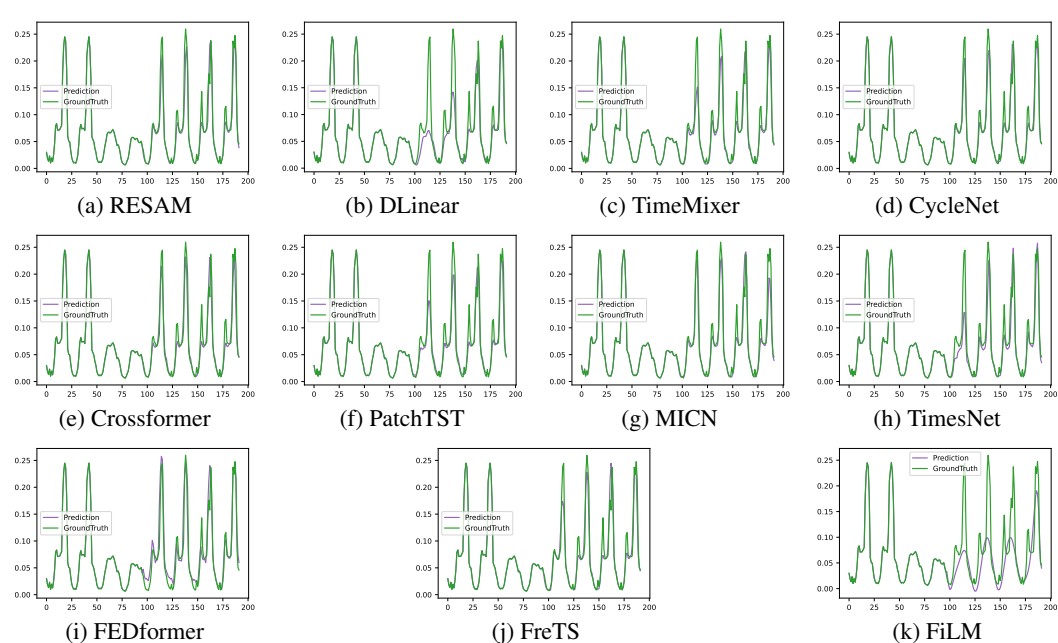

Figure 17: Prediction cases from Traffic by different models with historical length $H = 96$ and prediction length $F = 96$.

tone and clarity while strictly preserving the original technical meaning and content. We crafted and used the following prompt to guide this process:

> *Rewrite the following text in an academic paper in the area of computer science, which will be delimited by triple quotes, to be more professional and well-written while preserving the original meaning: """[input text]""" If the original expression of some words or sentences is good enough, there is no need to rewrite it. Try to express one meaning in one sentence, with enough information, and don't have too many long and difficult sentences. Provide only the rewritten text as your output, without any quotes or tags. Respond in the same language as the original text.*

All output generated by the model was carefully reviewed, edited, and verified.

