# OpenReview forum: "Robust Time Series Forecasting via Basis-Aligned Sampling in Decycled Residual Space"
_ICLR.cc/2026/Conference — Submitted to ICLR 2026_

### Official Review · Reviewer_YX6M · 2025-10-30

**Soundness:** 2
**Presentation:** 2
**Contribution:** 2
**Rating:** 2
**Confidence:** 4

**Summary:**

The paper addresses the issue that recent observations in time series have a significant impact on forecasting performance. To mitigate this, the paper proposes a new robust time series forecasting method, RESAM, aimed at improving prediction reliability. Specifically, RESAM uses a basis-aligned random sampling strategy to explore global context information while achieving a unified representation for irregularly sampled sequences. Additionally, RESAM incorporates a learnable periodicity extraction module to capture periodic patterns. Evaluations on multiple datasets show that it outperforms baseline models in terms of performance.

**Strengths:**

1. The paper clearly identifies the issue of the last points.
2. The architecture design is simple and efficient.

**Weaknesses:**

1. The experimental baseline models are not cutting-edge enough. It is recommended to include more recent, high-performing baseline methods. Additionally, the effectiveness of the method is not sufficiently demonstrated in the main experiments.
2. The paper seems more like an engineering-driven modular approach for performance improvement rather than an in-depth exploration of the issue, which significantly weakens its contribution.
3. The motivation behind the paper seems illogical. It is reasonable that the last points in a time series are important and that learning such biases is valid. It is also intuitive that perturbing these points would lead to a significant drop in performance.
4. The related work section is too broad and lacks adequate discussion of relevant research. In my knowledge, the importance of more recent observations is a widely accepted phenomenon, especially in fields like finance where "market efficiency" often dictates that recent data hold higher significance.

**Questions:**

See weaknesses.

**Details Of Ethics Concerns:**

NAN

---

> ### Author Response · Authors · 2025-11-27
>
> We sincerely thank Reviewer YX6M for their thoughtful comments and valuable feedback. We believe the reviewer may not fully understand our work.
>
> ### W1: Baselines not cutting-edge and insufficient demonstration of effectiveness
>
> **A1:** In our study, we selected baseline models that are widely recognized and cover diverse architectural paradigms to ensure a comprehensive comparison, as detailed in Section 4.1 and Appendix A. This approach allows for a fair evaluation of RESAM across different design philosophies. To further enhance the baseline coverage, we have now included additional models in the revised manuscript, such as RobustTSF [1], iTransformer [2], Leddam [3], and TimeKAN [4].
>
> The full results are updated in Table 1 (overall forecasting performance) and Table 2 (robustness) in the revised manuscript. Here, we present the results of added baselines and our method. The results consistently show that RESAM achieves superior performance in robustness against point-wise perturbations. Moreover, despite using a simple dual-layer MLP backbone, RESAM maintains competitive forecasting accuracy under normal (non-perturbed) conditions, as evidenced by its performance in Table 1.
>
> **Table a: Performance comparison of newly added baselines for multivariate time series forecasting task. The results are averaged from all prediction lengths of $F \in \{96, 192, 336, 720\}$. MSE/MAE are reported.**
>
> | Models | ETTh1 | ETTh2 | ETTm1 | ETTm2 | Weather | Exchange | Traffic | Solar |
> |--------|----------------|----------------|----------------|----------------|------------------|-------------------|------------------|----------------|
> | iTransformer | 0.448 / 0.439 | 0.381 / 0.401 | 0.394 / 0.389 | 0.281 / 0.321 | 0.254 / 0.271 | 0.365 / 0.406 | **0.450** / 0.282 | 0.249 / 0.245 |
> | TimeKAN | 0.445 / 0.427 | 0.378 / 0.397 | 0.388 / 0.384 | 0.281 / 0.321 | 0.249 / 0.269 | 0.394 / 0.422 | 0.633 / 0.349 | 0.321 / 0.291 |
> | Leddam | 0.437 / 0.425 | 0.371 / **0.391** | 0.390 / 0.383 | 0.277 / 0.317 | **0.243** / 0.272 | 0.362 / 0.403 | 0.491 / **0.278** | **0.220** / 0.256 |
> | RobustTSF | 0.446 / 0.438 | 0.435 / 0.438 | 0.395 / 0.390 | 0.284 / 0.332 | 0.270 / 0.293 | 0.374 / **0.396** | 0.639 / 0.359 | 0.342 / 0.343 |
> | **Ours** | **0.432** / **0.423** | **0.369** / 0.392 | **0.379** / **0.379** | **0.267** / **0.309** | 0.248 / 0.269 | **0.358** / 0.403 | 0.537 / 0.308 | 0.243 / **0.244** |
>
> **Table b: Performance comparison of newly added baselines for robustness evaluation under last-point perturbation ratio $\alpha = 3.0$. The results are averaged from all prediction lengths of $F \in \{96, 192, 336, 720\}$. MSE/MAE are reported.**
>
> | Models | ETTh1 | ETTh2 | ETTm1 | ETTm2 | Weather | Exchange | Traffic | Solar |
> |--------|----------------|----------------|----------------|----------------|------------------|-------------------|------------------|----------------|
> | iTransformer | 0.500 / 0.470 | 0.404 / 0.418 | 0.575 / 0.475 | 0.327 / 0.359 | 0.317 / 0.322 | **0.410** / 0.445 | 0.583 / 0.365 | 0.754 / 0.470 |
> | TimeKAN | 0.501 / 0.461 | 0.404 / 0.418 | 0.521 / 0.450 | 0.323 / 0.357 | 0.286 / 0.302 | 0.451 / 0.467 | 0.678 / 0.379 | 0.803 / 0.482 |
> | Leddam | 0.505 / 0.467 | 0.405 / 0.418 | 0.585 / 0.471 | 0.350 / 0.372 | 0.340 / 0.347 | 0.441 / 0.461 | 0.625 / 0.351 | 2.340 / 0.761 |
> | RobustTSF | 0.608 / 0.510 | 0.503 / 0.483 | 0.739 / 0.509 | 0.362 / 0.389 | 0.471 / 0.382 | 0.411 / **0.435** | 0.926 / 0.450 | 2.071 / 0.761 |
> | **Ours** | **0.461** / **0.442** | **0.379** / **0.401** | **0.416** / **0.404** | **0.278** / **0.319** | **0.263** / **0.286** | 0.421 / 0.460 | **0.550** / **0.319** | **0.340** / **0.308** |
>
> ### W2: Method seems like an engineering-driven modular approach rather than in-depth exploration
>
> **A2:** We **disagree** with this characterization. Our work presents a holistic research journey, from problem identification to solution design and validation, which constitutes a deep exploration of the issue. Specifically:
>
> - **Problem Identification**: We first systematically uncovered the vulnerability of existing models to point-wise perturbations on recent points through empirical analysis. As shown in Section 2.3, gradient-based importance scoring reveals that models excessively rely on the most recent points.
> - **Formalization and Solution**: We proposed RESAM as an integrated framework, not merely a modular combination. It includes a novel BARS strategy to enforce global context utilization, a LPE module to handle periodic components robustly, and a two-stage training protocol to ensure stability.
> - **Validation and Interpretation**: We conducted extensive experiments and provided global importance analysis (e.g., Figure 3 and Figure 7) to explain how RESAM achieves balanced temporal importance.

---

> ### Author Response · Authors · 2025-11-27
>
> ### W3: Motivation seems illogical
>
> **A3:** We understand the reviewer's perspective that recent points are intuitively important for forecasting, and learning such biases is valid. However, our motivation is **not** to negate the importance of recent points but to **address the excessive and unbalanced reliance** on them, which leads to fragility. For example, as demonstrated in Figure 1 of the manuscript, a single anomalous point at the end of the historical window can cause a dramatic prediction failure in existing models, which is undesirable in practical applications where data noise is common.
>
> Our research reveals that current models assign disproportionately high importance to the last points (Figure 3), often dozens of times higher than other points. This over-dependence undermines the use of global context and reduces robustness. RESAM aims to balance this by encouraging the model to utilize the entire historical context. This approach is logical for real-world scenarios where data quality cannot be guaranteed.
>
> ### W4: Related work is too broad and lacks discussion of relevant research
>
> A4: We agree that the importance of recent observations is a recognized phenomenon in fields like finance. However, while the importance of recent points is acknowledged, **the extent of this importance and its impact on robustness have not been systematically studied** prior to our work. Our contribution lies in quantitatively demonstrating that existing models over-rely on recent points and proposing a solution to mitigate the resulting vulnerability.
>
> ### References
>
> [1] RobustTSF: Towards Theory and Design of Robust Time Series Forecasting with Anomalies. ICLR 2024.
>
> [2] iTransformer: Inverted Transformers Are Effective for Time Series Forecasting. ICLR 2024.
>
> [3] Revitalizing multivariate time series forecasting: Learnable decomposition with inter-series dependencies and intra-series variations modeling. ICML 2024.
>
> [4] TimeKAN: KAN-based Frequency Decomposition Learning Architecture for Long-term Time Series Forecasting. ICLR 2025.

---

### Official Review · Reviewer_3Xag · 2025-10-31

**Soundness:** 2
**Presentation:** 2
**Contribution:** 2
**Rating:** 4
**Confidence:** 3

**Summary:**

This paper proposes a method that leverages a basis-aligned randomized sampling strategy and a learnable periodicity extraction module with a two-stage training protocol to forecast time series robustly.

**Strengths:**

1. Easy to follow and understand.
2. Well-motivated

**Weaknesses:**

1. There is a lack of a sufficient literature review. In the introduction part, the authors should discuss on the existing works and the reason why they cannot solve the challenges.
2. The baselines are not comprehensive enough to claim a SOTA performance. More baselines, such as iTransformer, Time-LLM, and GPT4TS, should be compared.
3. There is no experiment on short-term forecasting, limiting the application scope.
4. The overall performance increase is not significant and robust. It seems that only on ETTm1 and ETTm2, the proposed method shows clear advantages. Besides, on robustness evaluation, RESAM is worse than Crossformer and TimesNet according to the MSE increase.

**Questions:**

See Weakness.

---

> ### Author Response · Authors · 2025-11-27
>
> We sincerely thank Reviewer 3Xag for the valuable comments and suggestions. We have carefully revised our manuscript to address the concerns raised. Below we provide a point-by-point response.
>
> ### W1: Lack of Literature Review
>
> **A1:** While there are indeed excellent works on time series forecasting with anomalies or general robustness, they typically focus on different types of problems, such as common anomalies, or distribution shifts. To the best of our knowledge, no prior work has specifically targeted and systematically evaluated the vulnerability to **point-wise perturbations on recent points** in the historical context. Our work fills this gap. We have added a discussion in **Section Introduction (Line 62-73)** explaining why existing robustness methods are not sufficient for this specific vulnerability, as their objectives and mechanisms are different.
>
> We also add RobustTSF[1] (aiming for forecasting with anomalies) as a baseline in experiments.
>
> ### W2: Lack of Baselines
>
> **A2:** Thank you for this suggestion. In our experiments, we have selected many models to cover a wider range of architectural paradigms. Now we expand our experiments to include two additional baselines to ensure a comprehensive comparison, including iTransformer[2], Leddam[3] and TimeKAN[4] (a recent model from ICLR 2025). We did not include Time-LLM[5] as it is primarily designed for zero-shot forecasting, which is orthogonal to our supervised learning setting. We also did not include GPT4TS[6] as our baseline set already includes more recent and representative models, ensuring a fair and up-to-date comparison.
>
> The full results are updated in Table 1 (overall forecasting performance) and Table 2 (robustness) in the revised manuscript. Here, we present the results of added baselines and our method. Results demonstrate RESAM's superior robustness under point-wise perturbations, and competetive performance for non-perturbed forecasting.
>
> **Table a: Performance comparison of newly added baselines for multivariate time series forecasting task. The results are averaged from all prediction lengths of $F \in \{96, 192, 336, 720\}$. MSE/MAE are reported.**
>
> | Models | ETTh1 | ETTh2 | ETTm1 | ETTm2 | Weather | Exchange | Traffic | Solar |
> |--------|----------------|----------------|----------------|----------------|------------------|-------------------|------------------|----------------|
> | iTransformer | 0.448 / 0.439 | 0.381 / 0.401 | 0.394 / 0.389 | 0.281 / 0.321 | 0.254 / 0.271 | 0.365 / 0.406 | **0.450** / 0.282 | 0.249 / 0.245 |
> | TimeKAN | 0.445 / 0.427 | 0.378 / 0.397 | 0.388 / 0.384 | 0.281 / 0.321 | 0.249 / 0.269 | 0.394 / 0.422 | 0.633 / 0.349 | 0.321 / 0.291 |
> | Leddam | 0.437 / 0.425 | 0.371 / **0.391** | 0.390 / 0.383 | 0.277 / 0.317 | **0.243** / 0.272 | 0.362 / 0.403 | 0.491 / **0.278** | **0.220** / 0.256 |
> | RobustTSF | 0.446 / 0.438 | 0.435 / 0.438 | 0.395 / 0.390 | 0.284 / 0.332 | 0.270 / 0.293 | 0.374 / **0.396** | 0.639 / 0.359 | 0.342 / 0.343 |
> | **Ours** | **0.432** / **0.423** | **0.369** / 0.392 | **0.379** / **0.379** | **0.267** / **0.309** | 0.248 / 0.269 | **0.358** / 0.403 | 0.537 / 0.308 | 0.243 / **0.244** |
>
> **Table b: Performance comparison of newly added baselines for robustness evaluation under last-point perturbation ratio $\alpha = 3.0$. The results are averaged from all prediction lengths of $F \in \{96, 192, 336, 720\}$. MSE/MAE are reported.**
>
> | Models | ETTh1 | ETTh2 | ETTm1 | ETTm2 | Weather | Exchange | Traffic | Solar |
> |--------|----------------|----------------|----------------|----------------|------------------|-------------------|------------------|----------------|
> | iTransformer | 0.500 / 0.470 | 0.404 / 0.418 | 0.575 / 0.475 | 0.327 / 0.359 | 0.317 / 0.322 | **0.410** / 0.445 | 0.583 / 0.365 | 0.754 / 0.470 |
> | TimeKAN | 0.501 / 0.461 | 0.404 / 0.418 | 0.521 / 0.450 | 0.323 / 0.357 | 0.286 / 0.302 | 0.451 / 0.467 | 0.678 / 0.379 | 0.803 / 0.482 |
> | Leddam | 0.505 / 0.467 | 0.405 / 0.418 | 0.585 / 0.471 | 0.350 / 0.372 | 0.340 / 0.347 | 0.441 / 0.461 | 0.625 / 0.351 | 2.340 / 0.761 |
> | RobustTSF | 0.608 / 0.510 | 0.503 / 0.483 | 0.739 / 0.509 | 0.362 / 0.389 | 0.471 / 0.382 | 0.411 / **0.435** | 0.926 / 0.450 | 2.071 / 0.761 |
> | **Ours** | **0.461** / **0.442** | **0.379** / **0.401** | **0.416** / **0.404** | **0.278** / **0.319** | **0.263** / **0.286** | 0.421 / 0.460 | **0.550** / **0.319** | **0.340** / **0.308** |

---

> ### Author Response · Authors · 2025-11-27
>
> ### W3: Lack of Short-term Forecasting Experiments
>
> A3: We have followed the reviewer's suggestion and added short-term forecasting experiments. Due to time limit, we have carried out experiments across four datasets with different domains and sampling rates. Results for prediction length F=6, F=12, and F=24 are shown in the following Table c.
> The results demonstrate that RESAM maintains its strong and competitive performance in short-term forecasting scenarios.
>
>
> **Table c: Short-term forecasting results for robustness evaluation under last-point perturbation ratio $\alpha = 3.0$. The results are averaged from all prediction lengths of $F \in \{6, 12, 24\}$. MSE/MAE are reported.**
>
> | Models | ETTh2 | ETTm2 | Weather | Traffic |
> |--------|----------------|----------------|------------------|------------------|
> | DLinear | 0.301 / 0.365 | 0.469 / 0.452 | 0.655 / 0.408 | 2.065 / 0.712 |
> | TimeMixer | 0.392 / 0.409 | 0.338 / 0.380 | 0.194 / 0.244 | 1.029 / 0.492 |
> | CycleNet | 0.466 / 0.452 | 0.354 / 0.389 | 0.623 / 0.398 | 1.144 / 0.501 |
> | Crossformer | 0.369 / 0.418 | 0.350 / 0.395 | 0.274 / 0.314 | 1.448 / 0.684 |
> | PatchTST | 0.389 / 0.413 | 0.419 / 0.407 | 0.263 / 0.278 | 2.479 / 0.840 |
> | MICN | 0.228 / 0.321 | 0.234 / 0.327 | 0.302 / 0.297 | 0.839 / 0.465 |
> | TimesNet | 0.196 / 0.293 | 0.162 / 0.264 | **0.106** / **0.144** | 0.613 / 0.322 |
> | FEDformer | 0.237 / 0.327 | 0.178 / 0.281 | 0.189 / 0.246 | 0.608 / 0.392 |
> | FreTS | 0.652 / 0.498 | 0.532 / 0.458 | 0.269 / 0.326 | 6.917 / 1.217 |
> | FiLM | 0.286 / 0.336 | 0.425 / 0.423 | 0.311 / 0.260 | 1.940 / 0.704 |
> | iTransformer | 0.363 / 0.400 | 0.334 / 0.388 | 0.264 / 0.275 | 1.919 / 0.804 |
> | TimeKAN | 0.423 / 0.442 | 0.334 / 0.383 | 0.220 / 0.250 | 1.087 / 0.574 |
> | Leddam | 0.352 / 0.399 | 0.408 / 0.418 | 0.278 / 0.307 | 2.313 / 0.813 |
> | **Ours** | **0.186** / **0.283** | **0.127** / **0.228** | 0.127 / 0.168 | **0.483** / **0.318** |

---

> ### Author Response · Authors · 2025-11-27
>
> ### W4: Overall Performance and Robustness
>
> A4: We agree with the reviewer that overall performance and a meaningful assessment of robustness are crucial, and we believe our experiments validates the performance of our proposed RESAM. Reasons are as follows:
>
> 1. **Superior Robustness is the Key Contribution:** The primary goal of RESAM is to achieve robustness against perturbations. As shown in Table 2 of the manuscript, under point-wise perturbations, RESAM achieves the best overall performance, obtaining the highest number of 1st ranks (31) and the best average rank (1.33) across all datasets and prediction lengths. This clearly demonstrates its superior robustness.
>
> 2. **Competitive Forecasting Accuracy:** Despite the simple dual-layer MLP backbone, RESAM achieves competitive forecasting accuracy under normal (non-perturbed) conditions. This indicates the effectiveness of our proposed method.
>
> 3. **MSE vs. MSE$\Uparrow$:** We argue that the ultimate goal of a robust model is to maintain a low forecasting error under perturbed conditions, not just to have a small increase in error (MSE$\Uparrow$). A model with high initial MSE might show a small MSE$\Uparrow$ but still perform poorly after perturbation. As shown in Table 2 of the manuscript, while TimesNet and Crossformer have a lower relative MSE$\Uparrow$ (Table 3), their absolute MSE values after perturbation are often higher than RESAM's, meaning RESAM's predictions are actually more accurate in the presence of perturbations. In extreme cases, a constant predictor would have an MSE$\Uparrow$ of 0, but its predictions would be useless. Therefore, the absolute MSE after perturbation is a more meaningful metric.
>
> 4. **Robustness of TimesNet and Crossformer:** To further validate our method's general robustness, we conducted new experiments of **block-wise perturbations** on the recent points (corrupting a continuous segment of points). The full results are added in Table 5 in the revised manuscript. Here, we present the results of TimesNet, Crossformer and our method in the following Table d. Results demonstrate that RESAM also performs robustly under this more severe corruption type, whereas models like TimesNet and Crossformer exhibit significant performance degradation.
>
> **Table d: Performance comparison of TimesNet and Crossformer for robustness evaluation under last-block perturbation ratio $\alpha = 3.0$. The results are averaged from all prediction lengths of $F \in \{96, 192, 336, 720\}$. MSE/MAE are reported.**
>
> | Models | ETTh1 | ETTh2 | ETTm1 | ETTm2 | Weather | Exchange | Traffic |
> |--------|----------------|----------------|----------------|----------------|------------------|-------------------|------------------|
> | TimesNet | 0.501 / 0.468 | 0.412 / 0.419 | 0.445 / 0.428 | 0.309 / 0.342 | 0.292 / 0.308 | **0.443 / 0.454** | 0.766 / 0.397 |
> | Crossformer | 0.536 / 0.512 | 0.895 / 0.658 | 0.522 / 0.472 | 0.583 / 0.503 | 0.282 / 0.309 | 0.619 / 0.596 | 0.681 / 0.352 |
> | **Ours** | **0.492** / **0.459** | **0.391** / **0.410** | **0.436** / **0.434** | **0.290** / **0.329** | **0.276** / **0.296** | 0.458 / 0.483 | **0.596** / **0.340** |
>
>
> ### References
>
> [1] RobustTSF: Towards Theory and Design of Robust Time Series Forecasting with Anomalies. ICLR 2024.
>
> [2] iTransformer: Inverted Transformers Are Effective for Time Series Forecasting. ICLR 2024.
>
> [3] Revitalizing multivariate time series forecasting: Learnable decomposition with inter-series dependencies and intra-series variations modeling. ICML 2024.
>
> [4] TimeKAN: KAN-based Frequency Decomposition Learning Architecture for Long-term Time Series Forecasting. ICLR 2025.
>
> [5] Time-LLM: Time Series Forecasting by Reprogramming Large Language Models. ICLR 2024.
>
> [6] One Fits All: Power General Time Series Analysis by Pretrained LM. NIPS 2023.

---

### Official Review · Reviewer_FyDr · 2025-10-31

**Soundness:** 3
**Presentation:** 2
**Contribution:** 2
**Rating:** 4
**Confidence:** 4

**Summary:**

This paper proposes a randomized sampling strategy to exploit global context to mitigate the impact of point-wise perturbations for time series forecasting. Experiments on 8 datasets show the robustness to point-wise perturbations.

**Strengths:**

1. This motivation is easy to follow.
2. Visualization and showcases.
3. Code are provided for reproducibility.
4. Randomized sampling strategy to exploit global context to mitigate the impact of point-wise perturbations is very interesting.

**Weaknesses:**

1. The claimed contributions and novelty are exaggerated. There are existing works on anomalies or perturbations on time series forecasting, such as [1-2]. There are also many works on basis / decycle or decomposition, such as [3-4]. And the proposed Learnable Periodicity Extraction module seems to be from [3], or I cannot find how to learn periodic cycle $W$ from the presentation in Section 3.2.

[1] RobustTSF: Towards Theory and Design of Robust Time Series Forecasting with Anomalies. ICLR 2024.

[2] Weakly Guided Adaptation for Robust Time Series Forecasting. Proc. VLDB Endow. 17(4): 766-779 (2023).

[3] Revitalizing multivariate time series forecasting: Learnable decomposition with inter-series dependencies and intra-series variations modeling. ICML 2024.

[4] CycleNet Enhancing time series forecasting through modeling periodic patterns, Lin et al., NIPS 2024.

2. Key experimental comparison for baselines [1-2] is missing. The author should also compare state-of-the-art baselines such as [3].

3. How to evaluate the improvements by identifying the real perturbations in original datasets? Or the improvements are just gained by randomized sampling strategy which enhance generalization ability?

**Questions:**

see weaknesses.

---

> ### Author Response · Authors · 2025-11-27
>
> We sincerely thank Reviewer FyDr for their insightful comments. We have revised the manuscript to address the concerns, and provide a point-by-point response below.
>
> ### W1: Novelty Concerns
>
> **A1:** The primary novelty of our paper lies in being **the first to systematically identify and address** a critical vulnerability in time series forecasting models: their excessive reliance on the most recent points in the historical window. As demonstrated in Figure 1 and our empirical analysis in Section 2.3, even a single anomalous point, particularly at the end of the input sequence, can cause catastrophic failure in predictions from state-of-the-art models. This discovery is significant because it reveals a fundamental robustness limitation not previously explored in depth. Based on this finding, our **second** key contribution is the design of RESAM, an integrated framework specifically tailored to mitigate this vulnerability. RESAM represents the first solution to comprehensively exploit global context for robustness against point-wise perturbations.
>
> We now address the specific concerns:
>
> **(1) Regarding similarity to CycleNet**: We acknowledge that the Learnable Periodicity Extraction (LPE) module shares the high-level concept of learning periodic patterns with CycleNet[4]. However, its role and integration within RESAM are distinct and serve our specific goal. In our framework, LPE functions primarily to extract and preserve the periodic component before applying sampling. The LPE is not the core contribution; rather, it is a crucial component that enables the effective operation of our main innovation—the BARS strategy and the two-stage training protocol—within a decycled residual space.
>
> **(2) Regarding existing TSF with anomalies research:** While there are indeed excellent works on time series forecasting with anomalies or general robustness, they typically focus on different types of problems, such as common anomalies, or distribution shifts. To the best of our knowledge, no prior work has specifically targeted and systematically evaluated the vulnerability to **point-wise perturbations on recent points** in the historical context. Our work fills this gap. We have added a discussion in **Section Introduction (Line 62-73)** explaining why existing robustness methods are not sufficient for this specific vulnerability, as their objectives and mechanisms are different.
>
> ### W2: Lack of Baselines
>
> **A2:** We have added comparisons with RobustTSF [1] and Leddam [3] as suggested. We also add two other baselines which are published in ICLR 2024 and ICLR 2025 respectively, including iTransformer[5] and TimeKAN[6]. DARF[2] was excluded as it addresses distribution shifts between holidays and non-holidays, which is orthogonal to our focus.
>
> The full results are updated in Table 1 (overall forecasting performance) and Table 2 (robustness) in the revised manuscript. Here, we present the results of added baselines and our method. Results demonstrate RESAM's superior robustness under point-wise perturbations, and competetive performance for non-perturbed forecasting.
>
> **Table a: Performance comparison of newly added baselines for multivariate time series forecasting task. The results are averaged from all prediction lengths of $F \in \{96, 192, 336, 720\}$. MSE/MAE are reported.**
>
> | Models | ETTh1 | ETTh2 | ETTm1 | ETTm2 | Weather | Exchange | Traffic | Solar |
> |--------|----------------|----------------|----------------|----------------|------------------|-------------------|------------------|----------------|
> | iTransformer | 0.448 / 0.439 | 0.381 / 0.401 | 0.394 / 0.389 | 0.281 / 0.321 | 0.254 / 0.271 | 0.365 / 0.406 | **0.450** / 0.282 | 0.249 / 0.245 |
> | TimeKAN | 0.445 / 0.427 | 0.378 / 0.397 | 0.388 / 0.384 | 0.281 / 0.321 | 0.249 / 0.269 | 0.394 / 0.422 | 0.633 / 0.349 | 0.321 / 0.291 |
> | Leddam | 0.437 / 0.425 | 0.371 / **0.391** | 0.390 / 0.383 | 0.277 / 0.317 | **0.243** / 0.272 | 0.362 / 0.403 | 0.491 / **0.278** | **0.220** / 0.256 |
> | RobustTSF | 0.446 / 0.438 | 0.435 / 0.438 | 0.395 / 0.390 | 0.284 / 0.332 | 0.270 / 0.293 | 0.374 / **0.396** | 0.639 / 0.359 | 0.342 / 0.343 |
> | **Ours** | **0.432** / **0.423** | **0.369** / 0.392 | **0.379** / **0.379** | **0.267** / **0.309** | 0.248 / 0.269 | **0.358** / 0.403 | 0.537 / 0.308 | 0.243 / **0.244** |

---

> ### Author Response · Authors · 2025-11-27
>
> **Table b: Performance comparison of newly added baselines for robustness evaluation under last-point perturbation ratio $\alpha = 3.0$. The results are averaged from all prediction lengths of $F \in \{96, 192, 336, 720\}$. MSE/MAE are reported.**
>
> | Models | ETTh1 | ETTh2 | ETTm1 | ETTm2 | Weather | Exchange | Traffic | Solar |
> |--------|----------------|----------------|----------------|----------------|------------------|-------------------|------------------|----------------|
> | iTransformer | 0.500 / 0.470 | 0.404 / 0.418 | 0.575 / 0.475 | 0.327 / 0.359 | 0.317 / 0.322 | **0.410** / 0.445 | 0.583 / 0.365 | 0.754 / 0.470 |
> | TimeKAN | 0.501 / 0.461 | 0.404 / 0.418 | 0.521 / 0.450 | 0.323 / 0.357 | 0.286 / 0.302 | 0.451 / 0.467 | 0.678 / 0.379 | 0.803 / 0.482 |
> | Leddam | 0.505 / 0.467 | 0.405 / 0.418 | 0.585 / 0.471 | 0.350 / 0.372 | 0.340 / 0.347 | 0.441 / 0.461 | 0.625 / 0.351 | 2.340 / 0.761 |
> | RobustTSF | 0.608 / 0.510 | 0.503 / 0.483 | 0.739 / 0.509 | 0.362 / 0.389 | 0.471 / 0.382 | 0.411 / **0.435** | 0.926 / 0.450 | 2.071 / 0.761 |
> | **Ours** | **0.461** / **0.442** | **0.379** / **0.401** | **0.416** / **0.404** | **0.278** / **0.319** | **0.263** / **0.286** | 0.421 / 0.460 | **0.550** / **0.319** | **0.340** / **0.308** |
>
>
> ### W3: Evaluation & Robustness Mechanism
>
> **A3:**
> **Evaluation:** We employ controlled perturbations (Section 4.2) to isolate robustness from detection biases. This approach is consistent in robustness literature like RobustTSF[1] and allows direct comparison.
>
> **Robustness Mechanism:** RESAM's robustness stems from:
> **(1) BARS-enforced global context utilization:** Random sampling during training forces the model to learn from arbitrary point combinations, reducing dependence on specific points.
> **(2) Balanced temporal importance:** Unlike baselines that overweight recent points, RESAM shows relatively uniform importance distribution across the history (Figure 7). This balanced temporal utilization allows for robust predictions that withstand local perturbations while effectively capturing recent trends.
>
>
> ### References
>
> [1] RobustTSF: Towards Theory and Design of Robust Time Series Forecasting with Anomalies. ICLR 2024.
>
> [2] Weakly Guided Adaptation for Robust Time Series Forecasting. Proc. VLDB Endow. 17(4): 766-779 (2023).
>
> [3] Revitalizing multivariate time series forecasting: Learnable decomposition with inter-series dependencies and intra-series variations modeling. ICML 2024.
>
> [4] CycleNet: Enhancing time series forecasting through modeling periodic patterns. NIPS 2024.
>
> [5] iTransformer: Inverted Transformers Are Effective for Time Series Forecasting. ICLR 2024.
>
> [6] TimeKAN: KAN-based Frequency Decomposition Learning Architecture for Long-term Time Series Forecasting. ICLR 2025.

---

### Official Review · Reviewer_kTBq · 2025-11-06

**Soundness:** 3
**Presentation:** 3
**Contribution:** 2
**Rating:** 4
**Confidence:** 4

**Summary:**

This paper presents RESAM, a method for robust time series forecasting designed to mitigate vulnerability to point-wise perturbations, particularly on the most recent input points. The approach combines Basis-Aligned Randomized Sampling (BARS) to represent sampled time points via trigonometric basis functions and a Learnable Periodicity Extraction (LPE) module that learns the periodic component independently before sampling. A two-stage training procedure further stabilizes dual-space learning. Extensive experiments across eight benchmarks demonstrate improved perturbation robustness and competitive forecasting accuracy relative to prominent baselines.

**Strengths:**

1. The paper convincingly identifies and empirically demonstrates a core vulnerability in modern time series forecasting architectures: an over-reliance on recent input points, as shown in Figure 1 and Figure 3. This motivates the necessity for robustness-oriented model design.
2. A comprehensive set of ablation studies (Figure 8) shows that both LPE and the BARS sampling improve robustness and accuracy, offering deeper insight into the contribution of each component.
3. The paper is clearly and lucidly written, making the methodology easy to follow. The authors' provision of source code further enhances the credibility and reproducibility of their work.

**Weaknesses:**

1. On Novelty and Contribution: The Learnable Periodicity Extraction (LPE) module, a key component of the proposed framework, appears to be heavily inspired by or substantially similar to prior work, particularly CycleNet. This significant overlap raises concerns about the novelty and the incremental contribution of this paper. While integrating existing ideas is valid, the reliance on this established architecture may limit the perceived originality of the overall work.
2. In Stage 1 of training, the authors replace the basis-aligned sampling module and MLP backbone with a simple linear layer. However, the ablation study does not include a baseline that only uses this Stage 1 architecture for final prediction (i.e., without proceeding to Stage 2).
3. On Overall Performance: From a practical standpoint, the empirical results suggest that RESAM does not consistently achieve state-of-the-art performance across all benchmarks. Even on the datasets where RESAM performs favorably, the margin of improvement over existing baselines is often limited. This raises questions about the practical significance of the proposed method in standard, noise-free forecasting scenarios and whether the added complexity is justified by the marginal gains.
4. On the Perturbation Model and Scope of Conclusions: The paper's core motivation hinges on the finding that existing models are more sensitive to a last-point perturbation than a random-point one. However, the realism of this specific perturbation model is a significant concern; it is difficult to identify practical, real-world scenarios where noise or corruption would systematically affect only the single most recent data point. Furthermore, the proposed methodology, especially the basis-aligned sampling, seems inherently capable of addressing other non-point-wise disturbances, such as block-wise corruption or multi-point noise. The authors' failure to discuss or evaluate their model against these more general and arguably more realistic perturbation types restricts the applicability and scope of their conclusions. This leaves the model's robustness in a broader range of practical scenarios underexplored.
5. My concerns about the limited generalizability of the conclusions are further amplified by the appendix, where the sensitivity analysis for the perturbation ratio α is conducted exclusively on the ETT series of datasets. To strengthen the paper's claims, I recommend that the authors extend this important ablation study to a more diverse range of datasets.

**Questions:**

1. **On the Necessity of Stage 2 Training**: Fundamentally, fitting basis coefficients is a linear task. It is therefore plausible that replacing the basis-fitting module with a linear layer, as done in Stage 1, could theoretically achieve comparable results. My primary concern is whether the Stage 2 training genuinely provides a significant performance improvement over the foundation established in Stage 1. To address this, could the authors supplement the ablation study by reporting the performance of the Stage 1 model (trained and evaluated on its own) across the various datasets? This would clarify the true value added by Stage 2.
2. **On the Complexity of the Two-Stage Protocol**: The introduction of a linear or MLP layer in Stage 1, which is later replaced by the full RESAM architecture in Stage 2, appears to significantly increase the model's overall parameter count and training complexity. Is this added complexity truly necessary? Could the authors consider a simpler approach for Stage 1, where only a periodic pattern matrix is trained, and the forecast is generated by simply extrapolating this learned cycle? Such a method would likely be more computationally efficient and would help isolate the benefits of the periodicity learning itself.
3. **On the Realism of the Last-Point Perturbation Model**: The paper demonstrates that RESAM is effective at mitigating last-point perturbations, but its efficacy against other point-wise disturbances seems limited by design. Could the authors comment on the practical relevance of this specific last-point perturbation model? In which real-world scenarios does this type of isolated, terminal-point corruption actually occur? Justifying the focus on this particular failure mode is crucial for the paper's practical impact.
4. **On the Generalizability to Other Perturbation Types**: Could RESAM be extrapolated to handle other common disturbance patterns, such as block-wise corruption, random multi-point noise, or missing value imputation? The current analysis is narrowly focused on a single type of perturbation. An evaluation of RESAM's performance against these more varied and arguably more common types of data corruption would provide a much more comprehensive assessment of its robustness and practical utility.

I believe this paper has potential, but the concerns raised above are significant. Should the authors provide a convincing response that effectively addresses these weaknesses and clarifies the open questions, I would be happy to reconsider my evaluation and increase my score accordingly.

---

> ### Author Response · Authors · 2025-11-27
>
> We sincerely thank Reviewer kTBq for the thorough review and valuable feedback. We have carefully addressed each concern and provide a point-by-point response.
>
> ### W1: Novelty Concerns Regarding LPE Similarity to CycleNet
>
> **A1:** The primary novelty of our paper lies in being **the first to systematically identify and address** a critical vulnerability in time series forecasting models: their excessive reliance on the most recent points in the historical window. As demonstrated in Figure 1 and our empirical analysis in Section 2.3, even a single anomalous point, particularly at the end of the input sequence, can cause catastrophic failure in predictions from state-of-the-art models. This discovery is significant because it reveals a fundamental robustness limitation not previously explored in depth. Based on this finding, our **second** key contribution is the design of RESAM, an integrated framework specifically tailored to mitigate this vulnerability.
>
> We understand the concerns of the reviewer for the novelty on the Learnable Periodicity Extraction (LPE) module. We acknowledge that the LPE module shares the high-level concept of learning periodic patterns with CycleNet. However, its role and integration within RESAM are distinct and serve our specific goal. In our framework, LPE functions primarily to extract and preserve the periodic component before applying sampling. The LPE is not the core contribution; rather, it is a crucial component that enables the effective operation of our main innovation—the BARS strategy and the two-stage training protocol—within a decycled residual space.
>
> ### W2 & Q1 & Q2: Necessity of Stage 2 Training and Complexity of the 2-Stage Protocol
>
> **A2:** We thank the reviewer for this important suggestion. We have add new ablation studies as recommended:
>
> 1. Cycle-Naive: Using a simplified periodic extrapolation method in stage 1 learning where only a learned periodic pattern is repeated for forecasting.
> 2. Stage-1 Model: Using only the Stage-1 architecture (linear layer) for direct prediction.
>
> Results for non-perturbed and perturbed forecasting are presented in the following two tables. (1) Cycle-Naive shows worse performance because it doesn't perform proper seasonal-trend decomposition in the first stage, leading to inaccurate cycle initialization for the second stage. Since the stage-1 architecture of RESAM only adds a linear layer on Cycle-Naive, it still remains computationally simple. (2) Stage-1 Model performs worse because it lacks the BARS strategy which is essential for global context utilization and robustness.
>
> **Table a: Ablation study for two new variants. The results are calculated without perturbations and are averaged from all prediction lengths of $F \in \{96, 192, 336, 720\}$. MSE/MAE are reported.**
>
> | Models | ETTh1 | ETTh2 | ETTm1 | ETTm2 | Weather | Exchange | Traffic | Solar |
> |--------|----------------|----------------|----------------|----------------|------------------|-------------------|------------------|----------------|
> | Cycle-Naive | 0.437 / 0.428 | 0.373 / 0.395 | 0.382 / 0.380 | 0.268 / 0.309 | 0.250 / 0.271 | 0.377 / 0.415 | 0.561 / 0.304 | 0.247 / 0.249 |
> | Stage-1 Model | **0.429** / **0.418** | 0.375 / 0.394 | 0.386 / 0.383 | 0.273 / 0.312 | 0.258 / 0.276 | 0.390 / 0.419 | **0.489** / **0.292** | 0.292 / 0.266 |
> | **Ours** | 0.432 / 0.423 | **0.369** / **0.392** | **0.379** / **0.379** | **0.267** / **0.309** | **0.248** / **0.269** | **0.358** / **0.403** | 0.537 / 0.308 | **0.243** / **0.244** |
>
> **Table b: Ablation study for two new variants. The results are calculated after perturbations and are averaged from all prediction lengths of $F \in \{96, 192, 336, 720\}$. MSE/MAE after perturbations are reported.**
>
> | Models | ETTh1 | ETTh2 | ETTm1 | ETTm2 | Weather | Exchange | Traffic | Solar |
> |--------|----------------|----------------|----------------|----------------|------------------|-------------------|------------------|----------------|
> | Cycle-Naive | 0.469 / 0.448 | 0.383 / 0.404 | 0.569 / 0.463 | 0.281 / 0.323 | 0.268 / **0.286** | 0.485 / 0.491 | 0.574 / **0.314** | 1.009 / 0.458 |
> | Stage-1 Model | 0.565 / 0.483 | 0.434 / 0.436 | 0.619 / 0.479 | 0.320 / 0.353 | 0.349 / 0.331 | 0.429 / **0.452** | 0.588 / 0.340 | 0.664 / 0.405 |
> | **Ours** | **0.461** / **0.442** | **0.379** / **0.401** | **0.416** / **0.404** | **0.278** / **0.319** | **0.263** / 0.286 | **0.421** / 0.460 | **0.550** / 0.319 | **0.340** / **0.308** |

---

> ### Author Response · Authors · 2025-11-27
>
> ### W3: Overall Performance in Non-Perturbed Scenarios
>
> **A3:** The primary goal of RESAM is to achieve robustness against perturbations, where it demonstrates superior performance as shown in Table 2 of the manuscript. However, in non-perturbed forecasting scenarios, RESAM also achieves competitive performance with state-of-the-art methods, using a simple dual-layer MLP backbone. The marginal improvements in some cases are offset by the significant robustness gains, providing a favorable trade-off for practical applications.
>
> ### W4 & Q3: Practical Relevance of Last-Point Perturbation
>
> **A4:** In time series forecasting, predictions are generated sequentially in real-time environments. When an anomaly occurs, it invariably affects the most recent observation​ in the historical window used for forecasting. This makes last-point perturbations not just a theoretical concern but a fundamental challenge in operational forecasting systems.
>
> Besides, last-point perturbations frequently occur in critical applications:
>
> - IoT and Sensor Networks: Transmission failures or sensor malfunctions often affect the most recent data point in continuous monitoring systems
> - Financial Systems: Real-time data feed interruptions or corruption in high-frequency trading
> - AIOps: Anomalous events affecting the latest monitoring metrics
>
> ### W4 & Q4: Generalizability to Other Perturbation Types
>
> **A5:** We have expanded our experiments to include a comprehensive evaluation of seven different perturbation types within two different positions (last & random):
>
> - Last perturbations: Last anomalous point, last block, last missing point
> - Random perturbations: Random anomalous point, random multi anomalous point, random block, random missing point
>
> For "block" perturbations, we perturb a continuous points with a random length of 2-5. For "missing" perturbations, we set the value of one point to zero. "Last" perturbations are injected in the recent points of historical window, and "random" perturbations are injected in random positions. Results are presented in the following a few tables (Table c-i in the next comments, and also shown as Table 2 and Table 5-10 in revised manuscript), where they are averaged from prediction lengths $F \in \{96, 192, 336, 720\}$. Perturbation ratio is $\alpha = 3.0$. Post-perturbation MSE and MAE are reported.
>
> Results show that our RESAM generalizes well beyond just last-point perturbations. And in terms of "last" perturbations, RESAM performs greatly better than baselines. Note that in terms of "random" perturbations, existing methods tend to perform better than "last" perturbations. **This phenomenon is consistent with our observation that existing methods highly depend on recent points.**
>
> ### W5: Limited Generalizability of $\alpha$ Sensitivity Analysis
>
> **A6:** We have extended the sensitivity analysis on perturbation ratio $\alpha$ to include Weather and Traffic datasets, in addition to the ETT series. Results are presented in the following table, which are consistent with those of ETT series. Both the prediction MSE and MSE$\Uparrow$ after pertuabations exhibit an approximately linear relationship with the perturbation ratio $\alpha$, increasing as $\alpha$ grows.
>
> **Table j: Parameter sensitivity on perturbation ratio $\alpha$. The results are averaged from all prediction lengths of $F \in \{96, 192, 336, 720\}$. MSE (MSE$\Uparrow$) are reported.**
>
> | $\alpha$ | ETTm1  | ETTm2 | Weather | Traffic |
> |--------|----------------|----------------|----------------|----------------|
> | 1 | 0.384 (1.31%$\Uparrow$) | 0.269 (0.82%$\Uparrow$) | 0.252 (1.61%$\Uparrow$) | 0.544 (1.30%$\Uparrow$) |
> | 3 | 0.416 (10.33%$\Uparrow$) | 0.278 (4.94%$\Uparrow$) | 0.263 (5.65%$\Uparrow$) | 0.550 (2.80%$\Uparrow$) |
> | 5 | 0.484 (29.25%$\Uparrow$) | 0.292 (11.50%$\Uparrow$) | 0.273 (10.08%$\Uparrow$) | 0.589 (9.68%$\Uparrow$) |
> | 7 | 0.595 (59.96%$\Uparrow$) | 0.313 (21.22%$\Uparrow$) | 0.286 (15.32%$\Uparrow$) | 0.642 (19.55%$\Uparrow$) |

---

> ### Author Response · Authors · 2025-11-27
>
> Table c-e are results of three types of "last" perturbations.
>
> **Table c: Robustness evaluation under "last anomalous point" perturbation.**
>
> | Models | ETTh1 | ETTh2 | ETTm1 | ETTm2 | Weather | Exchange | Traffic | Solar | 1st Count | Avg Rank |
> |--------|----------------|----------------|----------------|----------------|------------------|-------------------|------------------|----------------|----------|---------|
> | DLinear | 0.566 / 0.493 | 0.448 / 0.451 | 0.623 / 0.477 | 0.331 / 0.366 | 0.371 / 0.345 | 0.373 / 0.434 | 0.902 / 0.422 | 3.175 / 0.851 | 5 | 5.38 |
> | TimeMixer | 0.552 / 0.486 | 0.407 / 0.419 | 0.534 / 0.452 | 0.318 / 0.352 | 0.298 / 0.311 | 0.439 / 0.465 | 0.568 / 0.346 | 0.569 / 0.449 | 0 | 3.46 |
> | CycleNet | 0.565 / 0.483 | 0.434 / 0.436 | 0.619 / 0.479 | 0.320 / 0.353 | 0.349 / 0.331 | 0.429 / 0.452 | 0.588 / 0.340 | 0.664 / 0.405 | 0 | 4.25 |
> | Crossformer | 0.524 / 0.503 | 0.980 / 0.690 | 0.514 / 0.467 | 0.571 / 0.496 | 0.274 / 0.303 | 0.612 / 0.591 | 0.593 / **0.289** | **0.327** / **0.288** | 8 | 4.13 |
> | PatchTST | 0.554 / 0.486 | 0.400 / 0.415 | 0.645 / 0.491 | 0.346 / 0.369 | 0.306 / 0.316 | 0.468 / 0.485 | 0.601 / 0.353 | 0.725 / 0.507 | 0 | 4.39 |
> | MICN | 0.474 / 0.467 | 0.474 / 0.459 | 0.484 / 0.441 | 0.309 / 0.360 | 0.295 / 0.313 | **0.325** / **0.398** | 0.623 / 0.318 | 0.359 / 0.345 | 6 | 2.51 |
> | TimesNet | 0.472 / 0.453 | 0.396 / 0.409 | 0.418 / 0.411 | 0.296 / 0.331 | **0.262** / **0.284** | 0.435 / 0.448 | 0.700 / 0.360 | 0.382 / 0.354 | 9 | 2.20 |
> | FEDformer | 0.467 / 0.459 | 0.408 / 0.426 | 0.789 / 0.591 | 0.348 / 0.388 | 0.314 / 0.346 | 0.530 / 0.497 | 0.666 / 0.389 | 0.657 / 0.607 | 3 | 4.97 |
> | FreTS | 0.677 / 0.532 | 0.576 / 0.492 | 0.745 / 0.519 | 0.404 / 0.404 | 0.310 / 0.336 | 0.423 / 0.456 | 0.792 / 0.405 | 1.369 / 0.586 | 0 | 5.93 |
> | FiLM | 0.492 / 0.453 | 0.389 / 0.406 | 0.511 / 0.438 | 0.315 / 0.348 | 0.303 / 0.305 | 0.380 / 0.421 | 1.281 / 0.708 | 1.173 / 0.629 | 0 | 3.24 |
> | iTransformer | 0.500 / 0.470 | 0.404 / 0.418 | 0.575 / 0.475 | 0.327 / 0.359 | 0.317 / 0.322 | 0.410 / 0.445 | 0.583 / 0.365 | 0.754 / 0.470 | 1 | 3.68 |
> | TimeKAN | 0.501 / 0.461 | 0.404 / 0.418 | 0.521 / 0.450 | 0.323 / 0.357 | 0.286 / 0.302 | 0.451 / 0.467 | 0.678 / 0.379 | 0.803 / 0.482 | 1 | 3.48 |
> | Leddam | 0.505 / 0.467 | 0.405 / 0.418 | 0.585 / 0.471 | 0.350 / 0.372 | 0.340 / 0.347 | 0.441 / 0.461 | 0.625 / 0.351 | 2.340 / 0.761 | 0 | 4.70 |
> | RobustTSF | 0.608 / 0.510 | 0.503 / 0.483 | 0.739 / 0.509 | 0.362 / 0.389 | 0.471 / 0.382 | 0.411 / 0.435 | 0.926 / 0.450 | 2.071 / 0.761 | 0 | 6.36 |
> | **Ours** | **0.461** / **0.442** | **0.379** / **0.401** | **0.416** / **0.404** | **0.278** / **0.319** | 0.263 / 0.286 | 0.421 / 0.460 | **0.550** / 0.319 | 0.340 / 0.308 | **31** | **1.33** |

---

> ### Author Response · Authors · 2025-11-27
>
> **Table d: Robustness evaluation under "last block" perturbation.**
>
> | Models | ETTh1 | ETTh2 | ETTm1 | ETTm2 | Weather | Exchange | Traffic | Solar | 1st Count | Avg Rank |
> |--------|----------------|----------------|----------------|----------------|------------------|-------------------|------------------|----------------|----------|---------|
> | DLinear | 0.594 / 0.507 | 0.456 / 0.457 | 0.654 / 0.488 | 0.337 / 0.371 | 0.388 / 0.351 | 0.373 / 0.435 | 1.113 / 0.481 | 3.403 / 0.877 | 5 | 5.01 |
> | TimeMixer | 0.617 / 0.514 | 0.434 / 0.436 | 0.617 / 0.480 | 0.344 / 0.370 | 0.324 / 0.326 | 0.484 / 0.490 | 0.610 / 0.378 | 1.073 / 0.593 | 0 | 4.05 |
> | CycleNet | 0.589 / 0.495 | 0.450 / 0.446 | 0.895 / 0.596 | 0.436 / 0.441 | 0.423 / 0.380 | 0.459 / 0.472 | 0.686 / 0.386 | 1.156 / 0.576 | 0 | 5.34 |
> | Crossformer | 0.536 / 0.512 | 0.895 / 0.658 | 0.522 / 0.472 | 0.583 / 0.503 | 0.282 / 0.309 | 0.619 / 0.596 | 0.681 / 0.352 | **0.342** / **0.295** | 10 | 3.99 |
> | PatchTST | 0.583 / 0.500 | 0.405 / 0.419 | 0.681 / 0.501 | 0.350 / 0.371 | 0.311 / 0.317 | 0.483 / 0.493 | 0.631 / 0.365 | 0.850 / 0.546 | 0 | 3.62 |
> | MICN | 0.503 / 0.475 | 0.475 / 0.460 | 0.495 / 0.446 | 0.312 / 0.363 | 0.315 / 0.326 | **0.329** / **0.401** | 0.625 / **0.320** | 0.370 / 0.354 | 7 | 2.01 |
> | TimesNet | 0.501 / 0.468 | 0.412 / 0.419 | 0.445 / **0.428** | 0.309 / 0.342 | 0.292 / 0.308 | 0.443 / 0.454 | 0.766 / 0.397 | 0.439 / 0.399 | 3 | 2.32 |
> | FEDformer | **0.487** / 0.472 | 0.410 / 0.427 | 0.959 / 0.642 | 0.353 / 0.393 | 0.357 / 0.373 | 0.530 / 0.497 | 0.691 / 0.408 | 0.823 / 0.680 | 4 | 4.61 |
> | FreTS | 0.734 / 0.553 | 0.509 / 0.473 | 0.894 / 0.564 | 0.422 / 0.413 | 0.307 / 0.334 | 0.445 / 0.472 | 0.837 / 0.436 | 1.541 / 0.627 | 0 | 5.66 |
> | FiLM | 0.530 / 0.474 | 0.400 / 0.414 | 0.584 / 0.465 | 0.330 / 0.359 | 0.342 / 0.326 | 0.400 / 0.435 | 1.321 / 0.722 | 1.673 / 0.730 | 0 | 3.63 |
> | iTransformer | 0.542 / 0.491 | 0.424 / 0.432 | 0.655 / 0.505 | 0.349 / 0.374 | 0.327 / 0.328 | 0.439 / 0.465 | 0.633 / 0.398 | 0.968 / 0.543 | 1 | 3.86 |
> | TimeKAN | 0.535 / 0.478 | 0.414 / 0.426 | 0.551 / 0.462 | 0.337 / 0.367 | 0.304 / 0.315 | 0.465 / 0.477 | 0.751 / 0.412 | 1.037 / 0.573 | 2 | 3.37 |
> | Leddam | 0.535 / 0.482 | 0.421 / 0.428 | 0.647 / 0.493 | 0.372 / 0.383 | 0.397 / 0.367 | 0.461 / 0.473 | 0.674 / 0.383 | 3.153 / 0.909 | 0 | 4.66 |
> | RobustTSF | 0.639 / 0.525 | 0.512 / 0.488 | 0.749 / 0.514 | 0.364 / 0.391 | 0.479 / 0.383 | 0.417 / 0.440 | 1.034 / 0.491 | 2.073 / 0.762 | 0 | 6.02 |
> | **Ours** | 0.492 / **0.459** | **0.391** / **0.410** | **0.436** / 0.434 | **0.290** / **0.329** | **0.276** / **0.296** | 0.458 / 0.483 | **0.596** / 0.340 | 1.618 / 0.609 | **32** | **1.85** |

---

> ### Author Response · Authors · 2025-11-27
>
> **Table e: Robustness evaluation under "last missing point" perturbation.**
>
> | Models | ETTh1 | ETTh2 | ETTm1 | ETTm2 | Weather | Exchange | Traffic | Solar | 1st Count | Avg Rank |
> |--------|----------------|----------------|----------------|----------------|------------------|-------------------|------------------|----------------|----------|---------|
> | DLinear | 0.520 / 0.486 | 0.451 / 0.478 | 0.532 / 0.475 | 0.409 / 0.433 | 77.463 / 2.232 | 19.005 / 2.652 | 0.729 / 0.383 | 0.771 / 0.440 | 2 | 5.28 |
> | TimeMixer | 0.534 / 0.490 | 0.416 / 0.443 | 0.474 / 0.446 | 0.391 / 0.422 | 340.153 / 4.787 | 29.535 / 4.570 | **0.516** / 0.317 | 0.343 / 0.328 | 3 | 4.41 |
> | CycleNet | 0.514 / 0.473 | 0.451 / 0.470 | 0.747 / 0.587 | 0.494 / 0.483 | 68.055 / 2.266 | 10.530 / 2.924 | 0.517 / 0.310 | 0.442 / 0.341 | 1 | 5.22 |
> | Crossformer | 0.548 / 0.521 | 0.887 / 0.655 | 0.548 / 0.489 | 0.554 / 0.494 | 0.488 / 0.438 | 1.851 / 1.109 | 0.664 / 0.338 | 0.310 / **0.265** | 6 | 4.02 |
> | PatchTST | 0.543 / 0.498 | 0.399 / 0.427 | 0.539 / 0.492 | 0.425 / 0.454 | 2.503 / 0.561 | 10.197 / 3.026 | 0.588 / 0.333 | 0.309 / 0.324 | 1 | 3.91 |
> | MICN | 0.481 / 0.473 | 0.467 / 0.473 | 0.471 / 0.448 | 0.331 / 0.396 | 27.866 / 1.569 | 5.791 / 2.253 | 0.622 / 0.314 | 0.357 / 0.315 | 1 | 3.38 |
> | TimesNet | 0.471 / 0.454 | 0.393 / 0.412 | 0.414 / 0.410 | 0.303 / 0.344 | 28.109 / 1.622 | 2.353 / 1.188 | 0.682 / 0.350 | 0.350 / 0.328 | 0 | 2.84 |
> | FEDformer | 0.464 / 0.457 | 0.416 / 0.432 | 0.649 / 0.562 | 0.365 / 0.404 | 26.379 / 2.979 | **0.523** / **0.507** | 0.696 / 0.406 | 1.043 / 0.742 | 5 | 4.45 |
> | FreTS | 0.613 / 0.532 | 0.495 / 0.490 | 0.587 / 0.512 | 0.537 / 0.513 | 4.172 / 0.631 | 8.267 / 2.574 | 0.619 / 0.360 | 0.327 / 0.290 | 0 | 4.91 |
> | FiLM | 0.474 / 0.448 | 0.385 / 0.408 | 0.467 / 0.436 | 0.362 / 0.392 | 15.618 / 1.114 | 1.260 / 0.791 | 1.251 / 0.693 | 0.503 / 0.420 | 3 | 3.29 |
> | iTransformer | 0.504 / 0.480 | 0.406 / 0.439 | 0.536 / 0.496 | 0.415 / 0.448 | 31.469 / 2.012 | 7.377 / 2.582 | 0.534 / 0.332 | 0.401 / 0.316 | 1 | 4.25 |
> | TimeKAN | 0.495 / 0.462 | 0.396 / 0.427 | 0.468 / 0.442 | 0.366 / 0.404 | 156.218 / 2.945 | 13.858 / 3.206 | 0.663 / 0.366 | 0.349 / 0.305 | 1 | 3.72 |
> | Leddam | 0.494 / 0.468 | 0.408 / 0.441 | 0.538 / 0.488 | 0.423 / 0.448 | 69.210 / 2.582 | 10.800 / 3.123 | 0.595 / 0.341 | 0.321 / 0.280 | 0 | 4.23 |
> | RobustTSF | 0.474 / 0.463 | 0.623 / 0.559 | 0.454 / 0.437 | 0.523 / 0.497 | **0.387** / **0.410** | 1.131 / 0.816 | 0.663 / 0.375 | 0.556 / 0.489 | 7 | 4.22 |
> | **Ours** | **0.455** / **0.440** | **0.373** / **0.407** | **0.394** / **0.394** | **0.277** / **0.330** | 3.029 / 0.732 | 69.465 / 7.477 | 0.555 / **0.308** | **0.306** / 0.276 | **33** | **1.89** |

---

> ### Author Response · Authors · 2025-11-27
>
> Next, Table f-i are results of four types of "random" perturbations.
>
> **Table f: Robustness evaluation under "random anomalous point" perturbation.**
>
> | Models | ETTh1 | ETTh2 | ETTm1 | ETTm2 | Weather | Exchange | Traffic | Solar | 1st Count | Avg Rank |
> |--------|----------------|----------------|----------------|----------------|------------------|-------------------|------------------|----------------|----------|---------|
> | DLinear | 0.454 / 0.442 | 0.418 / 0.430 | 0.403 / 0.392 | 0.287 / 0.331 | 0.272 / 0.291 | 0.318 / 0.387 | 0.723 / 0.372 | 0.427 / 0.336 | 0 | 3.52 |
> | TimeMixer | 0.469 / 0.444 | 0.376 / 0.396 | 0.405 / 0.392 | 0.280 / 0.320 | 0.246 / **0.266** | 0.383 / 0.416 | **0.509** / 0.317 | 0.358 / 0.338 | 4 | 2.41 |
> | CycleNet | **0.435** / **0.422** | 0.377 / 0.396 | 0.392 / 0.387 | 0.275 / 0.314 | 0.259 / 0.277 | 0.394 / 0.421 | 0.520 / 0.312 | 0.294 / 0.271 | 6 | 1.87 |
> | Crossformer | 0.509 / 0.492 | 0.977 / 0.688 | 0.487 / 0.448 | 0.555 / 0.487 | 0.242 / 0.273 | 0.606 / 0.585 | 0.578 / **0.279** | **0.223** / **0.220** | 15 | 3.80 |
> | PatchTST | 0.447 / 0.432 | **0.368** / **0.391** | 0.393 / 0.389 | 0.280 / 0.319 | 0.258 / 0.274 | 0.376 / 0.410 | 0.514 / 0.304 | 0.295 / 0.313 | 9 | 1.84 |
> | MICN | 0.438 / 0.444 | 0.463 / 0.451 | 0.389 / 0.392 | 0.285 / 0.338 | 0.247 / 0.276 | **0.309** / **0.379** | 0.613 / 0.310 | 0.313 / 0.309 | 11 | 2.63 |
> | TimesNet | 0.470 / 0.452 | 0.394 / 0.407 | 0.414 / 0.408 | 0.292 / 0.327 | 0.259 / 0.281 | 0.432 / 0.446 | 0.701 / 0.360 | 0.379 / 0.352 | 0 | 3.84 |
> | FEDformer | 0.454 / 0.451 | 0.407 / 0.425 | 0.599 / 0.530 | 0.342 / 0.383 | 0.305 / 0.337 | 0.530 / 0.497 | 0.654 / 0.380 | 0.548 / 0.551 | 1 | 4.80 |
> | FreTS | 0.510 / 0.470 | 0.517 / 0.456 | 0.448 / 0.418 | 0.303 / 0.342 | **0.242** / 0.272 | 0.394 / 0.425 | 0.550 / 0.325 | 0.288 / 0.281 | 1 | 3.29 |
> | FiLM | 0.452 / 0.433 | 0.386 / 0.403 | 0.408 / 0.391 | 0.288 / 0.324 | 0.283 / 0.288 | 0.376 / 0.418 | 1.241 / 0.690 | 0.423 / 0.382 | 0 | 3.59 |
> | **Ours** | 0.436 / 0.426 | 0.371 / 0.394 | **0.383** / **0.382** | **0.269** / **0.310** | 0.250 / 0.272 | 0.358 / 0.404 | 0.546 / 0.316 | 0.290 / 0.265 | **17** | **1.41** |

---

> ### Author Response · Authors · 2025-11-27
>
> **Table g: Robustness evaluation under "random block point" perturbation.**
>
> | Models | ETTh1 | ETTh2 | ETTm1 | ETTm2 | Weather | Exchange | Traffic | Solar | 1st Count | Avg Rank |
> |--------|----------------|----------------|----------------|----------------|------------------|-------------------|------------------|----------------|----------|---------|
> | DLinear | 0.469 / 0.452 | 0.420 / 0.431 | 0.417 / 0.400 | 0.289 / 0.333 | 0.276 / 0.292 | 0.319 / 0.387 | 0.856 / 0.411 | 0.439 / 0.344 | 0 | 4.54 |
> | TimeMixer | 0.485 / 0.453 | 0.383 / 0.401 | 0.422 / 0.402 | 0.282 / 0.322 | 0.248 / 0.268 | 0.391 / 0.421 | 0.538 / 0.343 | 0.483 / 0.399 | 0 | 3.53 |
> | CycleNet | 0.452 / 0.432 | 0.381 / 0.398 | 0.651 / 0.520 | 0.383 / 0.409 | 0.323 / 0.333 | 0.403 / 0.426 | 0.573 / 0.337 | 0.365 / 0.316 | 3 | 4.27 |
> | Crossformer | 0.515 / 0.497 | 0.892 / 0.656 | 0.493 / 0.451 | 0.557 / 0.488 | 0.243 / 0.274 | 0.606 / 0.585 | 0.670 / 0.344 | **0.237** / **0.230** | 13 | 5.26 |
> | PatchTST | 0.466 / 0.445 | **0.369** / **0.392** | 0.406 / 0.397 | 0.282 / 0.321 | 0.262 / 0.277 | 0.384 / 0.415 | 0.586 / 0.339 | 0.363 / 0.338 | 7 | 2.63 |
> | MICN | **0.439** / 0.445 | 0.463 / 0.451 | 0.398 / 0.398 | 0.286 / 0.338 | 0.251 / 0.279 | **0.310** / **0.380** | 0.616 / **0.313** | 0.335 / 0.327 | 12 | 2.88 |
> | TimesNet | 0.492 / 0.462 | 0.404 / 0.413 | 0.433 / 0.420 | 0.300 / 0.332 | 0.266 / 0.287 | 0.438 / 0.449 | 0.767 / 0.397 | 0.425 / 0.394 | 0 | 5.60 |
> | FEDformer | 0.462 / 0.457 | 0.408 / 0.426 | 0.659 / 0.550 | 0.343 / 0.384 | 0.309 / 0.340 | 0.530 / 0.497 | 0.669 / 0.392 | 0.583 / 0.570 | 1 | 6.34 |
> | FreTS | 0.525 / 0.478 | 0.432 / 0.429 | 0.562 / 0.464 | 0.305 / 0.346 | 0.245 / 0.276 | 0.394 / 0.426 | 0.603 / 0.358 | 0.348 / 0.315 | 0 | 4.45 |
> | FiLM | 0.466 / 0.442 | 0.390 / 0.406 | 0.424 / 0.399 | 0.290 / 0.325 | 0.284 / 0.289 | 0.382 / 0.420 | 1.257 / 0.696 | 0.437 / 0.390 | 0 | 4.75 |
> | iTransformer | 0.483 / 0.458 | 0.391 / 0.408 | 0.431 / 0.411 | 0.288 / 0.326 | 0.258 / 0.275 | 0.373 / 0.411 | **0.508** / 0.332 | 0.327 / 0.314 | 4 | 3.52 |
> | TimeKAN | 0.467 / 0.440 | 0.381 / 0.400 | 0.410 / 0.398 | 0.285 / 0.324 | 0.253 / 0.273 | 0.404 / 0.428 | 0.690 / 0.381 | 0.384 / 0.339 | 0 | 3.38 |
> | Leddam | 0.458 / 0.438 | 0.375 / 0.395 | 0.419 / 0.400 | 0.282 / 0.321 | **0.242** / **0.264** | 0.370 / 0.408 | 0.543 / 0.323 | 0.900 / 0.454 | 3 | 2.55 |
> | RobustTSF | 0.471 / 0.454 | 0.440 / 0.442 | 0.412 / 0.400 | 0.286 / 0.334 | 0.271 / 0.294 | 0.375 / 0.396 | 0.743 / 0.404 | 0.356 / 0.357 | 3 | 4.27 |
> | **Ours** | 0.446 / **0.431** | 0.375 / 0.397 | **0.392** / **0.390** | **0.272** / **0.313** | 0.253 / 0.275 | 0.367 / 0.411 | 0.587 / 0.332 | 0.378 / 0.326 | **18** | **2.02** |

---

> ### Author Response · Authors · 2025-11-27
>
> **Table h: Robustness evaluation under "random multi anomalous point" perturbation.**
>
> | Models | ETTh1 | ETTh2 | ETTm1 | ETTm2 | Weather | Exchange | Traffic | Solar | 1st Count | Avg Rank |
> |--------|----------------|----------------|----------------|----------------|------------------|-------------------|------------------|----------------|----------|---------|
> | DLinear | 0.451 / 0.442 | 0.417 / 0.431 | 0.400 / 0.392 | 0.287 / 0.333 | 1.691 / 0.372 | 0.571 / 0.457 | 0.678 / 0.360 | 0.340 / 0.314 | 2 | 4.25 |
> | TimeMixer | 0.470 / 0.446 | 0.375 / 0.398 | 0.401 / 0.391 | 0.280 / 0.323 | 13.074 / 0.768 | 2.027 / 0.950 | 0.498 / 0.306 | 0.338 / 0.325 | 0 | 4.16 |
> | CycleNet | **0.434** / **0.421** | 0.376 / 0.396 | 0.616 / 0.509 | 0.387 / 0.415 | 2.971 / 0.619 | 1.775 / 0.918 | 0.494 / 0.296 | 0.320 / 0.280 | 5 | 4.23 |
> | Crossformer | 0.511 / 0.493 | 0.888 / 0.655 | 0.486 / 0.446 | 0.554 / 0.487 | **0.257** / **0.284** | 0.696 / 0.627 | 0.654 / 0.333 | **0.220** / **0.219** | 13 | 4.75 |
> | PatchTST | 0.443 / 0.432 | **0.367** / **0.392** | 0.389 / 0.387 | 0.278 / 0.320 | 0.316 / 0.296 | 1.199 / 0.745 | 0.551 / 0.315 | 0.274 / 0.304 | 5 | 2.42 |
> | MICN | 0.439 / 0.445 | 0.461 / 0.451 | 0.389 / 0.392 | 0.283 / 0.337 | 0.825 / 0.377 | 0.509 / 0.460 | 0.612 / 0.309 | 0.284 / 0.282 | 2 | 3.51 |
> | TimesNet | 0.465 / 0.450 | 0.399 / 0.413 | 0.404 / 0.403 | 0.297 / 0.334 | 13.479 / 1.168 | 1.454 / 0.897 | 0.681 / 0.349 | 0.365 / 0.338 | 0 | 5.81 |
> | FEDformer | 0.453 / 0.450 | 0.406 / 0.426 | 0.577 / 0.524 | 0.340 / 0.383 | 2.670 / 0.910 | 0.470 / 0.469 | 0.656 / 0.380 | 0.670 / 0.597 | 1 | 5.72 |
> | FreTS | 0.503 / 0.467 | 0.427 / 0.428 | 0.420 / 0.407 | 0.303 / 0.347 | 0.373 / 0.327 | 0.645 / 0.499 | 0.533 / 0.313 | 0.264 / 0.262 | 0 | 4.23 |
> | FiLM | 0.449 / 0.432 | 0.386 / 0.405 | 0.405 / 0.391 | 0.288 / 0.325 | 1.116 / 0.389 | 1.219 / 0.684 | 1.236 / 0.687 | 0.382 / 0.365 | 0 | 4.45 |
> | iTransformer | 0.455 / 0.444 | 0.385 / 0.406 | 0.398 / 0.394 | 0.285 / 0.327 | 1.281 / 0.463 | 1.448 / 0.901 | **0.460** / 0.290 | 0.255 / 0.250 | 5 | 3.66 |
> | TimeKAN | 0.452 / 0.432 | 0.377 / 0.398 | 0.391 / 0.388 | 0.281 / 0.323 | 8.772 / 0.611 | 1.854 / 0.896 | 0.642 / 0.355 | 0.323 / 0.293 | 0 | 3.88 |
> | Leddam | 0.444 / 0.431 | 0.373 / 0.395 | 0.395 / 0.388 | 0.279 / 0.322 | 8.217 / 0.838 | 1.300 / 0.766 | 0.504 / **0.287** | 0.232 / 0.232 | 6 | 2.64 |
> | RobustTSF | 0.447 / 0.440 | 0.443 / 0.445 | 0.396 / 0.392 | 0.289 / 0.338 | 0.271 / 0.294 | **0.401** / **0.411** | 0.641 / 0.362 | 0.343 / 0.344 | 7 | 3.88 |
> | **Ours** | 0.434 / 0.425 | 0.369 / 0.394 | **0.379** / **0.380** | **0.267** / **0.311** | 1.301 / 0.531 | 4.390 / 1.151 | 0.552 / 0.305 | 0.246 / 0.245 | **18** | **2.40** |

---

> ### Author Response · Authors · 2025-11-27
>
> **Table i: Robustness evaluation under "random missing point" perturbation.**
>
> | Models | ETTh1 | ETTh2 | ETTm1 | ETTm2 | Weather | Exchange | Traffic | Solar | 1st Count | Avg Rank |
> |--------|----------------|----------------|----------------|----------------|------------------|-------------------|------------------|----------------|----------|---------|
> | DLinear | 0.500 / 0.470 | 0.427 / 0.436 | 0.447 / 0.419 | 0.295 / 0.338 | 0.282 / 0.297 | 0.324 / 0.392 | 1.078 / 0.487 | 0.749 / 0.442 | 0 | 4.49 |
> | TimeMixer | 0.520 / 0.472 | 0.391 / 0.407 | 0.465 / 0.427 | 0.292 / 0.330 | 0.259 / 0.279 | 0.400 / 0.428 | 0.595 / 0.389 | 0.580 / 0.458 | 3 | 3.43 |
> | CycleNet | 0.478 / 0.448 | 0.393 / 0.406 | 0.716 / 0.547 | 0.406 / 0.425 | 0.349 / 0.347 | 0.416 / 0.434 | 0.670 / 0.398 | 0.479 / 0.387 | 0 | 4.38 |
> | Crossformer | 0.531 / 0.509 | 0.896 / 0.658 | 0.506 / 0.461 | 0.562 / 0.491 | **0.247** / 0.278 | 0.608 / 0.586 | 0.703 / 0.369 | **0.263** / **0.247** | 13 | 4.66 |
> | PatchTST | 0.505 / 0.468 | **0.376** / **0.397** | 0.441 / 0.417 | 0.290 / 0.327 | 0.266 / 0.283 | 0.401 / 0.425 | 0.692 / 0.401 | 0.519 / 0.422 | 7 | 2.90 |
> | MICN | **0.448** / 0.451 | 0.466 / 0.453 | 0.422 / 0.415 | 0.289 / 0.341 | 0.262 / 0.291 | **0.310** / **0.381** | 0.630 / **0.322** | 0.631 / 0.536 | 15 | 2.69 |
> | TimesNet | 0.539 / 0.483 | 0.428 / 0.427 | 0.477 / 0.445 | 0.314 / 0.344 | 0.282 / 0.300 | 0.448 / 0.457 | 0.900 / 0.465 | 0.519 / 0.460 | 0 | 5.67 |
> | FEDformer | 0.481 / 0.468 | 0.410 / 0.427 | 0.752 / 0.585 | 0.345 / 0.386 | 0.316 / 0.345 | 0.530 / 0.497 | 0.693 / 0.409 | 0.643 / 0.604 | 1 | 5.78 |
> | FreTS | 0.559 / 0.496 | 0.443 / 0.435 | 0.751 / 0.533 | 0.315 / 0.354 | 0.251 / 0.283 | 0.399 / 0.430 | 0.701 / 0.418 | 0.464 / 0.390 | 0 | 4.56 |
> | FiLM | 0.493 / 0.459 | 0.400 / 0.413 | 0.452 / 0.419 | 0.296 / 0.331 | 0.291 / 0.295 | 0.389 / 0.425 | 1.288 / 0.709 | 0.587 / 0.457 | 0 | 4.34 |
> | iTransformer | 0.534 / 0.483 | 0.409 / 0.419 | 0.497 / 0.446 | 0.303 / 0.337 | 0.264 / 0.282 | 0.386 / 0.421 | **0.588** / 0.395 | 0.531 / 0.418 | 1 | 4.12 |
> | TimeKAN | 0.503 / 0.460 | 0.391 / 0.406 | 0.448 / 0.422 | 0.292 / 0.331 | 0.261 / 0.282 | 0.432 / 0.445 | 0.745 / 0.426 | 0.527 / 0.430 | 1 | 3.53 |
> | Leddam | 0.487 / 0.456 | 0.381 / 0.400 | 0.460 / 0.425 | 0.289 / 0.328 | 0.257 / **0.276** | 0.372 / 0.409 | 0.633 / 0.387 | 2.407 / 0.784 | 1 | 2.78 |
> | RobustTSF | 0.508 / 0.475 | 0.450 / 0.448 | 0.456 / 0.421 | 0.294 / 0.340 | 0.287 / 0.302 | 0.380 / 0.401 | 0.884 / 0.474 | 0.574 / 0.430 | 2 | 4.59 |
> | **Ours** | 0.465 / **0.442** | 0.383 / 0.402 | **0.409** / **0.402** | **0.281** / **0.320** | 0.260 / 0.282 | 0.368 / 0.415 | 0.654 / 0.380 | 0.554 / 0.427 | **20** | **2.05** |

---

### Meta-Review · Area_Chair_YcNS · 2026-01-06

**Summary:**

The following main concerns are raised by the reviewers:

1. Reviewer kTBq and FyDr have concerns on novelty and contribution.

2. Reviewer kTBq and 3Xag also has concerns on overall performance, where the empirical results suggest that RESAM does not consistently achieve state-of-the-art performance across all benchmarks.

3. Some missing baselines commented by reviewers FyDr, YX6M, and 3Xag

4. Reviewer YX6M also raised concerns about the possible illogical motivation of the paper.

**Reviewer Concerns:**

After the rebuttal, I think for the above concerns, concern 1 and 2 are only partially addressed, where I think the contribution and novelty is not significant enough for ICLR, and the performance compared to baselines are not good enough.

After adding the baselines, I think concern 3 is addressed. And I think concern 4 is addressed.

**Reviewer Scores:**

After the rebuttal, due to the partial address of the concerns 1 & 2, I think all reviewers will likely remain their score.

---

### Decision · Program_Chairs · 2026-01-26

Reject